# SFPQ directs histone H3.3 deposition to R-loops in DNA repeats to protect genome stability

Alessandro Ferrando [1,5,8], Michele Giaquinto[1,8], Luisa M. R. Napolitano[2],
Giulia Canarutto [1,3], Alessandro Framarini [1], Alice Gambelli[1],
Pamela Veneziano Broccia [1,6], Annie Zappone[1], Eleonora Petti [1,7],
Chiara Boncristiani[1], Andrea Parlante[1,3], Silvia Onesti [2], Silvano Piazza [1,3],
Roberta Benetti [4] ✉ & Stefan Schoeftner [1] ✉

R-loops are three-stranded nucleic acid structures composed of an RNA:DNA hybrid duplex and a displaced single-stranded DNA loop. Unscheduled or persistent R-loops drive genome instability by creating conflicts with transcription and replication. Up to 75% of the human genome comprises repetitive DNA elements that are prone to R-loop formation. We show that the RNA binding protein SFPQ suppresses R-loop mediated replication stress and DNA damage at repeat elements such as telomeres, (peri)-centromeres, LINE-1 and SINE elements. SFPQ exhibits in-vitro R-loop binding activity, associates with chromatin containing R-loops, and recruits the histone H3.3 specific chaperon DAXX to preserve a correct nucleosome template that counteracts R-loop accumulation. Loss of SFPQ results in DAXX displacement from repeat elements, reduced histone H3.3 incorporation, replication stress-mediated genome instability and the emergence of cytoplasmatic DNA. This leads to activation of innate immune signaling via the cGAS/STING pathway, ultimately correlating with improved survival of sarcoma patients.

R-loops are atypical three-stranded nucleic acid structures comprising a 60–2000 nucleotide RNA:DNA hybrid and a displaced single-stranded DNA loop[1,2]. R-loops are estimated to occupy up to 5% of the human genome and regulate diverse physiological processes, including chromatin organization, DNA methylation, transcriptional termination, immunoglobulin class switching, sister chromatid segregation, and mitochondrial DNA replication[3–5]. In contrast, unprogrammed or persistent R-loops pose a major threat to genome stability by generating transcription–replication conflicts, impairing DNA repair, and serving as substrates for mutagenesis[4,6]. R-loop formation

and resolution are orchestrated by multiple machineries acting at the RNA, DNA, and chromatin level. The THO complex, SRSF1, and related factors suppress hybridization of nascent transcripts to the DNA template[7,8]. Ribonucleases RNAseH1 and RNaseH2 degrade RNA components of R-loops[9]. ATP-dependent helicases such as DHX9, the Bloom's syndrome helicase (BLM), Aquarius (AQR), FANCM and members of the DExD/H family of RNA helicases or chromatin regulatory factors such as the FACT and SWI/SNF chromatin remodeling complex, INO80 and DAXX/ATRX histone chaperons prevent unprogrammed R-loops[4,10–14]. Finally, factors that control replication fork

[1]Dipartimento di Scienze della Vita, Università degli Studi di Trieste, Trieste, Italy. [2]Structural Biology Laboratory, Elettra—Sincrotrone Trieste S.C.p.A, Area Science Park Basovizza, Trieste, Italy. [3]Computational Biology, International Centre for Genetic Engineering and Biotechnology, Trieste, Italy. [4]Laboratory of Epigenomics, Department of Medicine, Università degli Studi di Udine, Udine, Italy. [5]Present address: Dipartimento di Scienze Cliniche e Biologiche, Università degli Studi di Torino, Orbassano, Italy. [6]Present address: Centro Nacional de Análisis Genómico (CNAG), Barcelona Science Park—Tower I, Barcelona, Spain. [7]Present address: Translational Oncology Research Unit, IRCCS—Regina Elena National Cancer Institute, Rome, Italy. [8]These authors contributed equally: Alessandro Ferrando, Michele Giaquinto. ✉e-mail: roberta.benetti@uniud.it; sschoeftner@units.it

stability—including BRCA1/2, Fanconi anemia proteins, topoisomerases or DNA damage signaling factors such as Ataxia telangiectasia and Rad3 related (ATR) prevent R-loop-mediated genome instability[15–22].

Privileged sites for R-loop-mediated genome instability are characterized by GC- and AT-skew, G-richness or arrangement in tandem repeats[23–26]. Accordingly, R-loops were found to be enriched at telomeric and (peri-)centromeric satellite repeats, simple/low-complexity repeats, rDNA repeat arrays and retroelements[25,27–34]. Mutations in canonical R-loop suppression factors induce instability of DNA repeats, a central feature of several genetic disorders[25]. Immunodeficiency, centromeric instability, and facial anomalies syndrome type I is marked by elevated R-loop levels and instability at (peri)centric, telomeric, and rDNA repeats[35]. Patients with Bloom syndrome or Ataxia-telangiectasia mutated deficiency exhibit instability of rDNA arrays[36]. Mutations or dysregulated expression of the *ATRX* or *DAXX* genes compromise retroelement stability, driving R-loop-mediated replication stress and increased recombination at telomere repeats[13,29,37–39]. The pervasive transcriptional activity across the human genome necessitates R-loop surveillance machineries that safeguard genome integrity within repetitive elements, which account for ~75% of the genome[40,41]. However, surveillance mechanisms that detect and resolve unprogrammed R-loops at repetitive elements remain poorly defined. SFPQ is a multifunctional RNA-binding protein that, together with NONO and PSPC1, belongs to the conserved DBHS (Drosophila behavior/human splicing) protein family[42]. DBHS proteins form homo- and heterodimers through their NOPS and RRM2 domains and are enriched in paraspeckles in vertebrates. Beyond paraspeckle assembly, SFPQ has been implicated in transcriptional regulation, splicing, DNA damage repair, stress responses, and telomere maintenance[43–51].

Here we demonstrate a critical role for the RNA-binding protein SFPQ in preventing aberrant R-loop formation across a broad spectrum of repetitive elements, including pericentromeric and centromeric satellite repeats, subtelomeres, telomeres, as well as LINE-1 and SINE elements. SFPQ displays binding specificity for R-loop structures in vitro, associates with chromosomal R-loops via its RRM domains, and employs its proline-rich domain to recruit the histone H3.3-specific chaperone DAXX. This recruitment preserves nucleosome architecture at repetitive elements, thereby counteracting R-loop–mediated genome instability and limiting the accumulation of cytoplasmic DNA. Disruption of this pathway induces genome instability in cancer cells and activates innate immune signaling, which correlates with improved survival in patients with sarcoma.

## Results

### SFPQ targets R-loops in repetitive elements
SFPQ suppresses R-loop formation at vertebrate telomeric repeats, thereby protecting chromosome ends from replication stress, DNA damage, and excessive recombination events[46]. Immuno-dot blot analysis of genomic DNA from SFPQ-depleted U-2 OS, H1299, MCF-7, and MCF-10A cells revealed a significant increase in steady-state DNA:RNA hybrid levels, consistent with a role for SFPQ in R-loop suppression at multiple loci (Supplementary Fig. 1a, b). To extend this link to repetitive elements, we performed RNA-FISH under non-denaturing conditions using probes derived from human $C_0t$-1 DNA, which is enriched for interspersed repetitive sequences[52]. Both the number and signal intensity of $C_0t$-1 RNA-FISH foci increased significantly following depletion of SFPQ or RNaseH1 in U-2 OS and H1299 cells (Fig. 1A, Supplementary Fig. 1c). This finding parallels the reported increase in focal TERRA staining intensity under conditions of elevated telomeric R-loop abundance induced by loss of SFPQ or RNaseH1[29,46]. Specificity of used short interfering RNAs for *SFPQ* and *RNaseH1* mRNA was validated by western blotting (Supplementary Fig. 1d, Supplementary Fig. 7j, see below). Importantly, $C_0t$-1 RNA-FISH signals in SFPQ- or RNaseH1–depleted U-2 OS cells co-localized with p-ATR

(Thr1989) and FANCD2, indicative of replication stress at repetitive elements (Fig. 1B, Supplementary Fig. 1e). Although $C_0t$-1 RNA-FISH lacks sufficient sensitivity to comprehensively monitor transcriptional activity across all repeat sequences, our findings demonstrate that loss of RNaseH1 or SFPQ induces aberrant RNA metabolism at repeat-containing loci, thereby promoting replication stress. Consistent with ATR activation, SFPQ depletion resulted in elevated phosphorylation of Chk1 and RPA32, as shown by western blot analysis (Supplementary Fig. 1f). The induction of replication stress upon SFPQ depletion is consistent with defects in S-phase and G2-M progression and previous reports (Supplementary Fig. 1g)[53–55]. SFPQ was found to accumulate at centromeric and telomeric repeats following experimental elevation of R-loop levels induced by transient RNaseH1 depletion or hydroxyurea treatment, as shown by co-immunofluorescence with CREST/SFPQ and TRF2/SFPQ antibodies (Fig. 1C, D; Supplementary Fig. 1h–j). Chromatin immunoprecipitation in RNaseH1-depleted U-2 OS cells further demonstrated recruitment of SFPQ to diverse classes of repetitive elements, including telomeric repeats, subtelomeres of Chr16p, Sat III, Sat II, centromeric α-satellite repeats, AluS, AluY, and LINE elements (Fig. 1E). Thus, elevated R-loop levels at major categories of vertebrate DNA repeats drive dynamic SFPQ recruitment. To assess whether SFPQ contributes to R-loop suppression at these loci, we transiently depleted SFPQ in U-2 OS cells and performed DNA:RNA immunoprecipitation (DRIP). Loss of SFPQ resulted in increased R-loop abundance across telomeric repeats, subtelomeres of Chr16p, Sat III, Sat II, centromeric α-satellite repeats, AluS, AluY and LINE elements, as well as at a known SFPQ binding site in the promoter of the *PD3A* gene[56] (Fig. 1F). Negative control regions in the *WRNIP* and *SSH* promoters did not show R-loop formation. Consistent with these findings, RNaseH1 depletion also led to elevated R-loop levels at all classes of repetitive elements and at the PD3A promoter (Supplementary Fig. 1k). Together, these data indicate that SFPQ is recruited to unscheduled R-loops at repetitive elements and exerts R-loop-suppressive activity.

### SFPQ has in vitro binding specificity for R-loops
We next examined the binding activity of SFPQ towards distinct, 5′-[6-FAM] labeled nucleic acid substrates, including single-, double- and triple-stranded structures, such as RNA:DNA hybrids, D-loops and R-loops (Supplementary data 1). Full-length human SFPQ carrying an N-terminal $His_6$–myc tag was purified as recombinant protein from *E. coli* and tested in electrophoretic mobility shift assays (EMSA). Native polyacrylamide electrophoresis revealed that recombinant SFPQ does not interact with dsDNA or dsRNA, but binds ssDNA and shows stronger affinity for ssRNA (Fig. 2A, B; Supplementary Fig. 2a). Binding to single-stranded, non-repetitive RNA was enhanced with increasing substrate length from 15 to 23 nucleotides, indicating a preference for longer RNA molecules (Supplementary Fig. 2b). Interestingly, SFPQ displayed markedly increased affinity for repeat-containing single-stranded RNAs, including $[UUAGGG]_4$ and $[GGAAU]_4$ repeats present in TERRA and pericentric human SatIII RNAs, respectively, as well as $[GAAA]_5$ repeats (Supplementary Fig. 2b). These findings indicate that SFPQ preferentially binds RNA repeats in vitro. Consistent with this observation, SFPQ was detected in eluates from TERRA RNA pull-down assays using nuclear protein extracts[46]. We next assessed the binding capacity of SFPQ towards atypical nucleic acid structures. Recombinant SFPQ did not interact with a fully paired DNA:RNA hybrid duplex or with dsDNA, but bound a non-repetitive dsDNA probe containing a central unpaired region (DNA bubble; Fig. 2B). Binding was more efficient with a synthetic R-loop probe lacking a protruding RNA tail (Fig. 2B), and was further enhanced with D-loop structures carrying 5′ or 3′ single-stranded DNA termini (Fig. 2C). Most efficient mobility shifts were observed using R-loop substrates containing single stranded 5′ or 3′ termini RNA termini that protrude from the loop structure (R-loop 3′tails, R-loop 5′tails; Fig. 2D, E;

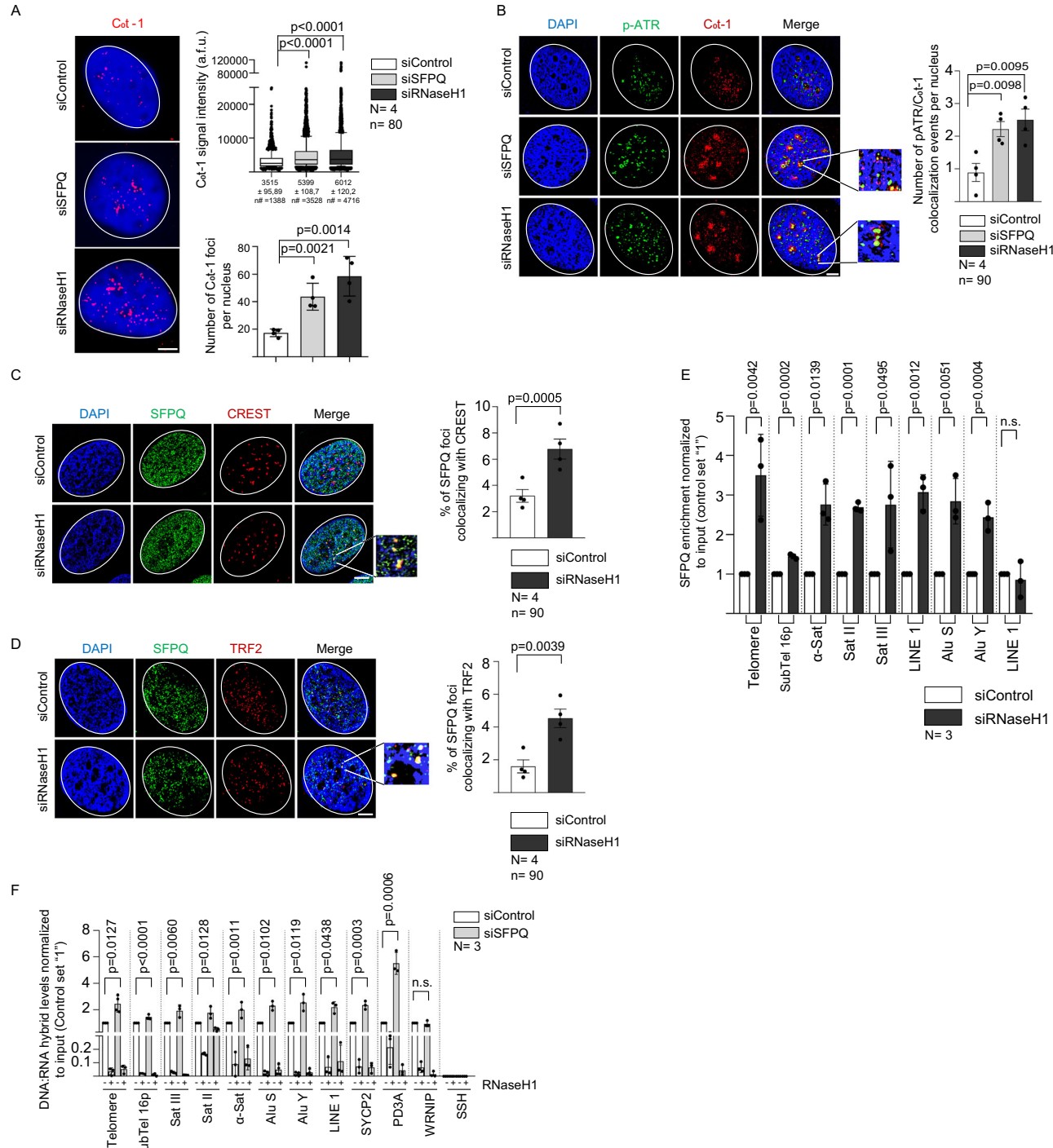

**Fig. 1 | SFPQ suppresses R-loop formation in repeat elements. A** $C_0t$-1 RNA-FISH performed in U-2 OS cells transiently transfected with indicated siRNAs (left panel). Quantification of focal $C_0t$-1 signal intensity (right, top panel) and number of $C_0t$-1 foci per nucleus (right bottom panel). **B** $C_0t$-1 RNA-FISH combined with anti-p-ATR (Thr1989) immunostaining performed in U-2 OS cells transiently transfected with indicated siRNAs. Images were obtained by super-resolution microscopy. Right panel, quantification of number of $C_0t$-1 foci co-localizing with p-ATR (Thr1989) per nucleus. **C** Combined immunofluorescence with anti-SFPQ and anti-CREST (centromere) antibodies in U-2 OS cells transfected with indicated siRNAs. Images were obtained by super-resolution microscopy. Right panel, quantification of co-localization events. **D** Representative images of combined immunofluorescence with anti-SFPQ and anti-TRF2 antibodies in U-2 OS cells transfected with indicated siRNAs. Images were obtained by super-resolution microscopy. Right panel,

quantification of co-localization events. **E** Anti-SFPQ ChIP-qPCR analysis using control and RNaseH1 depleted U-2 OS cells. Indicated sequence categories were amplified using specific PCR primer pairs. **F** Quantitative DRIP-PCR analysis after siRNA mediated SFPQ knock-down in U-2 OS cells. RNaseH1, treatment of genomic DNA with recombinant RNaseH1 prior to DRIP. Indicated repeat regions were amplified using specific primers. Mean values are shown, error bars indicate standard deviation. $N$ = number of independent experiments. $n$ = number of analyzed nuclei. An unpaired, two-sided Student's $t$ test was used to calculate statistical significance; $p$-values are shown. Scale bar (1 μm) applies to all images in respective immunofluorescence panels. DNA was stained using DAPI (4′,6-diamidino-2-phenylindole). Arrowheads and zoomed images indicate co-localization events. Source data are provided with this paper.

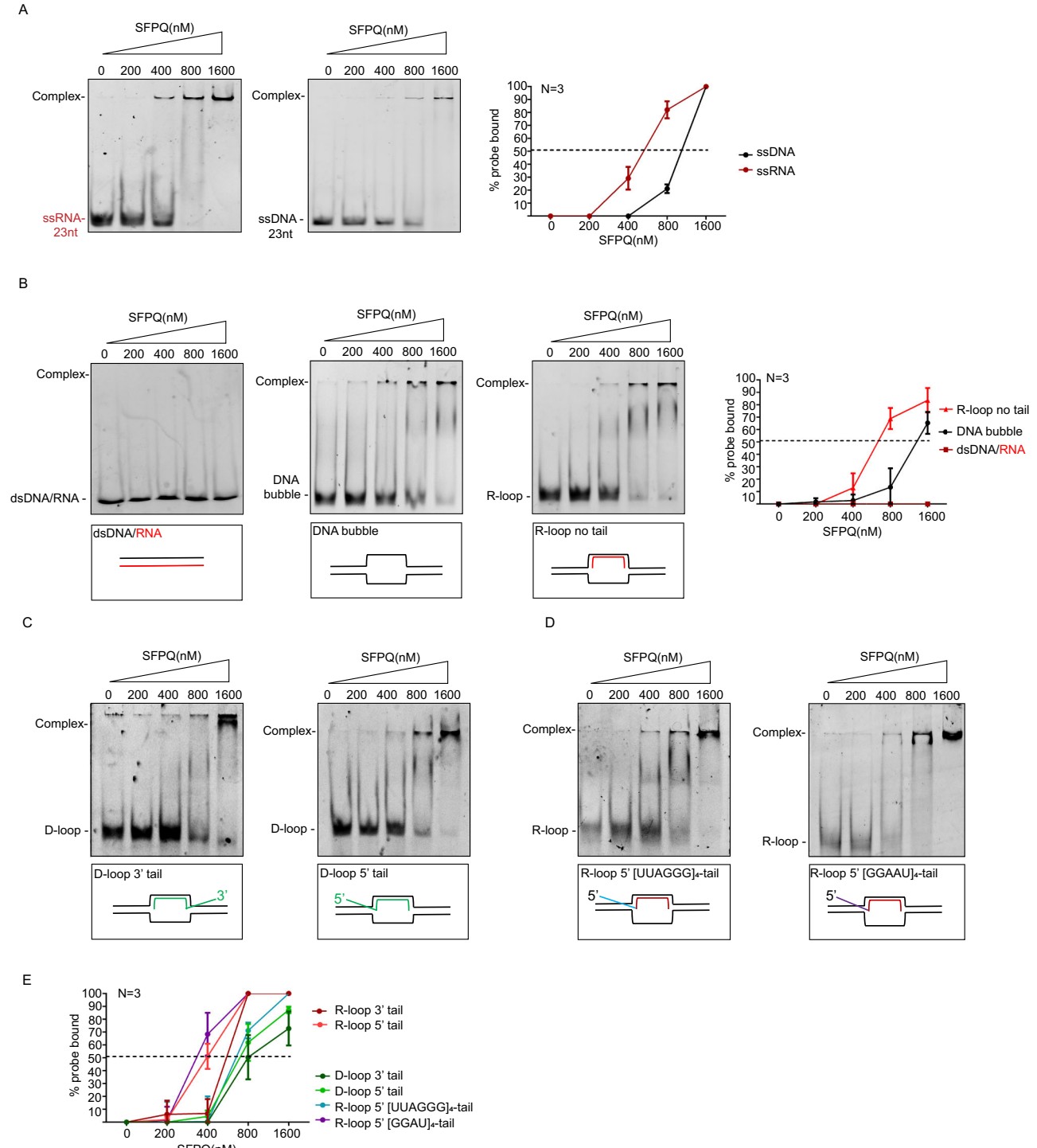

**Fig. 2 | SFPQ binds R-loop structures in vitro. A** EMSA showing recombinant SFPQ binding to ssDNA (left) and ssRNA (middle). Right panel, quantification of shifted ssRNA and ssDNA substrates. **B** Representative images of EMSA showing recombinant SFPQ binding to dsDNA:RNA duplexes (left), DNA bubble (center-left) and R-loop without protruding termini (center-right). Right panel, quantification of shifted nucleic acid substrates. **C** Representative images of EMSA showing recombinant SFPQ binding to D-loop substrate with protruding 3' ssDNA terminus (left) or 5' ssDNA terminus (right). **D** Representative images of EMSA showing recombinant SFPQ binding to R-loop substrate with protruding 5' ends composed of single stranded RNA repeats ([UUAGGG]₄ or [GGAAU]₅). **E** Quantification of EMSA assays shown in **C**, **D** and Supplementary Fig. 2c. **A**, **B**, **E** Data are represented as mean values; error bars indicate standard deviation; *N* = number of independent experiments. Source data are provided with this paper.

Supplementary Fig. 2c). To evaluate the contribution of these ssRNA protrusions to SFPQ binding, we performed competition assays with unlabeled ssRNA or dsDNA. As expected, dsDNA did not interfere with R-loop binding. Remarkably, tenfold molar excess of ssRNA was required to disrupt SFPQ association with R-loops containing ssRNA protrusions (Supplementary Fig. 2d). Together, these findings demonstrate that SFPQ exhibits in vitro specificity for three-stranded R-loop structures.

## SFPQ is a binding partner of the Death Domain Associated Protein (DAXX)

To date, no enzymatic activity has been attributed to SFPQ. To identify SFPQ-interacting proteins that may influence R-loop metabolism, we transiently overexpressed myc-tagged human SFPQ in the highly transfectable H1299 lung adenocarcinoma cell line and performed anti-myc immunoprecipitation. Control immunoprecipitations were carried out using myc-tagged Bax-interacting factor 1 (Bif-1)[57]. Coomassie staining revealed distinct bands in myc-SFPQ eluates, which were excised from the protein gel and analyzed by mass spectrometry, identifying a panel of candidate SFPQ-interacting proteins (Fig. 3A; Supplementary data 2–6). The detection of NONO, a previously reported SFPQ interactor, validated the specificity of our experimental approach (Fig. 3A; Supplementary data 6). Given its established role in genome stability, we directed subsequent analyses toward the Death Domain–Associated Protein (DAXX) (Supplementary data 2). DAXX is a multifunctional protein involved in apoptosis, senescence, gene transcription, and the DNA damage response[58]. In addition, DAXX functions as a histone chaperone, ensuring the incorporation of the histone variant H3.3 into repeat regions to preserve genome stability[37,59]. The SFPQ–DAXX interaction was validated by reciprocal co-immunoprecipitation of ectopically expressed myc-tagged SFPQ and HA-tagged DAXX in H1299 cells (Fig. 3B, C). Immunoprecipitation of endogenous proteins further demonstrated that a relevant fraction of total SFPQ and DAXX resides within a protein complex in U-2 OS cells (Fig. 3D). Direct interaction between SFPQ and DAXX was confirmed by pull-down assays using recombinant His$_6$–myc-tagged SFPQ and GST-tagged DAXX–HA proteins (Supplementary Fig. 3a, b). Western blotting further detected DAXX in reverse-crosslinked chromatin isolated from anti-SFPQ ChIP experiments (Supplementary Fig. 3c). Consistent with SFPQ knockdown data, siRNA-mediated depletion of DAXX in U-2 OS cells increased global R-loop levels and ATR phosphorylation (Supplementary Fig. 3d-f). This phenotype was rescued by inducible RNaseH1 expression, supporting the conclusion that both SFPQ and DAXX are critical for R-loop regulation in our model system (Supplementary Fig. 3D-F). The specificity of the S9.6 antibody was validated by pretreating fixed cells with RNaseT1, RNaseT3, or RNaseH1 (Supplementary Fig. 3e). Consistent with the observation that SFPQ is recruited to ectopic R-loops (Fig. 1C–F; Supplementary Fig. 1i-k), RNaseH1 knockdown increased DAXX localization at telomere repeats (Supplementary Fig. 3g). Additional immunofluorescence analyses revealed that SFPQ depletion in U-2 OS and H1299 cells produced a dispersed nuclear DAXX distribution, accompanied by reduced DAXX localization at telomeric and centromeric regions, both known R-loop hotspots (Fig. 3E, F; Supplementary Fig. 3m). Loss of SFPQ led to an ~20% increase in DAXX protein and mRNA levels in U-2 OS and H1299 cells (Supplementary Fig. 3h-k). DAXX is known to interact with ATRX to form a complex that enforces repressive chromatin architecture at repeat elements[37]. ATRX is epigenetically silenced in U-2 OS cells but expressed in H1299 cells (Supplementary Fig. 3i)[60]. Notably, SFPQ depletion in H1299 cells caused a pronounced reduction in steady-state ATRX protein levels, while only modestly affecting *ATRX* mRNA expression (Supplementary Fig. 3i–l). Treatment of SFPQ-depleted H1299 cells with MG132 restored ATRX protein levels (Supplementary Fig. 3n), indicating that ATRX is targeted for proteasomal degradation in the absence of SFPQ. We conclude that SFPQ regulates DAXX and ATRX function in cancer cells by directing DAXX localization and sustaining ATRX protein expression.

## SFPQ controls DAXX-dependent histone H3.3 deposition

To elucidate the role of SFPQ in orchestrating DAXX-dependent deposition of histone H3.3, we performed Chromatin Immunoprecipitation followed by sequencing (ChIP–seq) to map the genomic localization of SFPQ, DAXX, and histone H3.3 in control and SFPQ-depleted U-2 OS cells. In control cells, ChIP–seq identified 6375 peaks for SFPQ and 6,643 peaks for DAXX. Consistent with the observation that DAXX-associated chromatin is co-immunoprecipitated using SFPQ-specific antibodies (Supplementary Fig. 3c), DAXX occupancy was markedly reduced in SFPQ knockdown cells, with only 2816 peaks detected (Supplementary Fig. 4a). In control U-2 OS cells, SFPQ, DAXX, and histone H3.3 peaks were broadly distributed across all annotated sequence categories (Supplementary Fig. 4b, c). Notably, unlike SFPQ, a substantial proportion of DAXX and histone H3.3 peaks was enriched in regulatory regions, including promoters and transcription start sites (TSS; 9.54% and 5.24%, respectively), as well as CpG-rich domains (1.99% and 1.02%, respectively) (Supplementary Fig. 4c). To explore the potential role of SFPQ in genome-wide recruitment of DAXX, we focused on loci co-occupied by both factors. In control cells, 1331 of the 5168 total DAXX peaks overlapped with SFPQ binding sites (Supplementary Fig. 4d). SFPQ–DAXX co-occupied peaks were largely excluded from promoters, transcription start sites (TSS; 0.39%), and CpG-rich regions (0.26%), and were predominantly located within intergenic regions, introns, and repetitive elements including LINEs, SINEs, satellites, and simple repeats (Fig. 4A; Supplementary Fig. 4c). Consistent with immunofluorescence data, SFPQ depletion markedly altered the genome-wide distribution of DAXX (Fig. 4B). Specifically, DAXX occupancy at promoter–TSS regions increased from 9.54% in control cells to 21.12% following SFPQ knockdown, and from 1.99% to 5.34% in CpG-rich domains (Fig. 4B; Supplementary Fig. 4c). A comparable trend was found for 5′UTR and exonic sequences (Fig. 4B, Supplementary Fig. 4c). Increased DAXX occupancy at regulatory elements was paralleled by reduced frequency of DAXX peaks at intronic, LINE, SINE and intergenic sequences (Supplementary Fig. 4c, e). Although SFPQ depletion led to a marked reduction in the total number of DAXX peaks, significant log$_2$-fold changes in the remaining DAXX peaks were confined to promoter–TSS regions, CpG-rich sequences, intergenic domains, and LINE elements (Supplementary Fig. 4e). These findings underscore the requirement of SFPQ for proper genome-wide distribution of DAXX. To assess whether aberrant DAXX localization in SFPQ-deficient cells affects histone variant deposition, we examined H3.3 occupancy. Loss of SFPQ resulted in a pronounced increase in histone H3.3 peak numbers, rising from 29,869 in control cells to 55,388 following SFPQ knockdown (Supplementary Fig. 4a–c), suggesting dysregulated chromatin incorporation of H3.3. Despite the increase in H3.3 peak numbers following SFPQ depletion, we observed marked log$_2$-fold reductions in H3.3 peak height across all annotated sequence categories (Fig. 4C). A genome-wide area under the curve (AUC) analysis of log$_2$-fold changes further confirmed a global decrease in H3.3 chromatin occupancy in SFPQ-deficient cells (Supplementary Fig. 4a, f). These results support a model in which disruption of SFPQ–DAXX function redirects H3.3 deposition from discrete high-occupancy loci to a more diffuse, moderate-occupancy pattern across the genome. To validate the ChIP–seq findings, we performed ChIP–PCR on representative genomic regions previously shown to accumulate R-loops upon SFPQ loss (Fig. 1E). Consistent with impaired recruitment, SFPQ depletion led to reduced DAXX abundance at α-satellite, SatII, SatIII, (sub)telomeric regions, LINE1, AluY, and AluS elements (Fig. 4D, E). This redistribution was accompanied by a significant reduction in histone H3.3 abundance at affected loci, without changes in canonical histone H3 density (Fig. 4D, E). These effects were recapitulated at the promoters of *PD3A* and *SYCP2*, previously identified as SFPQ and DAXX target genes, respectively (Fig. 4F)[56,61]. In contrast, the R-loop-resistant *WRNIP* promoter exhibited no changes in chromatin marks following SFPQ depletion (Fig. 1F; Supplementary Fig. 1k, 4g). Furthermore, the *SSH* promoter, which lacks SFPQ binding, showed no detectable DAXX or H3.3 occupancy, reinforcing the specificity of our ChIP assays (Supplementary Fig. 4g). Importantly, SFPQ depletion did not affect total histone H3.3 protein levels, ruling out limited H3.3 availability as a contributing factor to the observed chromatin changes

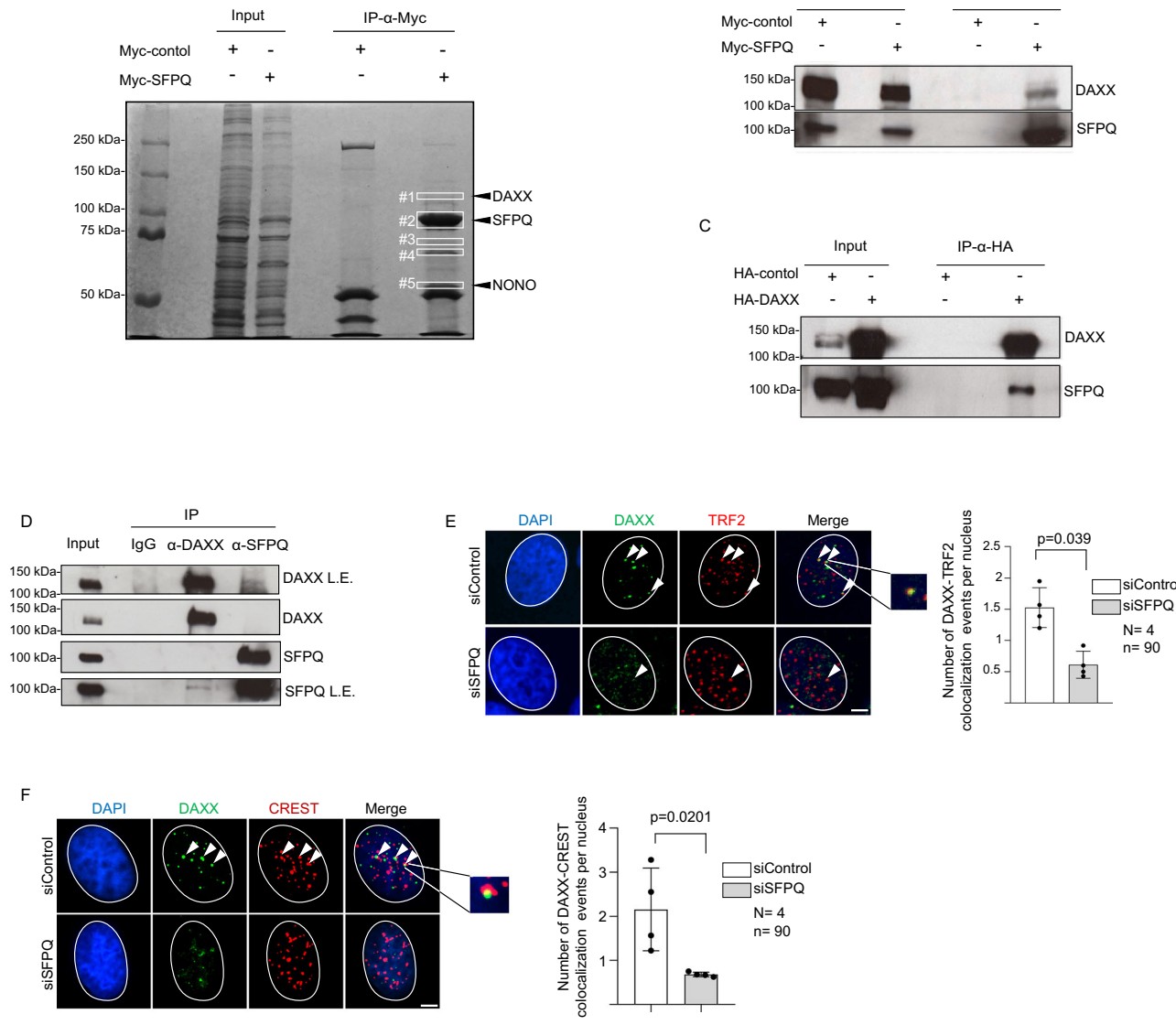

**Fig. 3 | SFPQ interacts with the histone H3.3 chaperone DAXX. A** Anti-SFPQ immunoprecipitation of ectopically expressed myc-tagged SFPQ followed by SDS-PAGE, Coomassie staining and identification of interacting proteins by MALDI-TOF mass spectrometry. **B** Immunoprecipitation of overexpressed myc-tagged SFPQ in H1299 cells. **C** Immunoprecipitation of overexpressed HA-DAXX in H1299 cells. **D** Co-immunoprecipitation of endogenous SFPQ and DAXX proteins in U-2 OS cells using specific antibodies. L.E., low exposure. **E** Combined immunofluorescence using anti-DAXX and anti-TRF2 antibodies in U-2 OS cells transfected with indicated siRNAs. Right panel, quantification of DAXX-TRF2 co-localization events. **F** Representative images of combined immunofluorescence with anti-DAXX and

anti-CREST (centromere) antibodies in U-2 OS cells transfected with indicated siRNAs. Right panel, quantification of DAXX-CREST colocalization events. **A**–**D** representative images are shown. For quantifications, data represents the mean, error bars indicate standard deviation. $N$ = number of independent experiments. $n$ = number of analyzed nuclei. Arrowheads and zoomed images indicate co-localization events. An unpaired, two-sided Student's $t$ test was used to calculate statistical significance; $p$ values are shown. Scale bar (1 µm) applies to all images in respective immunofluorescence panels. DNA was stained using DAPI (4′,6-diami-dino-2-phenylindole). Source data are provided with this paper.

(Supplementary Fig. 4h). Together, these findings support a model in which SFPQ binds R-loops and facilitates DAXX-dependent deposition of histone H3.3 at repetitive genomic elements.

## The Proline-rich domain and RRM motifs of SFPQ are central for R-loop suppression

To dissect the molecular basis of the SFPQ–DAXX interaction, we transiently expressed full-length HA-tagged DAXX together with a panel of myc-tagged SFPQ deletion mutants lacking defined protein domains, followed by co-immunoprecipitation assays (Fig. 5A; Supplementary Fig. 5a). Notably, deletion of the intrinsically disordered proline-rich domain (residues 105–204) abolished the interaction with

full-length DAXX (Fig. 5A, B; SFPQΔP). Other SFPQ domains did not significantly contribute to DAXX binding (Supplementary Fig. 5a, b). By contrast, the proline-rich P-domain, when fused to GFP and a nuclear localization signal, was sufficient to interact with full-length DAXX (Fig. 5C; Supplementary Fig. 5c). The requirement for the P-domain was further validated in pull-down assays using recombinant DAXX and recombinant full-length or ΔP SFPQ proteins (Fig. 5D). Consistent with this, whole-exome sequencing of an osteosarcoma patient cohort identified a P-domain mutation involving a non-frameshift substitution of Tyr150 with Lys-Pro (T150ΔPK)[62]. Full-length SFPQ carrying this mutation (SFPQ nFS) displayed impaired binding to ectopically expressed HA-DAXX in co-immunoprecipitation assays (Fig. 5B, C;

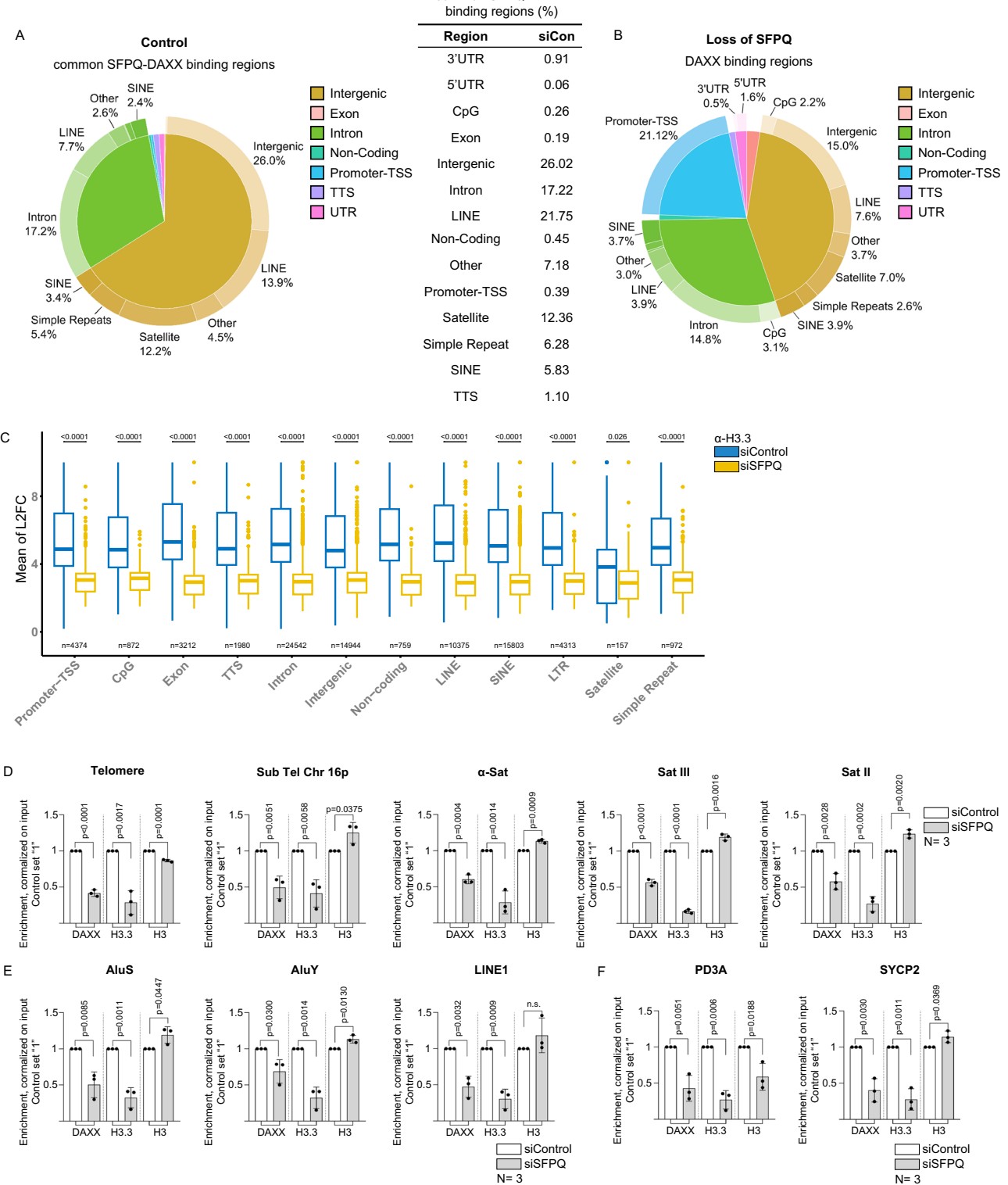

**Fig. 4 | SFPQ controls DAXX localization and deposition of histone H3.3. A** Pie chart representing percentage of SFPQ-DAXX common binding sites across indicated genome regions in U-2 OS cells transfected with control siRNAs (Control). Right panel, common SFPQ-DAXX peaks in control cells. **B** Pie chart representing the percentage of DAXX binding in indicated genome regions in SFPQ depleted U-2 OS cells (loss of SFPQ). **C** Box-plot representing H3.3 mean Log₂ fold change at reported genomic regions in control (blue) and SFPQ depleted (yellow) U-2 OS cells. *N*, number of peaks for reported genomic region for both conditions. *P*-values were calculated with Wilcox test and adjusted for False Discovery Rate. Center, median (50th percentile); box bounds, Q1 (25th percentile) and Q3 (75th percentile); whiskers, minimum and maximum non-outlier values (within 1.5 × IQR of Q1/

Q3); dots, outliers. Promoter-TSS Promoter-Transcription Starting Site, CpG CpG islands, TTS Transcription Termination Site, LINE Long Interspersed Nuclear Element, SINE Short Interspersed Nuclear Elements, LTR Long Terminal Repeat. Chromatin immunoprecipitation (ChIP) assay performed on U-2 OS cells transfected with indicated siRNAs. Data for (peri)centric and (sub)telomeric repeats (**D**) transposable elements (**E**) and positive control regions for SFPQ and DAXX are shown. For quantifications shown in (**D**–**F**), data represents the mean values, error bars indicate standard deviation. *N* = number of independent experiments. An unpaired, two-sided Student's *t* test was used to calculate statistical significance; *p*-values are shown. Source data are provided with this paper.

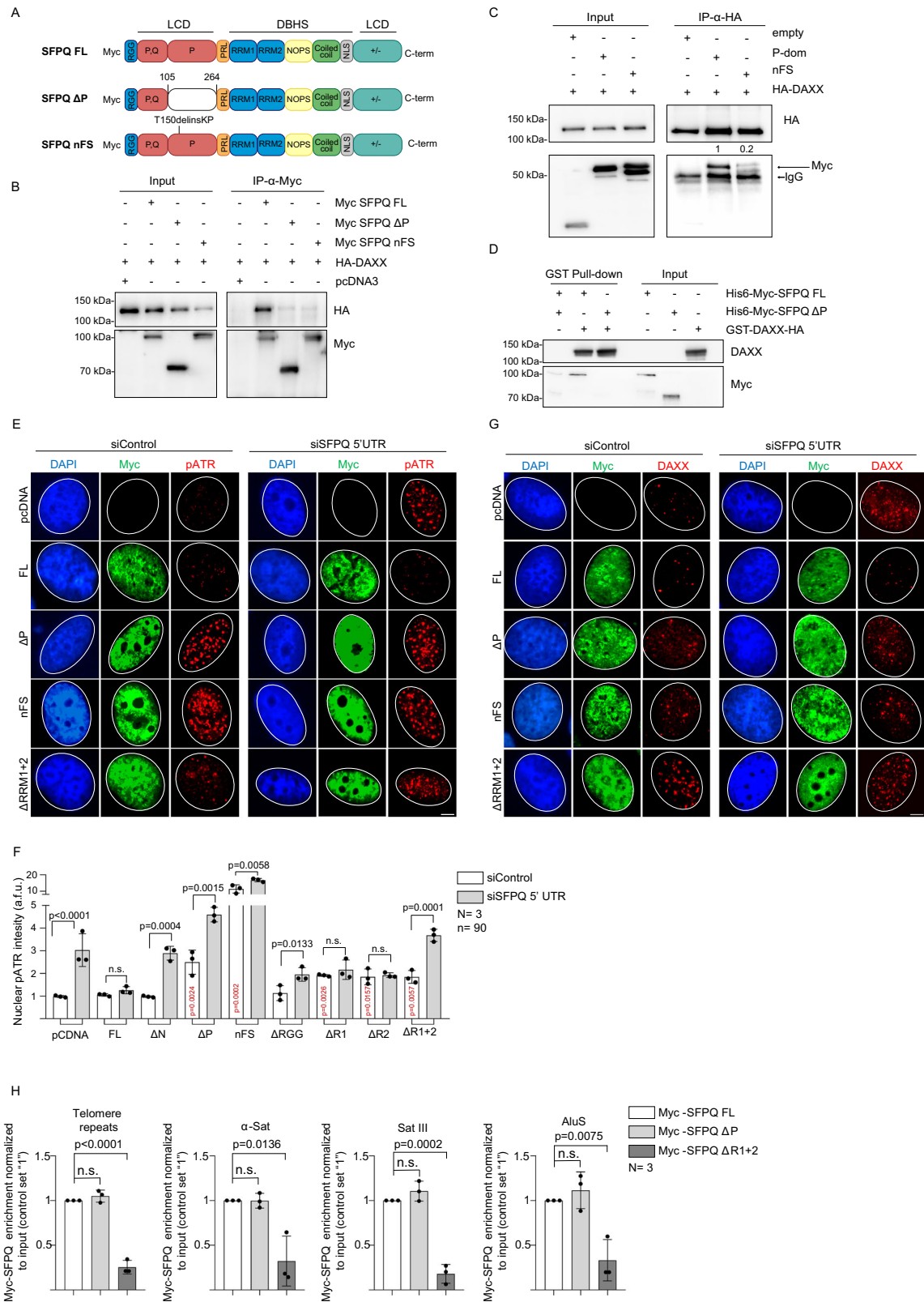

Supplementary Fig. 5c). We next examined the contribution of individual SFPQ domains to DAXX localization and the suppression of R-loop−induced replication stress. To this end, we generated U-2 OS cell lines stably expressing myc-tagged SFPQ deletion mutants lacking the 5' and 3' UTRs (Fig. 5A; Supplementary Fig. 5a). Endogenous full-length *SFPQ* was selectively depleted using 5'UTR-specific siRNAs, allowing assessment of truncated SFPQ variants on DAXX distribution and

replication stress (Supplementary Fig. 5d). In control cells stably transfected with a pcDNA control vector and subsequently treated with 5'UTR-targeting siRNAs, DAXX became delocalized and p-ATR (Thr1989) foci emerged, consistent with replication stress (Fig. 5E, F). As expected, ectopic expression of full-length myc-tagged SFPQ rescued phenotypes associated with depletion of endogenous SFPQ (Fig. 5E, F). In contrast, ectopic expression of myc-tagged ΔP or nFS

**Fig. 5 | Specific SFPQ domains mediate DAXX recruitment and R-loop binding.**
**A** SFPQ protein domains. Full-length (FL) SFPQ is composed by the arginine-glycine-glycine (RGG) box (blue); N- and C-terminal low-complexity domains (LCDs); proline/glutamine-rich subdomain (P,Q, red); proline-rich domain (P, red); PR linker (PRL, orange); DBHS conserved region (DBHS) containing RNA-recognition motifs (RRM1 and RRM2, blue), NonA/paraspeckles (NOPS) domain (yellow), the coiled-coil domain (green), a nuclear-localization sequence (NLS) (gray); and the +/- domain (bright green). Deleted domain are reported as white boxes; number refer to amino acid positions; patient mutation in nFS construct is indicated. **B** Co-immunoprecipitation of myc-tagged SFPQ versions and HA-tagged DAXX in U-2 OS cells. **C** Co-immunoprecipitation of myc-tagged SFPQ P-domain versions (myc-P-dom or myc-nFS) and HA-DAXX in U-2 OS cells. Arrowheads indicate the position of immunoglobulin bands; numbers indicate immunoprecipitation efficiency. **D** Pull-down assay of recombinant GST-DAXX-HA with recombinant His$_6$-myc-SFPQ FL or His$_6$-myc-SFPQ ΔP using anti-GST beads. **E** Representative images of combined immunofluorescence with anti-myc and anti-p-ATR(Thr1989) antibodies after transient expression of wild-type and mutant SFPQ versions

lacking the SFPQ 5'UTR and 3'UTR sequences. Endogenous SFPQ was knocked down using 5'UTR-specific siRNAs. Myc-positive cells were considered for quantification. **F** Quantification of pan-nuclear p-ATR in myc-positive cells described in (**E**). **G** Representative images of combined immunofluorescence with anti-myc and anti-DAXX antibodies in U-2 OS cells stably expressing SFPQ mutants. Endogenous SFPQ was knocked down using 5'UTR-specific siRNAs. **H** Anti-myc ChIP-qPCR analysis on U-2 OS cells transfected with indicated wild-type and mutant SFPQ expression constructs. Endogenous SFPQ was knocked down using 5'UTR-specific siRNAs. **B**–**D** representative images are shown. In quantifications, data represents the mean; error bars, standard deviation. $N$ = number of independent experiments; $n$ = analyzed nuclei. Statistical test, unpaired, two-sided Student's $t$ test. Scale bar (1 μm) applies to all images in respective immunofluorescence panels. DNA was stained using DAPI (4',6-diamidino-2-phenylindole). **F**; $p$-values (in black) refer to control siRNA and siSFPQ-transfected cells; $p$-values (in red) refer to U-2 OS cells stably transfected with a control pcDNA versus cells expressing mutant SFPQ versions and were transfected with control siRNAs. Source data are provided with this paper.

---

SFPQ variants resulted in elevated basal replication stress, which was further exacerbated upon siRNA-mediated knockdown of endogenous SFPQ, consistent with a dominant-negative effect of the mutant proteins (Fig. 5E, F). Notably, a subset of p-ATR foci induced by ΔP or nFS SFPQ colocalized with telomeres (Supplementary Fig. 5e). This is in agreement with the role of wild-type SFPQ in suppressing R-loop–associated replication stress at telomeres[46]. Replication stress induced by ΔP or nFS SFPQ variants was accompanied by DAXX delocalization (Fig. 5G; Supplementary Fig. 5g). Similarly, expression of myc-tagged SFPQ lacking the RNA-binding motifs (ΔRRM1 + 2) elevated basal replication stress, which was further exacerbated by siRNA-mediated depletion of endogenous SFPQ and likewise coincided with DAXX delocalization (Fig. 5E, F; Supplementary Fig. 5f, g). U-2 OS cells expressing single ΔRRM1 or ΔRRM2 SFPQ deletion constructs exhibited a modest but significant increase in replication stress, which did not further intensify upon depletion of endogenous SFPQ. By contrast, ΔRGG SFPQ expression did not elevate basal replication stress and induced only minor changes in p-ATR (Thr1989) foci following loss of endogenous SFPQ (Fig. 5E, F; Supplementary Fig. 5f). To assess the role of the P, RRM1 and RRM2 domains in SFPQ recruitment to chromatin, we transiently transfected U-2 OS cells with control, myc-tagged full-length, ΔP or ΔRRM1 + 2 SFPQ constructs while depleting endogenous SFPQ. Anti-myc ChIP assays revealed that the ΔRRM1 + 2 variant failed to localize to R-loop–rich regions, including telomeres, α-satellite, SatIII, SatII, and AluS sequences (Fig. 5H). By contrast, the P-domain was dispensable for chromatin loading (Fig. 5H). Together, these findings establish RRM1 and RRM2 as essential for SFPQ recruitment to R-loop hotspots, whereas the P-domain is specifically required for DAXX recruitment.

## Loss of SFPQ drives R-loop-mediated genome instability and mitotic defects

We next assessed the downstream consequences of unprogrammed R-loops by performing ChIP analysis in SFPQ-depleted U-2 OS cells. Loss of SFPQ from chromatin resulted in increased accumulation of ATR, RPA32 and γH2AX at telomeres, subtelomeres, α-satellite, SatII, SatIII and SINE/LINE repeats, underscoring the central role of SFPQ in suppressing replication stress and DNA damage at repetitive regions (Fig. 6A, B; Supplementary Fig. 6a, d). This effect was accompanied by recruitment of FANCD2 and RAD51, indicative of local activation of DNA repair pathways that counteract genomic instability (Supplementary Fig. 6b, c). Phenotypes associated with loss of SFPQ function were reproduced in independent H1299 cells (Supplementary Fig. 6e). Notably, a recent study showed that aberrant R-loop regulation at centromeres leads to mitotic defects, providing a compelling model to interrogate the role of SFPQ in safeguarding genome stability[63].

We analyzed R-loop formation and replication stress at centric and pericentric repeats using native metaphase chromosome spreads. Loss of SFPQ resulted in pronounced accumulation of activated ATR (pATR-Thr1989) and increased phosphorylation of RPA32 at Ser33 (pRPA32Ser33) at centromeres, indicative of replication stress (Fig. 6C, D). This effect was accompanied by accumulation of pericentric SatIID transcripts at centromeres of selected chromosomes, as detected by RNA-FISH (Fig. 6E). Phenotypes observed in SFPQ-deficient cells were reproduced in RNaseH1 knockdown cells (Fig. 6C–E). To further investigate mitotic progression, we performed time-lapse microscopy following transient SFPQ depletion in U-2 OS and H1299 cells stably expressing GFP-tagged histone H2B. Loss of SFPQ markedly increased the incidence of chromatin bridges, multilobular cells and micronuclei (Fig. 6F; Supplementary Fig. 6f; Supplementary Movie 1). To assess the impact of R-loops on chromosome stability, we next analyzed mitotic chromosome defects using Giemsa-stained metaphase spreads. U-2 OS and MCF-10A cells were transiently depleted of SFPQ and allowed to incorporate BrdU during two rounds of semi-conservative DNA replication. Metaphase spreads were subsequently UV-irradiated prior to Giemsa staining, enabling discrimination of sister chromatids by differential staining intensity. Loss of SFPQ increased chromatid breaks and reciprocal sister chromatid exchanges (SCEs) in both cell models (Fig. 6G; Supplementary Fig. 6g). As expected, these phenotypes were accompanied by alterations in cell-cycle progression (Supplementary Fig. 1g; Supplementary Fig. 6h). Collectively, these findings establish SFPQ as a key suppressor of R-loop–mediated genome instability.

## SFPQ-dependent R-loop resolution suppresses innate immunity pathways in human sarcoma

To investigate the physiological consequences of SFPQ loss of function, we performed RNA sequencing (RNA-seq) in U-2 OS cells subjected to SFPQ knock-down. Principal component analysis revealed marked alterations in the transcriptomic landscape (Supplementary Fig. 7a). Differential expression analysis identified 1268 genes significantly upregulated and 1163 genes downregulated upon SFPQ depletion (Supplementary Fig. 7b, c). Gene ontology enrichment demonstrated a pronounced activation of pathways associated with cytokine production and signaling, immune responses, antiviral defense, and innate immunity (Fig. 7A; Supplementary data 7).

Network analysis revealed activation of gene expression programs associated with epidermal development and secretion, alongside a 29-gene signature linked to innate immunity encompassing antiviral responses, cGAS–STING-mediated interferon activation, and inflammatory pathways (Fig. 7B). Enhanced interferon signaling provides a mechanistic basis for the elevated DAXX expression observed in SFPQ-depleted cells (Supplementary Fig. 3h, I)[64]. The presence of cytoplasmic DNA species indicates activation of the cGAS–STING axis,

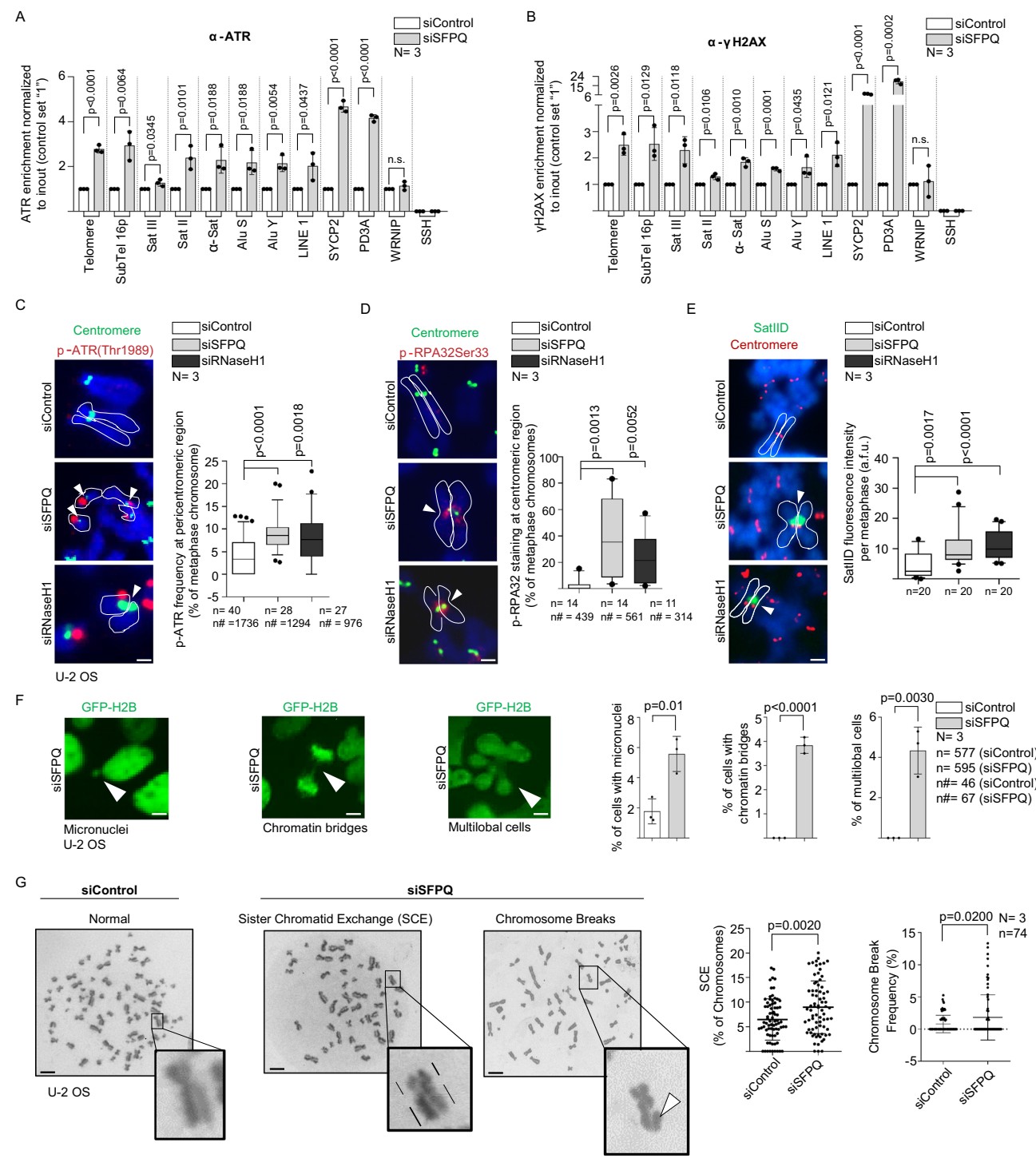

culminating in IRF3 and NF-κB transcriptional activity. This cascade drives interferon responses and the production of pro-inflammatory cytokines, ultimately reshaping the tumor microenvironment[65,66]. Independent quantitative RT-PCR and western blot analyses confirmed the upregulation of canonical target genes associated with the cGAS-STING axis, interferon signaling, inflammation, and NF-κB activation in both, U-2 OS and H1299 cells following *SFPQ* knock-down (Fig. 7C, D; Supplementary Fig. 7d, e). These findings underscore a broad role for SFPQ in restraining innate immune pathways. Immunofluorescence revealed that depletion of SFPQ or DAXX increased the frequency of micronuclei, which co-stained with cGAS, consistent with the exposure of cytoplasmic DNA (Fig. 7E). The presence of cytoplasmic DNA in

SFPQ-deficient U-2 OS cells was further validated by immunofluorescence using anti-dsDNA antibodies (Supplementary Fig. 7f). Transient depletion of SFPQ in U-2 OS and H1299 cells resulted in robust phosphorylation of the transcription factor IRF3, a key driver of interferon-stimulated gene expression (Fig. 7F; Supplementary Fig. 7g). Immuno-DNA FISH analysis further demonstrated that SFPQ loss increased the frequency of micronuclei stained with CREST antibodies, pericentric SatIID probes, or telomeric repeat probes, suggesting that enhanced R-loop-mediated genome instability at these regions may contribute to micronuclei formation (Supplementary Fig. 7h). To directly assess the role of SFPQ in micronuclei formation, we utilized U-2 OS cells ectopically expressing SFPQ deletion

**Fig. 6 | Loss of SFPQ mediates defects in mitotic progression.** ChIP on U-2 OS cells transfected with control and SFPQ-specific siRNAs. The graphs report the binding of ATR (**A**) and γH2AX (**B**) to the indicated region of interest. Representative images of native metaphase spreads obtained from control and SFPQ knock-down U-2 OS silenced with indicated siRNAs. Immunostaining was carried out using anti p-ATR(Thr1989) and anti-CREST (centromere) antibodies (**C**) or anti-p-RPA32(Ser33) and anti-CREST antibodies (**D**). Left panels, representative images. Arrowheads indicate position of p-RPA32Ser33/p-ATR staining in vicinity to centromeres. Right panels, quantification of positive staining at (peri)-centromeric regions. **E** Immuno-RNA FISH staining of native metaphase chromosomes derived on siRNA-transfected U-2 OS cells. Staining with anti-CREST antibody and a pericentric SatIID RNA-FISH probe is shown. Left panel, representative images. Right panel, quantification of SatIID signals at (peri)centric regions; Arrowheads, position of SatIID RNA-FISH staining in vicinity to centromeres. **F** Representative images of time-lapse microscopy using U-2 OS cells constitutively expressing GFP-tagged histone H2B after siRNA transfection. Left panels, representative images of observed mitotic defects. Right panels, relative quantifications of mitotic defects. **G** Giemsa-stained metaphases of U-2 OS cells transfected with indicated siRNAs to evaluate rates of sister chromatid exchange (SCE) and chromosome breaks. Left panels, representative images. Right panels, quantification of metaphase chromosome defects. Zoomed images, observed defects. Bars in data blots represent mean values, error bars indicate standard deviation. $N$ = number of independent experiments. An unpaired, two-sided Student's $t$ test was used to calculate statistical significance; $p$ values are shown. For (**C**–**E**), $n$= number of analyzed metaphase spreads, $n\#$ = number of analyzed chromosomes, graph shows median and 10–90 percentile; dots, outliers. Scale bars correspond to (**C**–**E**), 0.1 μm; (**F**) 1 μm and (**G**) 5 μm. Scale bars apply to all images in the respective immunofluorescence panels. **C**–**E** DNA (blue) was stained using DAPI (4′,6-diamidino-2-phenylindole). For (**F**), $n$= number of analyzed nucleus (used for micronuclei quantification), $n\#$ = number of analyzed cell divisions (used for chromatin bridges and multilobed cells qualifications). For **G**, $n$ = number of analyzed metaphases. Source data are provided with this paper.

constructs. Consistent with data on replication stress and chromatin recruitment, ectopic expression of myc-tagged ΔP or ΔRRM1 + RRM2 variants failed to rescue elevated micronuclei frequency observed in cells lacking endogenous SFPQ (Fig. 7G). Strikingly, expression of the patient-derived SFPQ nFS variant exerted a dominant-negative effect, driving micronuclei formation, which was further exacerbated upon depletion of endogenous SFPQ (Fig. 7G). These findings highlight the importance of the SFPQ P and RRM1 + RRM2 domains in maintaining genome stability. To directly test whether elevated R-loop levels underlie micronuclei and cytoplasmic DNA formation in SFPQ-deficient cells, we employed U-2 OS cells carrying a doxycycline-inducible mCherry-tagged RNaseH1 (ref. 67). Ectopic RNaseH1 expression suppressed both micronuclei formation in SFPQ- or DAXX-depleted cells and the activation of 10 out of 11 pro-inflammatory genes (Fig. 7H, I; Supplementary Fig. 7i). Notably, consistent with the reported role of SFPQ as a transcriptional repressor of IL8 (ref. 48) SFPQ depletion increased *IL8* expression in an RNaseH1-independent manner (Supplementary Fig. 7i). Although U-2 OS cells have been reported to express STING at low or undetectable levels (refs. 68–70), SFPQ knock-down induced robust STING phosphorylation. Moreover, treatment with the STING antagonist H151 suppressed the expression of pro-inflammatory genes, including *IFIT1*, *INFB1*, *CXCL10*, *IL1-alpha*, *IRF2*, and *IL12*, while leaving *CGAS*, *STING*, and *IL8* mRNA levels unaffected (Supplementary Fig. 7j, k).

To explore the clinical relevance of SFPQ-mediated regulation of innate immunity in human cancer, we analyzed patient survival data. Across pan-cancer cohorts, SFPQ expression showed no significant association with overall survival (Supplementary Fig. 7l). However, selective analysis of the TCGA sarcoma dataset revealed that elevated SFPQ expression correlated with reduced patient survival (Fig. 7J). In contrast, high expression of the innate immunity gene signature identified by RNA-seq in U-2 OS cells lacking SFPQ (29 genes including *IFIT1*, *ACTA2*, *UNC13D*, *IFI44*, *STING1*, *OASL*, *MX2*, *NLRP1*, *MLKL*, *RNF125*, *APOBEC3H*, *RTP4*, *PLA2G10*, *BIRC3*, *ABCC9*, *MMP12*, *DDX60*, *OAS2*, *PTPN22*, *IFIT2*, *IFIT3*, *AIM2*, *CCL5*, *CXCL10*, *RSAD2*, *HERC5*, *TSPAN32*, *ZBP1*, and *IL1A*) was associated with improved overall survival (Fig. 7B, K; Supplementary Fig. 8; Supplementary Table 1).

Consistent with these findings, SFPQ expression inversely correlated with the innate immunity gene signature in human sarcoma (Fig. 7L). Patients with low SFPQ and high innate immunity signature expression exhibited markedly improved overall survival compared to those with high SFPQ and low signature expression (Fig. 7M; Supplementary Fig. 9; Supplementary Table 2). Together with mechanistic evidence from cell lines, these data support a central role for SFPQ in suppressing R-loop-mediated genome instability at repetitive sequences, thereby preventing activation of innate immune pathways via the cGAS-STING axis and fostering a tumor-promoting microenvironment in sarcoma. Accordingly, the SFPQ-based gene expression signature provides a valuable tool for improved patient stratification and highlights the SFPQ–DAXX interaction as a potential therapeutic target to stimulate antitumor immunity.

## Discussion

Recent studies have shown that transcription of telomeric and centromeric regions, simple/low-complexity repeats, satellite repeats, rDNA arrays, and retroelements can promote R-loop formation[25,26,28,34]. We demonstrate that SFPQ safeguards genome stability by preventing R-loop accumulation across major classes of repetitive elements in the human genome. Loss of SFPQ led to pronounced R-loop accumulation at telomeres, subtelomeres, α-satellite, SatII, SatIII, LINE1, AluS, and AluY elements. Ectopic induction of R-loops triggered SFPQ recruitment to these repeat regions, indicating a dynamic role for SFPQ in sensing R-loop abundance. In vitro, SFPQ bound synthetic R-loops with protruding RNA termini but failed to interact with RNA:DNA duplexes, revealing a preference for three-stranded nucleic acid structures.

A comparable preference for R-loop structures over RNA:DNA duplexes has recently been reported for the Tudor domain-containing protein TDRD3, which recruits the RNA-dependent helicase DHX9 to resolve R-loops at gene promoters, the FANCI-FANCD2 complex, and ATRX[71–73]. We found that the RNA-recognition motifs RRM1 and RRM2 of SFPQ are required for its binding to R-loop-containing chromatin in U-2 OS cells. Consistent with our hypothesis that SFPQ functions as a recruitment factor for R-loop management machineries, we identified the replication-independent H3.3 histone chaperone DAXX as a novel SFPQ-interacting protein. DAXX has been reported to interact with the ATP-dependent helicase ATRX to incorporate the replication-independent histone variant H3.3 into nucleosomes at repetitive elements, including telomeres, thereby maintaining repressive chromatin structure[37,59,74–88]. Independent studies have implicated ATRX and DAXX in suppressing R-loops, although the mechanism of complex recruitment has remained unclear[13,39]. We show that DAXX binds to the proline-rich (P) domain of SFPQ; accordingly, expression of an SFPQ P-domain deletion construct recapitulated the replication stress and aberrant DAXX localization observed in SFPQ-depleted U-2 OS cells. These findings support a model in which SFPQ employs its RRM1 and RRM2 domains to bind R-loops at repetitive elements and subsequently recruits DAXX-dependent histone H3.3 chaperone activity via its P-domain to re-establish functional nucleosome templates. Notably, SFPQ depletion in H1299 cells was accompanied by reduced ATRX protein, but not mRNA, levels—recapitulating phenotypes observed in DAXX-deficient mouse embryonic stem cells[59]. Together, these results underscore the central importance of SFPQ in the assembly of a functional DAXX–ATRX complex. In line with the role of SFPQ in recruiting DAXX, ChIP-seq analysis revealed that loss of SFPQ causes

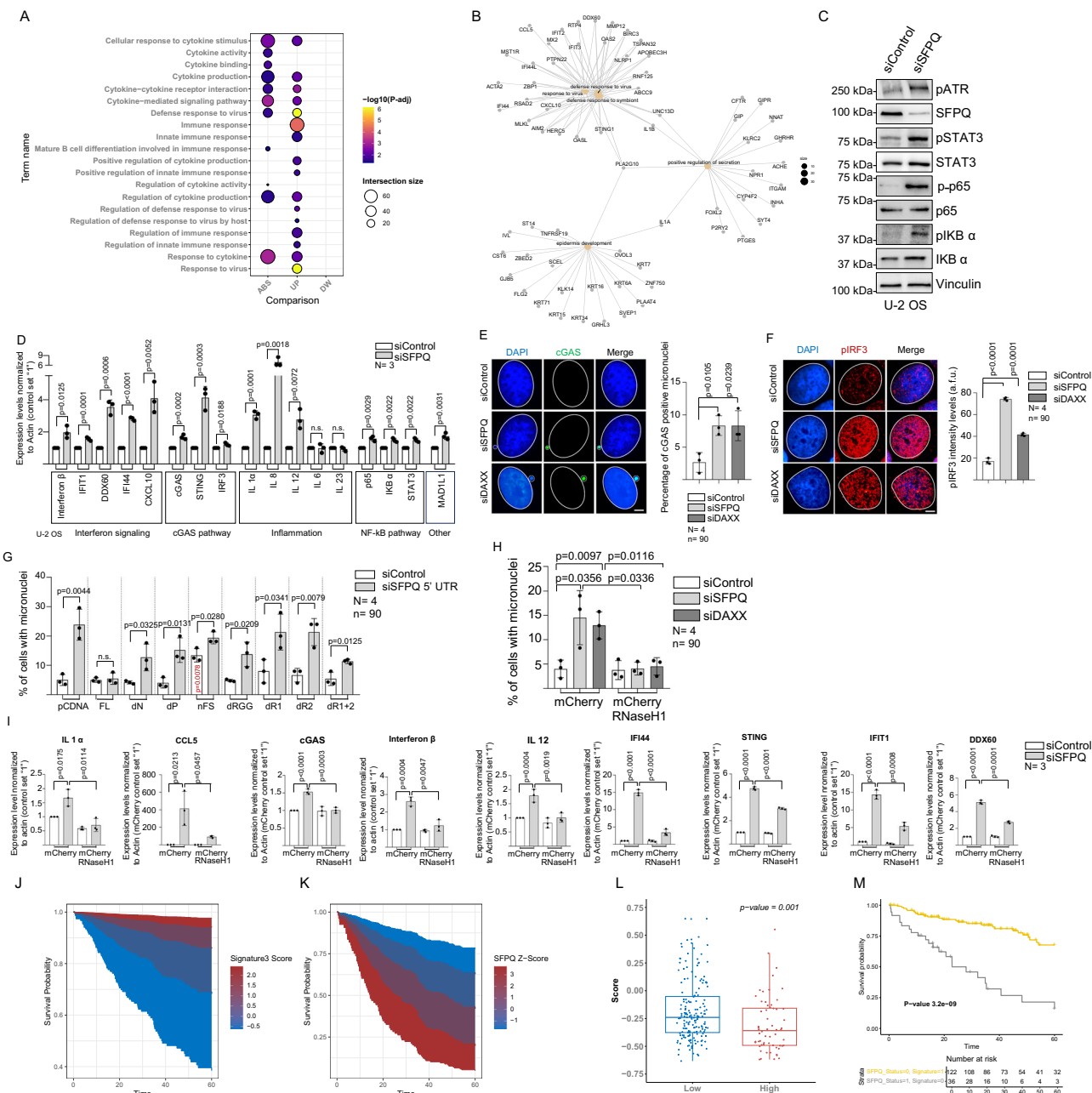

**Fig. 7 | Loss of SFPQ mediates activation of pathways of innate immunity.**
**A** Enriched terms in SFPQ knock-down U-2 OS cells. UP, upregulated; DW, down-regulated; ABS, absolute value terms. Circle size, number of genes; color code, −log₁₀ alterations (*p*-adjusted); enrichment *p*-values were computed using a one-sided Fisher's exact and Benjamini-Hochberg FDR correction. **B** Network representation for 5 enriched biological terms in siSFPQ U-2 OS cells. Orange nodes, term names; black dots, individual genes. **c** Western blot of siRNA-transfected U-2 OS cells. **D** Quantitative RT-PCR of indicated genes in siSFPQ U-2 OS cells. **E** cGAS immunofluorescence analysis. Left panel, representative images. Right panel, percentage of micronuclei with cGAS staining. **F** Immunofluorescence on siRNA-transfected U-2 OS cells. Left panel, representative images. Right panel, quantification of pan-nuclear pIRF3 staining. **G** Quantification of micronuclei in U-2 OS cells stably expressing my-tagged, full-length or mutant SFPQ. Endogenous SFPQ was depleted using 5′UTR-specific siRNAs. **H** Micronuclei in U-2 OS cells after transfection with indicated siRNAs and expression of mCherry-tagged RNaseH1 or control-mCherry. **I** Quantitative RT-PCR analysis in siRNA-transfected U-2 OS cells

expressing mCherry-tagged RNaseH1 or control-mCherry. Survival plot showing correlation between low SFPQ Z-score and improved patient overall survival (**j**) high signature score (Fig. 7b) and improved patient overall survival (**K**). **L** Correlation of expression between SFPQ and pro-inflammatory gene signature. Score based on mean Z-score trimmed at 20%. The p-value was obtained by Anova-test. Center, median (50th percentile); box bounds, Q1 (25th percentile), Q3 (75th percentile); whiskers, minimum and maximum non-outlier values; dots, individual samples. **M** Kaplan-Meier plot analysis based on innate immunity signature expression and survival. Yellow, low SFPQ expression (Status=0) and high signature expression (Signature=1); gray, high SFPQ expression (Status=1) and low signature expression (Signature=0). Patient numbers are shown. *P*-value was calculated using a two-sided log-rank (Mantel–Cox) test. **D–I** Data presented as mean values ± standard deviation. *N*, number of independent experiments; *n*, analyzed nuclei; statistical test, unpaired, two-sided Student's *t* test; scale bar (5 μm) applies to all images; DAPI, 4′,6-diamidino-2-phenylindole. **J–M** patient data were obtained from the TCGA pan-cancer sarcoma dataset. Source data are provided with this paper.

genome-wide DAXX delocalization and a global reduction in histone H3.3 abundance across all tested repeat elements. Thus, SFPQ depletion recapitulates the impaired H3.3 deposition at repetitive elements previously observed in model systems lacking DAXX or ATRX expression[37,59,74,77,79,80,82,84,86]. Notably, studies in mouse embryonic stem cells have shown that ATRX binding to trimethylated histone H3 lysine 9 (H3K9me3) recruits DAXX to endogenous retroviral elements (ERVs)[59,89]. Interestingly, DAXX has been shown to interact with a SETDB1-KAP1-HDAC1 protein complex in an ATRX-independent manner[90]. KAP1, together with the nucleosome remodeler SMARCAD1, directs SETDB1 deposition at ERVs to enforce H3K9me3 and HP1-mediated repression; in addition, SMARCAD1 has been reported to influence H3.3 incorporation[91–94]. Future studies should clarify the role of R-loops and SFPQ in coordinating ATRX-dependent and ATRX-independent functions of DAXX in mediating H3.3 deposition and epigenetic silencing through KAP1, SETDB1, and SMARCAD1. R-loop-mediated replication stress at repetitive elements in SFPQ-depleted cells led to the recruitment of R-loop-associated DNA damage repair factors, including RAD51 and FANCD2[72,95–97]. Failure of the cellular response to ectopic R-loop induction resulted in DNA breaks, elevated sister chromatid exchange (SCE) events, and defects in mitotic progression, accompanied by the accumulation of SatIID transcripts and replication stress markers at (peri)centric regions of native metaphase chromosomes. These findings are consistent with a recent study reporting mitotic instability driven by excessive R-loop formation and ATR activity at centromere regions[63]. In addition, multiple R-loop regulators—including BRCA1, BRCA2, RBMX, DHX38, SETX, FUS, DAXX, FA pathway components, and the BLM–TOP3A–RMI1–RMI2 (BTR) complex—have been implicated in maintaining (peri)centromere stability (for review ref. 25). Genome instability in cancer cells promotes the accumulation of cytoplasmic DNA species, including micronuclei, which activate the cGAS–STING pathway and drive the expression of pro-inflammatory cytokines and type I interferons, thereby stimulating immune cells and enhancing antigen presentation within the tumor microenvironment[65,66]. Consistently, mitotic instability induced by SFPQ depletion resulted in cytoplasmic dsDNA accumulation and increased micronuclei frequency, recapitulating phenotypes previously observed in cells lacking DAXX, ATRX, or H3.3[60,87,98,99]. Micronuclei arising in SFPQ-depleted cells matured into cytoplasmic DNA, as evidenced by cGAS recruitment, leading to full activation of the cGAS–STING pathway via NF-κB and IRF3 and induction of key innate immunity genes. Ectopic expression of RNaseH1 suppressed micronuclei formation and blocked activation of both cGAS–STING signaling and innate immunity gene expression. Consistently, treatment with the STING antagonist H151 recapitulated the suppression of innate immune gene expression. Thus, R-loop–mediated genome instability represents a primary source of cytoplasmic DNA production and cGAS–STING pathway activation in SFPQ loss-of-function cells. Nevertheless, we cannot exclude contributions from repeat RNAs escaping R-loop structures at DNA repeats, cytoplasmic R-loops, or chromatin alterations at promoters linked to interferon response genes[100,101]. Data from osteosarcoma whole-exome sequencing (WES) and TCGA sarcoma cohorts underscore the clinical relevance of SFPQ in maintaining R-loop–mediated genome stability. A recent WES study of 31 osteosarcoma patients identified two disabling SFPQ mutations[62]. Expression of a Tyrosine150-to-Proline/Lysine (T150PK) substitution within the P-domain forced DAXX delocalization, inducing replication stress and genome instability in a dominant-negative manner in our cell model. Gene expression analyses of the TCGA sarcoma dataset revealed that high SFPQ expression correlated with poor overall patient survival and reduced expression of the innate immunity gene signature. Conversely, low expression of the signature genes was associated with poor survival, which was further exacerbated in combination with high SFPQ expression. In conclusion, our data indicate that SFPQ

safeguards repeat stability by monitoring R-loop levels and recruiting DAXX-dependent histone H3.3 chaperone activity to maintain a stable chromatin template. In sarcoma patients, this pathway is predicted to suppress genome instability, limit the accumulation of cytoplasmic DNA species, and restrain activation of the cGAS–STING pathway, thereby fostering an immunosuppressive tumor microenvironment that promotes cancer aggressiveness. This scenario may significantly contribute to the limited applicability of immune checkpoint therapies in this tumor type. Future studies need to demonstrate whether targeting SFPQ-mediated R-loop suppression may promote innate immunity and increase the efficacy of immune pathway receptor and STING agonizts in programming a tumor-suppressive microenvironment in sarcoma[102].

# Methods

## Cell culture

Human cell lines used were obtained from ATCC and have not been cultured for longer than 2 months, minimizing the risk of cross-contamination or genetic drift. Based on the nature of the experiments and the short culture duration, authentication was not deemed necessary. Used cells were tested negative for mycoplasma contamination. Cells were cultivated at 37 °C, 5% $CO_2$. U-2 OS cells (osteosarcoma, ATCC HTB-96) were cultured in low glucose Dulbecco's modified Eagle's (DMEM) medium (Euroclone) with 10% fetal bovine serum (Corning), 1% L-glutamine (Gibco), 1% penicillin/streptomycin (Gibco). H1299 cells (carcinoma; non-small cell lung cancer, ATCC CRL-5803) were cultured in Roswell Park Memorial Institute (RPMI) medium (Euroclone) supplemented with 10% fetal bovine serum (Euroclone), 1% L-glutamine (Gibco), 1% penicillin/streptomycin (Gibco). MCF 10A (breast epithelial, ATCC CRL-10317) were cultured in high glucose Dulbecco's modified Eagle's (DMEM) medium and HAMS F12 (1:1 ratio, Euroclone), supplemented with 5% horse serum (Gibco), 1% L-glutamine (Gibco), 1% penicillin/streptomycin (Gibco), 10 μg/ml insulin (Sigma), 0.5 μg/ml hydrocortisone (Sigma), and 20 ng/ml EGF (Cell Guidance Systems). MCF-7 cells (ATCC HTB-22) were cultivated in Eagle's Minimum Essential Medium (EMEM) supplemented with 10% FBS (Gibco), 1% L-glutamine (Gibco), 1% penicillin/streptomycin (Gibco), and 10 μg/ml insulin (Sigma). HEK293 (human embryonic kidney, ATCC CRL-1573) cells were cultured in high-glucose DMEM medium (Euroclone) supplemented with 10% fetal bovine serum (Euroclone), 1% L-glutamine (Gibco), 1% penicillin/streptomycin (Gibco). U-2 OS TetON-RNaseH1 cells, kindly provided by Professor Sébastien Britton (IPBS, Toulouse), were maintained in the presence of puromycin (0.25 μg/ml; Sigma). Expression of mCherry-RNaseH1 was induced with doxycycline (2 μg/ml; Sigma) for 24 h. MCF-10A and U-2 OS cells transduced with retroviral vectors (pBABE or pBABE-GFP-H2B) were selected with puromycin (1 μg/ml). U-2 OS cells stably transfected with linearized control pcDNA or pCMVmyc-SFPQ deletion constructs were selected with puromycin (0.5 μg/ml). Where indicated, U-2 OS cells were treated with H151 (2 μM; Sigma) for 24 h, MG132 (20 μM; Sigma) for 4 or 8 h, or hydroxyurea (2 mM; Sigma) for 24 h.

## Plasmids

SFPQ vectors purchased from Addgene are listed in Supplementary data 9. SFPQ ΔP was generated by overlap extension PCR using primers containing KpnI and XhoI restriction sites for cloning the cDNA element into KpnI and XhoI sites of pCMV-Myc (ClonTech). pLV-eGFP was kindly provided by Prof. Giannino Del Sal (University of Trieste, Italy) and used for subsequent cloning. The derivative vector pAle was generated by inserting a fragment containing a nuclear localization sequence, a myc-epitope tag and multiple cloning site containing SpeI, NheI, XhoI sites using BsrGI. pAle allows the expression of fusion proteins containing a C terminal myc tag and N terminal eGFP SFPQ wild-type P domain or P domain containing T150delinsKP patient mutation (P dom nFS) were generated by PCR using primers

containing SpeI and XhoI sites for cloning the cDNA elements derived from FL SFPQ or SFPQ containing T150delinsKP patient mutation (SFPQ nFS) into SpeI-XhoI sites of pAIe. HA-DAXX was generated by PCR using primers containing EcoRI and XbaI restriction sites for cloning the cDNA element into EcoRI and XbaI sites of pcDNA3.1-HA (Addgene). SFPQ mutants have been generated using the Quick-Change II XL Site-Directed Mutagenesis Kit (Agilent) following the manufacturing instructions. Briefly, forward and reverse primers are designed to be complementary one each other, harboring the desired mutation in the middle of the sequence, allowing for PCR amplification of the entire plasmid. His$_6$-myc SFPQ and GST-DAXX-HA were cloned using LIC cloning protocol, as previously described[103]. Briefly, backbone vectors (pNIC-CTHF for His$_6$-Myc SFPQ, pGTvL1-SGC for GST-DAXX-HA) were enzymatically digested (pNIC-CTHF with BfuAI, pGTvL1-SGC with BsaI) and subsequently treated with T4 DNA polymerase to generate protruding ends. In parallel, PCR amplicons of desired inserts were treated with T4 DNA polymerase to generate protruding ends complementary to the one generated on the backbone. Backbone-insert were annealed at room temperature for 1 h, then used for transformation of competent E. coli. pNIC-CTHF was a gift from Opher Gileadi (Addgene plasmid # 26105). pGTvL1-SGC was a gift from Nicola Burgess-Brown (Addgene plasmid # 39188).

### Transfection, transduction, and production of stable cell lines

Transient transfection: specific siRNAs (Supplementary Table 3) were transfected using RNAi-MAX Lipofectamine (Invitrogen) at 30 nM final concentration, according to the manufacturer's suggestions. Plasmids were transfected using Lipofectamine 2000 (Invitrogen), according to the manufacturer's suggestions.

Generation of stable cell lines: U-2 OS cells stably expressing myc-tagged SFPQ deletion mutants were produced by co-transfecting U-2 OS cells with I) a linearized plasmid containing SFPQ expression construct (1 μg) and II) a linearized pcDNA vector containing puromycin resistance (0.1 μg). 48 h post-transfection puromycin (0.5 μg/ml) was used to select for stable integrants.

Viral transduction: For lentiviral particle production, HEK 293 cells were co-transfected using Polyethylenimine (PEI, Sigma, 1 mg/ml) with transfer vectors and psPAX2 and pMD2-env packaging vectors. Virus-containing supernatant was collected 72 h post-transfection, centrifuged 5 min at 500 g, and filtered with a 0.45 μm low-protein-binding filter to remove cellular debris. The virus-containing medium was added to target cells with the addition of Polybrene (Sigma, 6 μg/ml). Puromycin (1 μg/ml) was used to select for stable integrants.

### Western blotting

Whole-cell lysates were prepared as previously described and subjected to Western blotting according to standard procedures[104]. Briefly, cells were lysed in RIPA buffer (50 mM Tris pH 7.4, 250 mM NaCl, 1% Triton-X, 1% DOC, 0.1% SDS) for 1 h at 4 °C. Subsequently, material was sonicated using a Fisherbrand Model 120 Sonic Dismembrator for 20 s at 20% amplitude and centrifuged for 10 min (14.000 g, 4 °C). Protein quantification was performed using a Bradford assay (Bio-Rad). Samples were mixed with 6x sample buffer (0.375 M Tris, pH 6.8; 12% SDS; 60% glycerol; 0.6 M DTT; 0.06% bromophenol blue) to a final 1× concentration and denatured at 95 °C for 5 min. Proteins were resolved on homemade SDS-PAGE gels in Tris-Glycine running buffer (25 mM Tris, 190 mM glycine, 0.1% SDS) and transferred to nitrocellulose membranes (Cytiva) using transfer buffer (25 mM Tris, 190 mM glycine, 0.01% SDS, 20% methanol). Membranes were blocked with blocking buffer (5% BSA in 1xPBS containing 0.1% Tween-20) and incubated overnight at 4 °C with primary antibodies diluted in blocking buffer. The following day, membranes were washed three times with 1xPBS containing 0.1% Tween-20, incubated with HRP-conjugated secondary antibodies diluted in blocking buffer, and washed three times again with 1xPBS containing 0.1% Tween-20. Signal

detection was performed using Clarity Western ECL substrate (Bio-Rad) and imaged with an iBright FL1500 Imaging System (Invitrogen). Primary and secondary antibodies are listed in Supplementary data 10.

### Protein immunoprecipitation

Cell extracts were prepared using IP buffer (50 mM Tris, pH 8.0; 150 mM NaCl; 1% NP-40; 5 mM EDTA; 5% glycerol) supplemented with 1 mM PMSF (Sigma), 1× protease inhibitor cocktail (PIC; Sigma), and 1 mM NaF (Sigma). Extracts were passed several times through a small syringe to ensure efficient nuclear disruption. For each immunoprecipitation, 800 μg of whole-cell extract was incubated with specific antibodies (listed in Supplementary data 10) overnight at 4 °C. Antibody–antigen complexes were captured using 25 μl Protein A Dynabead slurry (Invitrogen) by rocking samples for 2 h at 4 °C. Beads were washed four times with IP buffer, and bound proteins were eluted in 2× sample buffer (125 mM Tris, pH 6.8; 0.5% SDS; 10% glycerol; 5% 2-mercaptoethanol). Eluted proteins were subsequently analyzed by Western blotting.

### Mass spectrometry

Cell lysates were prepared with ice-cold MS Lysis Buffer (50 mM Tris-HCl pH 8, 150 mM NaCl, 5 mM EDTA, 1 mM DTT, 5% glycerol, 1% NP-40, supplemented with 1 mM PMSF, 1 mM beta-glycerophosphate, 1 mM Na$_3$VO$_4$, 5 mM NaF, and 1x PIC). After centrifugation and preclearing, lysates were incubated at 4 °C with specific antibodies or rabbit IgG as negative control (Supplementary data 10). After 2 h, protein-A agarose-beads (Santa Cruz) were added and incubated overnight. The resin was then washed in MS Lysis Buffer four times and bound proteins were eluted in 2× sample buffer. Immunoprecipitated proteins were separated by SDS-PAGE and stained using mass spectrometry-compatible silver staining (SilverQuest™ Kit, Invitrogen) or Coomassie Brilliant blue R-250 (42660, Sigma). Selected lanes were excised from gels, de-stained and washed twice in 50% ACN with 50 mM ammonium bicarbonate, and dehydrated in 100% ACN. In-gel reduction was performed by incubating lanes in a 10 mM DTT, 100 mM ammonium bicarbonate solution for 30 min at 56 °C and alkylation in 55 mM iodoacetamide, 100 mM ammonium bicarbonate for 20 min at room temperature. Protein samples were dehydrated again in 100% ACN and digested by rehydrating gel pieces in 50 mM ammonium bicarbonate containing 4 ng/μL of trypsin (#V5111, Promega) overnight at 37 °C. Tryptic peptides were desalted and concentrated using ZipTip mC18 pipet tips (Millipore) and co-eluted onto the MALDI target in 1 μL of α-cyano-4-hydroxycinnamic acid matrix (5 mg/ml in 50% ACN,0.1% TFA). Mass spectra were acquired over a mass range of 800–4000 m/z (Nd:YAG laser at 355 nm, 40 Shots/Sub-Spectrum for 2000 Total Shots/Spectrum) by reflectron positive mode on an Applied Biosystems 4800 Proteomics Analyzer mass spectrometer (Applied Biosystems) and calibrated using a standard mixture (Mass Standards Kit, AB SCIEX). MS/MS spectra were acquired in positive mode (Nd:YAG laser at 355 nm, 40 Shots/Sub-Spectrum for 4000 Total Shots/Spectrum) and MS/MS calibration was achieved by using the default calibration method. Protein identifications were performed with the ProteinPilotTM software (version 2.0.1; Applied Biosystems) using the ParagonTM algorithm as the search engine. Each MS/MS spectrum was searched against the Uniprot/SwissProt database. The search parameters allowed for cysteine modification by iodoacetamide and biological modifications programmed in the algorithm (i.e., phosphorylations, semitryptic fragments, etc.). The detected protein threshold (ProtScore) in the software was set to 1.3 to achieve 95% confidence interval.

### Immunofluorescence

Cells were fixed in 4% paraformaldehyde (PFA) in 1xPBS for 15 min, followed by treatment with citrate buffer (0.1% w/v; 0.5% Triton X-100) for 5 min at room temperature. Subsequently, cells were then blocked

for 1 h in blocking solution (3% BSA, 0.1% Tween-20 in 1× PBS) and incubated with primary antibodies (listed in Supplementary data 10) diluted in blocking solution for 2 h at room temperature. After incubation, cells were washed three times for 5 min each with washing solution (0.5% BSA, 0.1% Tween-20 in 1xPBS) and subsequently incubated with secondary antibodies (Supplementary data 10) diluted in washing solution for 1 h at room temperature. Slides were washed twice for 10 min in washing solution, stained with DAPI (1 μg/ml; Sigma) in 1×PBS containing 0.1% Tween-20, and mounted with ProLong (Invitrogen). For immunofluorescence analysis using monoclonal S9.6 antibodies, cells were fixed and permeabilized with ice-cold methanol for 10 min followed by acetone for 1 min on ice, as previously described[105]. Blocking, antibody dilution and washing solutions were prepared as aforementioned, using 4xSSC instead of 1xPBS. To evaluate target specificity of the S9.6 antibody, U-2 OS cells fixed on glass slides were treated with RNaseT1 (NEB, 1U/μl; NEBuffer 2), RNaseT3 (NEB, 1U/μl; NEBuffer 2) or RNaseH1 (NEB, 1U/μl; NEBuffer 2) at 37 °C for 30 min. Samples were incubated at 65 °C for 20 min to inactivate the enzymes, washed in 1xPBS and processed following the immunofluorescence protocol. For immunofluorescent detection of cytosolic dsDNA with dsDNA-specific antibodies, cells were fixed in 4% PFA, permeabilized with 0.1% saponin (Sigma) in 1xPBS on ice for 5 min and subjected to the standard immunofluorescence protocol. For conventional immunofluorescence analysis, a Leica DM4000B microscope was used; quantitative analysis of immunofluorescence was performed with ImageJ (1.54i). A Student's *t* test was used to calculate statistical significance.

A ZEISS ELYRA 7 Structured Illumination Super-Resolution Microscope with ZEN Black software was used for high-resolution microscopy.

## Immuno-DNA FISH
Cells were washed in 1xPBS and fixed in 4% paraformaldehyde (PFA) in 1xPBS for 20 min at room temperature, followed by permeabilization with 1xPBS containing 0.1% Triton X-100 for 7 min at room temperature. Cells were then blocked with 5% BSA in 1xPBS at 37 °C for 20 min, and immunofluorescence staining was performed using specific antibodies (listed in Supplementary data 10). Antibody–antigen complexes were post-fixed at room temperature for 2 min with 4% PFA in 1xPBS, dehydrated through an ethanol series (70%, 90%, 100%; 5 min each), and air-dried. DNA-FISH was carried out in hybridization solution [10 mM Tris-HCl (pH 7.0) containing 70% (v/v)] deionized formamide, 0.25% (w/v) blocking reagent (Sigma), 5% (v/v) of $MgCl_2$ buffer solution (25 mM $MgCl_2$, 9 mM citric acid, 82 mM $Na_2HPO_4$), and 0.25 μg/ml FISH probe]. Samples were by denatured at 80 °C for 3 min. Subsequently, slides were incubated for 2 h at room temperature in a light-protected, humid chamber. Following hybridization, slides were washed in FISH washing solution [50% formamide, 10 mM Tris-HCl (pH 7.2), 0.1% BSA], stained with DAPI (1 μg/ml; Sigma) in 1xPBS containing 0.1% Tween-20, dehydrated again through an ethanol series (70%, 90%, 100%; 5 min each), air-dried, and mounted with ProLong (Invitrogen). $C_0t$-1 FISH probes were generated using Human $C_0t$-1 DNA (Invitrogen) using the FISH Tag DNA kit (Invitrogen), following manufacturer's instructions. 5'-TYE™563 labeled pericentromeric SATIID probes (GATCGAATGGAATCTGAATGGAA) were purchased from Panagene. Telomere probes (TelG-Cy5) were purchased by Panagene. A ZEISS ELYRA 7 Structured Illumination Super-Resolution Microscope with ZEN Black software was used for high-resolution microscopy.

## Chromatin immunoprecipitation
Chromatin immunoprecipitation was performed as previously described[106]. Briefly, $1 \times 10^7$ cells per immunoprecipitation were fixed in 1% formaldehyde (Sigma) in 1xPBS for 15 min at room temperature. The reaction was quenched with 140 mM glycine (final concentration) for 5 min at room temperature. Cells were washed twice with ice-cold

1xPBS and scraped into 1xPBS supplemented with 1× protease inhibitor cocktail (Sigma-Aldrich), 1 mM NaF, and 1 mM PMSF. Cell pellets were collected by centrifugation at 4000 *g* for 5 min at 4 °C. Cells were lysed in Lysis Buffer I (50 mM HEPES, pH 7.5; 10 mM NaCl; 1 mM EDTA; 10% glycerol; 0.5% NP-40; 0.25% Triton X-100) by rocking at 4 °C for 10 min, followed by centrifugation at 1500 *g* for 5 min at 4 °C. The pellet was resuspended in Lysis Buffer II (10 mM Tris-HCl, pH 8.0; 200 mM NaCl; 1 mM EDTA; 0.5 mM EGTA) and centrifuged at 1000 *g* for 10 min at 4 °C. The pellet was then resuspended in Lysis Buffer III (10 mM Tris-HCl, pH 8.0; 200 mM NaCl; 1 mM EDTA; 0.5 mM EGTA; 0.1% sodium deoxycholate; 0.5% N-lauroylsarcosine). Chromatin was sonicated using a BioRuptor (Diagenode) to obtain DNA fragments of 150−300 bp. Chromatin was diluted in Equilibration Buffer (10 mM Tris-HCl, pH 8.0; 100 mM NaCl; 1 mM EDTA; 1.66% Triton X-100; 0.166% sodium deoxycholate) at a 1:1.5 ratio (1 ml chromatin with 1.5 ml buffer) to a final volume of ~900 μl. An aliquot of processed chromatin was reverse-crosslinked (see below) and amount of recovered DNA was determined. A quantity of chromatin corresponding to 1 μg purified DNA was used in downstream immunoprecipitation experiments. Specific antibodies or rabbit/mouse control IgGs (Supplementary data 10) were added, and samples were incubated overnight at 4 °C with rocking. Protein A Dynabeads (Invitrogen) were blocked overnight in 1xPBS containing 0.5% BSA and 5 mM EDTA. The following day, antibody–antigen complexes were captured using 25 μl Protein A Dynabeads per sample, rocking for 2 h at 4 °C. Beads were sequentially washed in Low Salt Wash Buffer (20 mM Tris-HCl, pH 8.0; 150 mM NaCl; 2 mM EDTA; 0.1% SDS; 1% Triton X-100; 5 min), High Salt Wash Buffer (20 mM Tris-HCl, pH 8.0; 500 mM NaCl; 2 mM EDTA; 0.1% SDS; 1% Triton X-100; 5 min), and twice in TE buffer (10 mM Tris-HCl, pH 8.0; 1 mM EDTA; 5 min each). Proteins were eluted twice in Elution Buffer (10 mM Tris-HCl, pH 8.0; 300 mM NaCl; 5 mM EDTA; 0.5% SDS) at 37 °C for 15 min each, shaking at 850 rpm. Immunoprecipitated chromatin was de-crosslinked by adding 200 mM NaCl and incubating at 65 °C overnight. Samples were treated with RNase A (20 μg; Invitrogen) and Proteinase K (40 μg; Invitrogen) for 1 h at 45 °C, followed by phenol−chloroform extraction and ethanol precipitation. DNA was resuspended in TE buffer. Quantitative real-time PCR was performed using the CFX Connect Real-Time PCR Detection System (Bio-Rad) and iTaq Universal SYBR Green Supermix (Bio-Rad), according to the manufacturer's instructions. The PCR protocol for the amplification of telomere repeats has been described in ref. [107]. For data analysis, samples were normalized on input, IgG sample was subtracted as noise, siControl was set as 1. For ChIP-seq experiment, DNA sample were purified using QIAquick PCR Purification Kit (Qiagen). Purified DNA (10 ng) was used for the preparation of a TruSeq ChIP Library (Illumina). 2x150pb paired-end sequencing was performed on an Illumina HiseqX platform; data throughput: 18 Gb. Library preparation and sequencing service was performed by Macrogen Europe B.V.

## DNA:RNA hybrid immunoprecipitation (DRIP)
$1 \times 10^7$ cells were scraped into 1xPBS, centrifuged for 5 min at 4000 *g*, 4 °C, and lysed in Lysis Buffer (1% SDS, 20 mM Tris-HCl pH 7.5, 40 mM EDTA pH 8.0, 100 mM NaCl, $ddH_2O$). TE buffer (100 mM Tris-HCl pH 8.0, 10 mM EDTA pH 8.0) was added at a 1:1 ratio, and samples were supplemented with Proteinase K (Invitrogen) to a final concentration of 150 μg/ml. Samples were incubated at 37 °C overnight. Genomic DNA was purified by phenol−chloroform extraction, ethanol precipitation, and resuspended in TE buffer. Chromatin was sonicated using a BioRuptor (Diagenode) to obtain DNA fragments ranging from 500−1200 bp. For each sample, 4 μg of genomic DNA was either treated with recombinant RNase H1 (New England Biolabs; 3 units per 1 μg DNA at 37 °C) or left untreated. Protein A Dynabeads (Invitrogen) were blocked overnight in 1xPBS containing 0.5% BSA and 5 mM EDTA. The following day, antibody–bead conjugation was performed by incubating 2 μg of S9.6 monoclonal antibody (Kerafast) with 25 μl Protein A

Dynabeads per sample. Samples were rocked for 4 h at 4 °C in IP Buffer (50 mM HEPES/KOH pH 7.5, 140 mM NaCl, 5 mM EDTA pH 8.0, 1% Triton X-100, 0.1% sodium deoxycholate). Genomic DNA was then added to antibody-coupled beads and incubated overnight at 4 °C with rocking. The following day, samples were washed (5 min each) sequentially in IP buffer, Wash Buffer I (50 mM HEPES/KOH pH 7.5, 500 mM NaCl, 5 mM EDTA pH 8.0, 1% Triton X-100, 0.1% sodium deoxycholate), Wash Buffer II (10 mM Tris-HCl pH 8.0, 250 mM LiCl, 0.5% NP-40, 0.5% sodium deoxycholate, 1 mM EDTA pH 8.0), and twice in TE buffer (10 mM Tris-HCl pH 8.0, 1 mM EDTA). DNA was eluted twice in Elution Buffer (50 mM Tris-HCl pH 8.0, 10 mM EDTA pH 8.0, 1% SDS) at 65 °C for 15 min each, shaking at 850 rpm. DNA was purified by phenol–chloroform extraction, ethanol precipitation, and resuspended in TE buffer. Enrichment of RNA:DNA hybrids at target regions was validated by quantitative real-time PCR using the CFX Connect Real-Time PCR Detection System (Bio-Rad) and iTaq Universal SYBR Green Supermix (Bio-Rad), according to the manufacturer's instructions. The PCR protocol for telomere repeat amplification has been previously described[107]. For data analysis, samples were normalized to input DNA; RNase H1–treated DNA was used to confirm immunoprecipitation specificity, and siControl RNase H1–untreated samples were set to "1."

### Gene expression analysis by quantitative RT-PCR
Cells were washed with 1xPBS, and total RNA was prepared using Tri-fast (Euroclone) following manufacturer's instructions. Reverse transcription with random primers was performed by using QuantiTect® Reverse Transcription Kit (Qiagen) following manufacturer's instructions. Quantitative PCR (RT-qPCR) was performed using gene specific primers (Supplementary data 8) and the iTaq™ Universal SYBR® Green Supermix (BioRad) in a CFX Connect Real-Time PCR Detection System (BioRad). Specificity of PCR products was routinely checked by melting curve analysis and agarose gel electrophoresis. The PCR protocol for the amplification of TERRA cDNA has been previously described[3].

### RNA sequencing (RNA-Seq)
Total RNA was prepared using the Norgen Biotech RNA extraction Kit following manufacturer's instructions. A TruSeq stranded mRNA library (Illumina) was sequenced on a NovaSeq 6000 RNA Seq platform. 2 × 150 bp paired end sequencing with 18 Gb throughput was performed by Macrogen Europe B.V.

### Purification of recombinant proteins
*E. coli* Rosetta 2 (DE3) cells (Novagen) were transformed with expression vectors for human SFPQ and DAXX (pNIC-CTHF His$_6$-myc SFPQ, pGTvL1-SGC-GST-DAXX-HA). Cells were grown to an OD$_{600}$ of 0.6–0.8 in media containing appropriate antibiotics (chloramphenicol, 34 μg/ml for both plasmids; ampicillin, 100 μg/ml for pNIC-CTHF His$_6$-myc SFPQ; kanamycin, 50 μg/ml for pGTvL1-SGC GST-DAXX-HA) and subsequently transferred to 18 °C. Protein expression was induced with 0.1 mM isopropyl-β-D-1-thiogalactopyranoside (IPTG; Sigma). Sixteen hours post-induction, cells were harvested by centrifugation at 3000 g for 15 min. To purify recombinant human SFPQ, cell pellets were resuspended in Lysis Buffer I (50 mM Tris-HCl, pH 7.5; 300 mM NaCl; 5% glycerol; 5 mM imidazole; 1 mM TCEP) supplemented with DNase I (0.1 mg/ml; Sigma), PMSF (1 mM; Sigma), benzamidine (15 μg/ml; Sigma), leupeptin (4 μg/ml; Sigma), and aprotinin (2 μg/ml; Sigma). Cells were disrupted by sonication, and crude extracts were centrifuged for 1 h at 30,000 g. The soluble fraction was incubated with 1 ml Ni–NTA resin (Qiagen) for 1 h at 4 °C. Resin was washed with 20 volumes of lysis buffer, and bound protein was eluted with lysis buffer containing 300 mM imidazole. Appropriate fractions were pooled and concentrated using Amicon Ultra-15 centrifugal filter units (Merck). The protein was further purified on a Superdex 200 10/300 GL column (Cytiva) equilibrated in SEC Buffer (20 mM Tris-HCl, pH 8.0; 300 mM

NaCl; 0.5 mM TCEP; 10% glycerol). Purified protein was flash-frozen in liquid nitrogen and stored at −80 °C. To purify recombinant human DAXX protein, cell pellets were resuspended in Lysis Buffer II (50 mM NaH$_2$PO$_4$; 300 mM NaCl; 10 mM imidazole; 1 mM TCEP) supplemented with DNase I (0.1 mg/ml), PMSF (1 mM), and complete EDTA-free protease inhibitor tablets (Roche). Cells were disrupted by sonication, and crude extracts were centrifuged at 30,000 g for 1 h. The soluble fraction was incubated with 1 ml Glutathione Sepharose 4B resin (Cytiva) for 1.5 h under rotation at 4 °C. Beads were sequentially washed with lysis buffer II supplemented with decreasing concentrations of KCl (1.0, 0.5, 0.25, and 0 M). Bound protein was eluted with lysis buffer II containing 25 mM glutathione. Appropriate fractions were pooled and concentrated using Amicon Ultra-15 centrifugal filter units (Merck). For further purification, recombinant DAXX was applied to a Superdex 200 10/300 GL column (Cytiva) equilibrated in SEC Buffer. Purified protein was flash-frozen in liquid nitrogen and stored at −80 °C.

### Protein pull-down assays
To validate SFPQ–DAXX interaction, pull-down experiments were carried out using recombinant His$_6$-myc-SFPQ and GST-DAXX-HA. 10 μg of purified His$_6$-myc-tagged SFPQ were added to 10 μl c-myc magnetic beads (Thermo Scientific) in SEC buffer (20 mM Tris-HCl, pH 8.0; 300 mM NaCl; 0.5 mM TCEP; 10% glycerol) at 4 °C under rotation for 2 h. Beads were washed three times with SEC Buffer to eliminate excess SFPQ protein. Recombinant GST-DAXX-HA (5 μg) was incubated with immobilized recombinant SFPQ at 4 °C under rotation for 2 h in SEC buffer. Beads were then washed four times with SEC Buffer. Alternatively, 10 μg recombinant GST-DAXX-HA was immobilized to Glutathione Sepharose 4B beads (10 μl) under rotation for 1 h at 4 °C in SEC buffer. Excess recombinant DAXX protein was removed by three washes in SEC buffer. 5 μg of recombinant His$_6$-myc-tagged SFPQ was incubated with immobilized recombinant GST-DAXX-HA at 4 °C under rotation for 2 h in SEC buffer. Beads were washed four times with SEC Buffer. Beads were resuspended in 2x Sample buffer (2% SDS, 10% glycerol, 0.004% bromophenol blue, 0.125 M Tris-Cl, pH 6.8, 5% DTT) and boiled for 10 min at 95 °C. Samples were analyzed by western blot using an anti-myc (SFPQ) and anti-HA(DAXX) antibodies (Supplementary data 10).

### Preparation of probes for EMSA
RNA and DNA oligonucleotides were chemically synthesized and purified by reverse-phase high pressure liquid chromatography and PAGE (Biomers.net). Selected oligonucleotides were 5′ labeled with fluorescent 6-Carboxyfluorescein (6-FAM). Used DNA and RNA oligos are listed in Supplementary data 1. Oligonucleotides were resuspended in TE buffer (10 mM Tris–HCl pH 7.5, 1 mM EDTA pH 8.0) at a final concentration of 100 μM and stored at −20 °C. For fork-DNA probes (D1L:D3L) the fluorescent strand and the complementary strand were annealed at a 1:3 molar ratio in 20 mM Tris pH 8.0 and 100 mM NaCl. 100 μl reactions including 10 μM of each strand were heated to 95 °C for 5 min and gradually cooled to 15 °C at a rate of 1 °C min$^{-1}$ using a PCR thermal cycler (Eppendorf). For *D-loop* (D4:D9:D11) and *R-loop* (R4:D9:D11) the molar ratio used were 1:1.25:2.5 in 6 mM Tris-HCl pH 7.5, 7 mM MgCl$_2$, 50 mM NaCl and 1 mM DTT. The annealing method used for D-loop and R-loop was heating at 99 °C for 5 min followed by incubations at 67 °C for 1 h, at 37 °C for 30 min and at 25 °C 3–4 h or overnight[108]. Probes were run on native 10% acrylamide (29:1), 1xTBE gels at room temperature in 1x TBE running buffer. Nucleic acid substrates were eluted from gel slices by dialysis in 1xTE buffer.

### Electrophoretic mobility shift assays (EMSA)
Nucleic acid substrates (10 nM) were incubated with increasing amounts of recombinant His$_6$-myc-SFPQ (0, 200, 400, 800, 1600 nM). Binding was performed in EMSA buffer (20 mM Tris-HCl, pH 7.5, 100 mM KCl, 0.5 mM MgCl$_2$, 0.5 mM EDTA, 0.2 mM DTT, 1% glycerol)

at room temperature for 30 minutes. RNase inhibitor (RNaseOUT, Thermo Fisher) was added to reactions containing RNA oligonucleotides at a final concentration of 0.16 units/μl. When indicated, competitor nucleic acids were added. Reactions were run on a non-denaturing 5% polyacrylamide gel at room temperature in 1xTBE buffer for 40 min. Fluorescent-labeled substrates were detected using a fluorescent scanner (ImageQuant, GE Healthcare). Quantification of protein-bound nucleic acid was performed using ImageJ software.

## Sister chromatid exchange assays

Experimental cells were grown in the presence of 5′-bromo-2′-deoxyuridine (BrdU, Sigma, 10 μM) for two rounds of DNA replication. Colcemid was added during the final period of BrdU treatment (0.2 μg/ml, 4 h). Single cells were recovered by trypsinization and incubated with hypotonic solution (46.5 mM KCl, 8.5 mM Na Citrate) at 37 °C for 30 min. Cells in suspension were fixed by slowly adding MeOH/Acetic Acid (ratio 3:1) under soft mixing. Cells were centrifuged (1000 g, 5 min, 4 °C) and fixation procedure was repeated twice. Finally, cells were dropped on glass slides to obtain metaphase spreads. Differential staining of sister chromatids was performed, as previously described[109]. Briefly, metaphase spreads were stained with Hoechst 33,258 (10 μg/ml in 1xPBS, 20 min), washed in Sorensen buffer (0.1 M Na$_2$HPO$_4$, 0.1 M NaH$_2$PO$_4$, pH 6.8) and exposed to UV treatment (UV transilluminator, BioRad) for 30 min to degrade preferentially DNA with incorporated BrdU, thus allowing a differential staining of sister chromatids after Giemsa staining. Metaphase spreads were incubated at 50 °C for 1 h in 1x SSC solution and subsequently dyed in Sorensen buffer supplemented with 10% Giemsa stain solution (Sigma, 0.4%, w/v). Slides were washed in 1xSSC and mounted in ProLong (Invitrogen). Chromosomal aberrations, such as condensation and cohesion defects, recombination and breaks were quantified by visual inspection.

## Immunofluorescence on native metaphase chromosome spreads

For chromosome spreads, asynchronous cells were treated with 0.2 μg/ml Colcemid for 4 h. Cells were collected by mitotic shake-off, as previously described[63]. Mitotic cells were resuspended in 65 mM KCl buffer and allowed to swell for 10 min at room temperature. Subsequently, cells were spun onto glass slides using a Cytopro cytospin-centrifuge (Wescor) at 250 g for 5 min. Slides were incubated in KCM buffer (10 mM KCl, 20 mM NaCl, 10 mM Tris-HCl, pH 8, 0.5 mM EDTA, 0.1% Triton X-100) for 10 min. Primary antibodies (Supplementary data 10) were diluted in 1% BSA/KCM buffer. Slides were incubated with primary antibodies in a wet chamber at room temperature for 2 h. Slides were washed three times with KCM buffer for 5 min each, and incubated with secondary antibodies (Supplementary data 10) in 1% BSA/KCM buffer in a wet chamber at room temperature for 1 h. Cells were subsequently washed three times with KCM buffer, 5 min each, fixed with 4% paraformaldehyde (PFA, in 1xPBS) at room temperature for 15 min. Slides were mounted in ProLong (Invitrogen). All buffers used were supplemented with NaF (Sigma, 1 mM) and sodium ortho-vanadate (Na$_2$VO$_4$, Sigma, 1 mM) to prevent dephosphorylation of proteins. Images were captured using a classic immunofluorescence microscope (Leica DM4000B) and were analyzed by visual inspection.

## RNA-fluorescence in situ hybridization (RNA-FISH)

Cells were permeabilized at room temperature by incubation with Cytobuffer (100 mM NaCl, 300 mM Sucrose, 3 mM MgCl$_2$, 10 mM Pipes pH 6.8) for 30 s, followed by treatment with Cytobuffer supplemented with 0.5% Triton-X for 30 s and a final wash in Cytobuffer for 30 s. Subsequently, cells were fixed for 10 min at room temperature in ice-cold 4% paraformaldehyde (in 1xPBS). Fixed cells were dehydrated by two washes in ice-cold 70% ethanol for 2 min, one wash in ice-cold 90% ethanol for 1 min, and one wash in ice-cold 100% ethanol

for 1 min. Slides were allowed to air-dry and were incubated overnight with an AlexaFluor555 labeled human C$_0$t-1 probe in RNA FISH Hybridization Buffer (2x SSC, 50% formamide) in a humid chamber (2x SSC, 50% formamide) at 37 °C. Subsequently, slides were washed once in 2x SSC for 3 min at room temperature and three times in 2x SSC for 5 min at 37 °C. Slides were transferred to 4x SSC buffer containing DAPI (Sigma, 1 μg/ml). Slides were mounted in ProLong (Invitrogen). Images were captured using an immunofluorescence microscope (Leica DM4000B). For spot signals intensity of interphase nuclei, TFL-TELO software was used. For high-resolution microscopy, a ZEISS ELYRA 7 Structured Illumination Super-Resolution Microscope was used.

## Combined immuno-RNA FISH on interphase cells

Cells were permeabilized with Cytobuffer (100 mM NaCl, 300 mM Sucrose, 3 mM MgCl$_2$, 10 mM Pipes, pH 6.8) and Cytobuffer supplemented with Triton X-100, as described for RNA-FISH and subsequently subjected to the immunofluorescence protocol. Primary and secondary antibodies are listed in Supplementary data 10. After washing, antibody–antigen conjugates were fixed in 4% paraformaldehyde (in 1xPBS) at room temperature for 2 min, washed in 2x SSC and dehydrated in ethanol series (ice-cold 70%, 90%, 100%; 5 min each) followed by RNA-FISH. A ZEISS ELYRA 7 Structured Illumination Super-Resolution Microscope was used to detect co-localization events.

## Combined immuno-RNA FISH on native metaphase chromosomes

Immuno-RNA FISH on native chromosome spreads was carried out with modification to Johnson et al.[110]. Briefly, asynchronous cells were treated with 0.2 μg/ml colcemid for 4 h, collected by trypsinization, resuspended in hypotonic solution (75 mM KCl), and incubated at room temperature (RT) for 20 min. Cells were spun onto glass slides pre-treated with poly-L-lysine (50 μg/ml; Sigma). Cells were permeabilized in KCM buffer (10 mM Tris-HCl, pH 8.0; 120 mM KCl; 20 mM NaCl; 0.5 mM EDTA; 0.1% Triton X-100) for 5 min at RT, then fixed in 4% paraformaldehyde (PFA in 1×PBS) for 10 min. Slides were blocked in KCM buffer supplemented with 2% BSA for 1 h at RT, followed by overnight incubation at 4 °C with primary antibodies (Supplementary data 10) diluted in blocking buffer. The next day, slides were washed three times in KCM buffer and incubated with secondary antibodies (Supplementary data 10) diluted in KCM buffer with 2% BSA for 1 h at RT. Slides were washed three times in KCM buffer, fixed again in 4% PFA (1xPBS) for 10 min, and permeabilized in CSK buffer (100 mM NaCl; 300 mM sucrose; 3 mM MgCl$_2$; 10 mM PIPES, pH 6.8) supplemented with 0.5% Triton X-100 on ice for 10 min. Slides were dehydrated in an ethanol series (70%, 90%, 100%; 2 min each, on ice), air-dried for 10 min at RT, and incubated with RNA-FISH probes (0.1 μM) in hybridization mix (50% formamide, 2× SSC) overnight at 37 °C in a humid chamber. Slides were washed three times in Washing Buffer I (50% formamide, 2× SSC, 0.05% Tween-20) for 5 min at 37 °C, followed by three washes in Washing Buffer II (2× SSC, 0.05% Tween-20) for 5 min at 37 °C. Chromosomes were counterstained with DAPI (1 μg/ml; Sigma) in Washing Buffer II. Excess DAPI was removed with Washing Buffer III (4× SSC, 0.05% Tween-20), and slides were mounted with ProLong (Invitrogen). Images were captured using a classic immunofluorescence microscope (Leica DM4000B) and were analyzed by visual inspection.

## RNA:DNA hybrid dot-blot analysis

$1 \times 10^7$ cells were scraped into 1×PBS, centrifuged for 5 min at 4000 g, 4 °C, and lysed in Lysis Buffer (1% SDS, 20 mM Tris-HCl pH 7.5, 40 mM EDTA pH 8.0, 100 mM NaCl). Subsequently, TE buffer (100 mM Tris-HCl pH 8.0, 10 mM EDTA pH 8.0) was added at a 1:1 ratio. Proteinase K (Invitrogen) was added to a final concentration of 150 μg/ml, and samples were incubated overnight at 37 °C with shaking. Genomic DNA

was purified by phenol−chloroform extraction, ethanol precipitation, and resuspended in TE buffer. DNA was fragmented by sonication using a BioRuptor (Diagenode) to obtain fragments of 150−300 bp. For each sample, 4 μg of genomic DNA was either treated with recombinant RNase H1 (New England Biolabs; 3 units per 1 μg DNA, 37 °C overnight) or left untreated. Samples were diluted in 2x SSC, spotted onto nylon membranes (PerkinElmer), and UV-crosslinked (UVP/Analytik Jena CX-2000; two rounds at 1,200,000 μJ/cm²). Membranes were blocked in Blocking Buffer (5% BSA in 1xPBS) and subjected to standard Western blotting using the S9.6 antibody to detect RNA:DNA hybrids. Total genomic DNA was visualized with SybrGold (Invitrogen).

## Microscopy

Cells subjected to immunofluorescence, RNA-FISH, DNA-FISH, or immunofluorescence combined with RNA-FISH were analyzed using a Leica DM4000B microscope equipped with a Leica DFC420C digital camera. Images were captured using Leica Application Suite (LAS) imaging software. For live-cell imaging, confocal analysis was performed using a Nikon Eclipse C1si confocal microscope system. Images were processed by using ImageJ 1.46r (NIH, Bethesda, USA). In all experiments, single focal planes were used for analysis. A ZEISS ELYRA 7 Structured Illumination Super-Resolution Microscope with ZEN Black software was used for super-resolution microscopy. Colocalization analysis was performed in FIJI (ImageJ) using the JaCoP plugin. For each field, at least three DAPI-defined nuclear ROIs were analyzed. Manders' M2 (fraction of signals overlapping) was computed with Costes' automatic thresholding on a per-nucleus basis and averaged per image to yield one M2 value per field. $n = 10$ fields per condition were used for statistics. Where indicated, Costes' randomization (≥100 permutations) and Van Steensel's cross-correlation analysis were applied to assess non-random overlap and spatial correlation.

## Live-cell imaging analysis

Cell cycle progression of H2B-GFP expressing cells was followed for 14−16 h using Live imaging microscope. Pictures of three separate optical fields were acquired every 10 min. Defects in mitotic progression were quantified by visual inspection of three independent experiments.

## Cell cycle analysis

Cell cycle distribution was assessed by propidium iodide (PI) staining followed by flow cytometry. Briefly, $1 \times 10^6$ cells were harvested, washed with 1xPBS, fixed with cold 70% ethanol, and incubated at 4 °C for at least overnight. Fixed cells were centrifuged at 250 $g$ for 5 min to remove residual ethanol, washed twice with 1xPBS, and rehydrated in 1xPBS at 4 °C for 1 h. Cells were subsequently resuspended in PI staining solution (10 μg/ml PI in 1xPBS; Sigma-Aldrich) supplemented with RNase A (100 μg/ml; Thermo Fisher Scientific) and incubated at 4 °C overnight. Flow cytometry assays were performed using an Attune NxT® flow cytometer (Thermo Fisher Scientific) with acoustic focusing technology, equipped with the standard optical bench configuration, with one blue laser (488 nm). PI fluorescence was collected at BL3 (695/40) PMT. Hierarchical gates and an acquisition flow rate of 100 μl/min were set to avoid coincident events. After recording at least 20,000 events per run, data were saved as FCS files and analyzed using FCS Express V7 De Novo Software with the "Multicycle" function for cell cycle analysis.

## Bioinformatics analysis−ChIP-Seq

ChIP-seq analysis was performed using the ChIP-AP pipeline (ref. 111), performing quality control (FastQC), read trimming and filtering (Bbduk and Trimmomatic), genome mapping (BWA-MEM), peak calling (Genrich, HOMER, MACS2, SICER2), peak merging and annotation (HOMER). Using R/Bioconductor (ref. 112) environment, peaks that were called by three out of four peak calling methods were considered

as present. Peaks were considered differentially expressed when having $p$-value < 0.05 for at least three methods. In particular, for HOMER, MACS2, and SICER2, the Fold Change must be greater than 1 or for Genrich the enrichment score must be positive (upregulated peaks). Overlaps between peaks were calculated with findOverlaps function (IRanges package, ref. 113) with a minimum overlap of 100 bp. Distribution of genomic region annotation was visualized with piecharts generated with the PieDonut function (webr package). Boxplots of the mean of Log₂Fold Change were performed with ggboxplot function (ggpubr package), p-value was calculated with geom_pwc (ggpubr package) function with standard Wilcox test, $p$-adjusted calculation was performed using "fdr" flag. The number of peaks belonging to each genomic region was calculated using the stat_n_text function (EnvStats package). Venn of overlapping peaks was performed using the makeVennDiagram function (ChIPpeakAnno package) using a minimum overlap of 100 bp. For area under curve (AUC) analysis, peak calling was followed by quantification of enrichment signals using the area under the curve (AUC) for each peak. The AUC was calculated as the integral of the normalized ChIP-seq signal (per-base bigWig coverage values) across the full width of each peak. The log₂-transformed average enrichment signal (AUC + 1) was used for downstream comparisons. Boxplots were generated to visualize signal distribution across experimental conditions.

## Bioinformatics analysis−RNA-Seq

RNA sequencing was performed in triplicate for individual conditions. Quality of raw sequence files was checked via FastQC[114]. Transcript quantification was conducted with STAR (version v.2.7.9a; ref. 115) using Ensembl GRCh38 GTF file and genome version (accessed on September 2022). The generated gene counts were consequently analyzed using the R package DESeq2[116]. The normalized count matrix was obtained from variance stabilizing transformation (VST) method as implemented in DESeq2 package. In order to explore high-dimensional data property, among the available algorithms, Principal Component Analysis (PCA) coupled with a dimensionality reduction algorithm was used to scale data and visualized with ggplot2[117]. In order to select only the statistically significant changing genes between comparisons of interest, a differential gene expression analysis was performed. The differentially expressed genes (DEGs) were selected with a $p$-adjusted cut off of 0.05 and a log₂ Fold Change (LFC) value greater than 1.5 (up-regulated DEGs) or lower than −1.5 (down-regulated DEGs). $P$-value was adjusted for multiple testing using the Benjamini−Hochberg (BH) correction with a false discovery rate (FDR) ≤ 0.05. Gene distribution was visualized with a volcano plot using the EnhancedVolcano function (EnhancedVolcano package)[118]. Genes with a Log₂Fold Change greater than 10 or lower than −10 were respectively considered to have a LFC equal to 10 or −10. Genes with a p-adjusted equal to "NA" were considered to have a $p$-adjusted equal to 1. Genes with a LFC equal to "NA" were considered to have an LFC equal to 0. Genes with a p-adjusted inferior to $10^{-10}$ were considered to have a $p$-adjusted equal to $10^{-10}$. DEGs heatmap was performed with the Heatmap function (ComplexHeatmap package)[119], clustering was performed using Euclidean distance and "average" as clustering method. To be able to identify features biological identities and the pathways they belong to, a functional annotation analysis was performed for all the comparisons and for feature list of interest with gprofiler2 package[120]. Different databases were used to annotate the DEGs: Gene Ontology (Molecular Functions−MF, Biological Processes−BP, Cellular Component−CC), Kyoto Encyclopedia of Genes and Genomes (KEGG), Reactome, WikiPathways (WP), Transfac (TF), miRTarBase (MIRNA), Human Protein Atlas (HPA), CORUM (CORUM protein complexes), Human Phenotype Ontology (HP), RNA central. Functional annotation results were visualized with ggpubr package (ref. 121) via Balloon plots. The Gene Ontology network visualization was obtained with the cnetplot function (enrichplot package)[122].

## Analysis of patient data

To analyze patient overall survival, The Cancer Genome Atlas (TCGA) Pancancer Sarcoma dataset (accessed October 2024, $n = 253$)[123] was retrieved from cBioPortal (ref. 124), jointly with patient metadata. The analysis was performed in R/Bioconductor environment. Patients were categorized into high-expression and low-expression groups using the surv_cutpoint (survminer package)[125]. The $p$-value was computed using the Survdiff function from the survival package, and the hazard ratio was determined using the coxph function from the same package. These results were visualized through Kaplan-Meier plots with the ggsurvplot function. Censoring was applied for cases starting at 60 months (equivalent to 5 years) onward, and patients who passed away after that point were treated as though they were still alive before that time. Survival plots were created using the plot_surv_area function (contsurvplot)[126]. The signature score displayed in the survival plot was calculated by taking the trimmed mean at 0.2 for each gene across all samples. TCGA Pancancer data from all cancer datasets (accessed July 2024, $n = 10967$; ref. 123) was retrieved and analyzed as described above.

## Statistics and reproducibility

Data was analyzed by Prism 8 (GraphPad). Results in bar blots are shown as mean ± standard deviation. Unless differentially specified, differences between experimental groups were tested by student's $t$ test. Results were considered significant at $p < 0.05$, and not significant (n.s.) for $p > 0.05$. Precise $p$-values are reported in figures; $p$-values below 0.0001 are indicated as $p < 0.0001$. Number of $N$ (biological replicates), $n$ (analyzed nuclei), as well as analyzed features can be found in figures and figure legends. For native metaphase analyses, "$n$" refers to number of analyzed metaphase spreads and "$n\#$" to the total number of chromosomes. The original data values used to generate the quantification graphs are available in the Source Data file associated with this manuscript. No statistical method was used to pre-determine sample size; no data were excluded from the analyses; experiments were not randomized; investigators were not blinded to allocation during experiments and outcome assessment. Western blots, immunofluorescence images and EMSA data shown in figures are representative images obtained from individual experiments. Immunoprecipitation and protein pull-down experiments were carried out in duplicate. Tests used for bioinformatics analysis are described in the sections "Bioinformatics analysis – ChIP-Seq" and Bioinformatics analysis—RNA-seq". For patient survival, Kaplan-Meier survival curves were generated and analyzed by Log-Rank test.

## Reporting summary

Further information on research design is available in the Nature Portfolio Reporting Summary linked to this article.

## Data availability

The sequencing data for ChIP-Seq and RNA-Seq have been deposited in NCBI's Gene Expression Omnibus with the GEO Series accession number GSE281893 [https://www.ncbi.nlm.nih.gov/geo/query/acc.cgi?acc= GSE281893] and GSE281892 [https://www.ncbi.nlm.nih.gov/geo/query/acc.cgi?acc= GSE281892], respectively. This study analyzed existing, publicly available data from the TCGA Research Network. The mass spectrometry proteomics data have been deposited to the ProteomeXchange Consortium via the PRIDE partner repository with the dataset identifier PXD060026 [http://proteomecentral.proteomexchange.org/cgi/GetDataset?ID=PXD060026]. Requests for further information and resources should be directed to the corresponding authors. Source data are provided with this paper.

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

## Acknowledgements

The laboratory of S.S. is funded by the Fondazione AIRC per la Ricerca sul Cancro Investigator grant IG 2019 Id.23074; the European Union - Next Generation EU, Mission 4, Component 2, CUP B93D21010860004; by the European Union - Next Generation EU, Mission 4, Component 1, Progetti di Ricerca di Rilevante Interesse Nazionale (PRIN) 2022, Prot. 2022KWLYA, CUP J53D23012780006; by the European Union - Next-GenerationEU. Progetti di Ricerca di Rilevante Interesse Nazionale (PRIN) 2022 PNRR Prot. P2022RSP2C. R.B. is supported by the European Union - Next Generation EU, Mission 4 Component 1, Progetti di Ricerca di Rilevante Interesse Nazionale (PRIN) 2022, Prot. 2022KWLYA, CUP G53D2300558006. S.O. is supported by Fondazione AIRC per la Ricerca sul Cancro Investigator grant IG20778 and the European Union's Horizon 2020 research and innovation program under the Marie Skłodowska-Curie grant agreement n. 859853. S.P. and G.C. acknowledge the International Center for Genetic Engineering and Biotechnology (ICGEB) for financial support. G.C. is a PhD student of the Molecular Biomedicine program and acknowledges support provided by University of Trieste. E.P. was supported by an AIRC fellowship 18026. The funders had no role in study design, data collection and analysis, decision to publish, or preparation of the manuscript. We are grateful to P. Calsou and S. Britton for providing U-2 OS cells containing an inducible system for RNaseH1 expression and M. Pagani and G. Della Chiara for support on the preparation of material for ChIP-Seq. We thank G. Del Sal for providing pLV-eGFP. The results presented here are in part based upon data generated by the TCGA Research Network: https://www.cancer.gov/tcga.

## Author contributions

Conceptualization, S.S and R.B. Methodology, A.F., M.G., L.M.R.N, G.C, A.F., A.G., P.V.B., A.Z, E.P., C.B., S.P., R.B. and S.S. Software, G.C., A.P. and S.P. Formal analysis, A.F., M.G., A.F., A.G., P.V.B., A.Z., L.M.R.N., G.C. and E.P. Investigation, A.F., M.G., A.F., A.P., A.G., P.V.B., A.Z., L.M.R.N., G.C. and E.P. Resources, S.P., S.O., R.B. and S.S. Data curation, S.P., L.M.R.N and S.S. Writing - original draft, R.B and S.S Visualization, A.F., A.G., P.V.B., A.Z., M.G., L.M.R.N. and G.C. Supervision, S.P., S.O., R.B., S.S. Project administration, S.S. Funding acquisition, S.S.

## Competing interests

The authors declare no competing interests.
