## [Transparent Peer Review file · Nature Communications]

SFPQ Directs Histone H3.3 Deposition to R-Loops in DNA Repeats to Protect Genome Stability

Corresponding Author: Professor Stefan Schoeftner

Version 0:

Reviewer comments:

Reviewer #1

(Remarks to the Author)

Ferrando et al., describe a role for SFPQ in H3.3 deposition at repetitive DNA repeats in human cells. The authors examine the impact of SFPQ disruption and reveal effects on R-loop binding and homeostasis through a connection to the H3.3 chaperone DAXX. By conducting ChIP-Seq experiments the authors present data showing that the distribution of DAXX and H3.3 at repeats are altered by depleting SFPQ. Furthermore, data is presented showing that these patterns may be anti-correlated with excessive R-loop formation. Ultimately, these alterations lead to mitotic defects, altered genome integrity and elicits innate immune signatures – which parallels circumstances seen in sarcoma. The authors have presented an extraordinary amount of data, comprehensively showing several effects that occur following SFPQ depletion in distinct cell lines. This is commendable. My critique of this paper is more that it represents a compendium of data, revealing little with respect to the mechanistic role of SFPQ. For instance, insights of how SFPQ contributes to DAXX and H3.3 deposition are lacking – largely based on colP of delta proline mutants. Biochemical/in vitro evidence is presented that SFPQ binds to R-loops in the context of D-loops. Yet, information regarding the in vivo context where this activity occurs is not clearly demonstrated. That these defects can be attributed to SFPQs RNA binding/proline rich domains is not particularly novel. Thus, the study provides much data, but the advance is limited.

Some specific issues that could be addressed include

1. Figure 1C and 1D. The colocalization data are not convincing and the depletion of RNaseH1 to induce R-loops is not ideal. Alternative approaches to induce R-loops should be employed and more controls included.
2. Figures 3-5. Greater attention/information relating to the context of the DAXX/SFPQ interaction and function would be very important to include in this study. Do these interactions occur in a particular cell cycle stage or are they transcription-coupled, or replication-coupled. The IF for ATR in Figure 5E would suggest these effects may be linked to replication stress – where are those foci accumulating? It would be positive to elaborate on this. Also, the effect of SFPQ knockdown on ATR accumulation is striking. Can other rep stress markers be assessed by western (not ChIP) and loading clearly shown? Related to this are the increased R-loops generated co-transcriptionally or as an intermediate of replication? What about the relationship of this interaction/function in relation to DAXXs known functional interactions with SETDB1-KAP1 in H3.3 deposition at repeats and transposable elements? Is this SFPQ-DAXX complex distinct from that?
3. How does the depletion of affect DAXX and ATRX protein levels in the H1299 cells? Westerns in supplementary figures show and text describes effects but the basis for this is unexplained. This is a novel finding that could be explained.
4. Figure 6 – given the evident chromosomal instability of SFPQ cells, are these viable and if so, how severe are cell cycle and proliferation defects across multiple cell lines and genetic backgrounds? A G2/M arrest is predicted in U2OS cells but what about other cell lines?

Other issues

Figure 6C – what are these panels showing? Is this a broken metaphase? Not clear.

Figure 7B – extremely difficult to read and evaluate due to size of text

Figure 7D – STING is reported to be suppressed in U2OS cells. How/Why does SFPQ loss lead to its reactivation? Figure 7E – the change in IRF3 expression appears minimal in D – but the IF looks profound. Why is this?

For the ChIP-PCR, are the authors certain that the amplification of telomere repeats is robust? Also, the subtle data – are these solely for chromosome 16? If so, please indicate on the figure?

Supplementary Figure 3 – Per Cimprich and Chedin labs, S9.6 IF is not reliable. Should be validated with RnaseIII/T1/H treatments or removed.

Supplementary Figure 6 – Knockdown of SFPQ does not seem to strongly reduce its levels as detected by ChIP. Why?

Reviewer #2

(Remarks to the Author)

In this manuscript the authors claim that depletion of SFPQ protein results in RNA::DNA hybrid accumulation at repetitive sequences that results in DNA damage and replicative stress. They also claim that SFPQ binds in vitro to R-loops, to chromatin containing R-loops and recruits the histone H3.3 specific chaperon DAXX. Moreover they show a clear interaction of SFPQ with the histone chaperone DAXX and how SFPQ depletion results in nuclear delocalization of DAXX from repeat elements, reduction of histone H3.3 incorporation, replication stress mediated genome instability and presumably, the formation of cytoplasmatic DNA species, which activate innate immunity pathways via the cGAS/STING pathway. While the results reported are potentially interesting and some are sound for others I'm not convinced. In the present form the manuscript requires a major revision before acceptance.

Major points:

1. Concerning Section "SFPQ targets R-loops in repetitive elements"

Lines 87-88: "Immuno-dot blot experiments using genomic DNA from SFPQ depleted U-2 OS, H1299, MCF-7 and MCF-10A cells revealed an significant increase in steady state DNA:RNA hybrid levels"

I note the tests are indicating significance though fold changes are only modest (1 to ~1.4-1.6 for H1299 and MCF-10A; 1 to ~2.5 for U-2 OS and MCF-7 cells)

Lines 92-93: "intensity of C0t-1 RNA FISH foci increased significantly after depletion of SFPQ from U-2 OS cells, indicative for elevated RNA levels at repeat elements (Fig. 1A)"

Is this reproducible in the other cell lines as well?

Lines 95-97: "Importantly, C0t-1 RNA FISH signals in SFPQ depleted U-2 OS cells frequently co-localized with p-ATR-(Thr1989) and FANCD2, indicative for replication stress at repetitive elements"

To my eyes colocalization of C0t-1 signal with p-ATR and FANCD2 in SFPQ depleted cells is low; i.e. 2-3 events/nucleus when total number of dots for each of them is much larger. Moreover, this was derived from conventional IF microscopy and not even confocal or super-resolution microscopy so that it was co-localization at low resolution. What is the ratio of co-localized p-ATR/FANCD2 vs total spot number? I think this ratio will more accurately reflect the results shown.

Lines 99-102. Fig 1C. Where does the increased SFPQ signal observed in Fig 1C upon transient RNAseH1 depletion come from?

Again, due to the high number of SFPQ spots colocalization of events with CREST is very modest. What is the ratio of co-localized/total CREST spots?

Same for TRF2 and SFPQ (Fig 1D).

Lines 111-112 "We conclude that SFPQ is recruited to R-loops and protects different categories of repetitive elements from unscheduled R-loops."

I disagree with this conclusion at this point. The presence of SFPQ at R-loops must be shown directly either microscopically showing co-localization of SFPQ and R-loops and/or with higher resolution techniques like ChIPseq/DRIPseq or similar. The experiments shown only indirectly suggest that. A relevant overlap of R-loops caused by SFPQ depletion with SFPQ ChIPseq peaks would substantiate this conclusion.

2. The whole section "SFPQ has in vitro binding specificity for R-loops" must be improved before I can accept the authors' conclusions.

Frequently, where authors claim to see a complex I rather see some probe precipitation. This usually takes place at the bottom of wells and while in some pictures the wells are visible in others are not. In a few gels a faint indication of a complex is visible (e.g. Fig 2B middle and right panels, 2C all panels, Suppl. Fig 2c left panel) but these results must be clearly improved to become convincing.

Additional controls showing that the different probes indeed correspond to the expected structures are missing. And if relative affinities are analyzed, competition experiments with unlabeled probes must be performed.

3. Concerning section "SFPQ is a binding partner of the Death Domain Associated Protein (DAXX)"

Lines 155-157: Authors stated "Immunoprecipitation of endogenously expressed proteins demonstrated that a significant fraction of total SFPQ and DAXX are present in a protein complex in U-2 OS cells (Fig. 3D)."

In fact, I think that the fraction that ColPs is minor as demonstrated by its detection only after a long exposure using apparently good antibodies (see controls).

Lines 174-178: In the paragraph is written "Transient depletion of SFPQ from H1299 cells resulted a significant reduction of steady state ATRX protein levels without altering ATRX mRNA expression as detected by immunofluorescence, western blotting and quantitative RT-PCR analysis (Supplementary figure 3H, J, K). Thus, reduced ATRX expression in SFPQ loss of function H1299 cells recapitulates ATRX and DAXX expression status in U-2 OS cells." I don't understand what does it mean that SFPQ loss of function recapitulates ATRX and DAXX expression status in U-2 OS cells. I couldn't find ATRX expression in SFPQ loss of function U-2 OS cells (Suppl. Fig 3G does not include it). On the other hand, I can appreciate an increase of DAXX expression in these conditions that it is not even mentioned in the text and deserve some explanation.

4. Concerning section "SFPQ controls DAXX dependent histone H3.3 deposition"

DAXX peaks are reduced in SFPQ depleted cells but H3.3 peaks clearly increase. How can this be explained?

Lines 222-224: "We conclude that SFPQ binding of R-loops recruits DAXX dependent H3.3 chaperon deposition to suppress the formation of unprogrammed R-loops at repeat elements." But being DAXX a chaperone for H3.3 how it comes that there are more H3.3 peaks in SFPQ depleted cells? Since DAXX is depleted from repetitive elements and H3.3 is depleted as well, where are those extra peaks? Is H3.3 deposited independent of DAXX elsewhere?

5. Concerning section "The Proline-rich domain and RRM motifs of SFPQ are central for R-loop suppression"

In overexpression experiments it is mandatory to show levels of overexpression. To my eye they look quite high to reasonably address colocalization experiments and certainly much higher than the endogenous levels shown in Fig 1C,D.

6. Concerning section "SFPQ dependent R-loop resolution suppresses innate immunity pathways in human sarcoma"

Lines 314-318 "Independent quantitative RT-PCR analysis and western blotting confirmed the upregulation of classic target genes linked to the cGAS/STING pathway, interferon response, inflammation and the NFkB pathway in U-2 OS but also in H1299 SFPQ knock-down cells (Fig.7C, D Supplementary figure 7D, E). This indicates general relevance for SFPQ in suppressing innate immunity pathways."

While this is a potential explanation, I wonder whether any excess of R-loops triggers interferon response and immunity pathways. For instance, Histone H1 depletion has been shown to trigger interferon response in T47D cells via heterochromatic repeats (Izquierdo-Bouldstridge et al, Nucleic Acids Res. 2017 45:11622-11642). This point should be discussed.

Lines 339-341: "The expression of the direct transcriptional target of SFPQ, IL8 (reference 55), CXCL10 and IRF3 was found to be R-loop independent (Supplementary figure 7H)".

In fact upon RNase H overexpression they even increase, remarkably CXCL10 and IL8. Could the authors explain these results? Isn't it curious that being direct targets of SFPQ they are R-loop independent?

7. Concerning several sections and Figures.

In DRIPseq and DRIPqPCR experiments RNaseH sensitivity is usually shown as an indication of true R-loops presence. The authors, however, do not show it in the Figs and they "subtracted" these results assuming they are noise indications. I'm unsure about that calculation because RNaseH sensitivity varies with sequence. Rather, I'd like to see RNaseH sensitivity control in all of the experiments reported.

On the other hand, the authors pretend to make a clear connection between SFPQ absence and R-loop formation but I see nowhere how the DRIPseq regions overlap with the SFPQ ChIPseq peaks. This information must be provided together with some representative profiles of regions showing the overlaps in a Figure.

8. If authors want to keep in the abstract that loss of SFPQ results in the formation of cytoplasmic DNA species they must show the presence of this DNA directly, not only indirectly via cGAS/STING activation. Micronuclei are not equivalent to cytoplasmic DNA.

Minor points:

- It would be convenient to include an introductory paragraph devoted to SFPQ protein to put in context the reader.
- Suppl. Fig 4A It would be clearer if instead of labelling "siS" siSFPQ is used. Also, What is "n" in Fig 4E?
- Line 298 Labeling mistake. It is Fig. 6G, and Supplementary figure 6G
- I can't find Suppl. Table 7
- Line 664 "Gene expression analysis by RT-PCR". It is not indicated whether the RTs were performed using oligodT or random oligomers. If oligodT was used all non-polyadenylated transcripts derived from repetitive elements would have been missed...

Reviewer #3

(Remarks to the Author)

This is a strong followup paper to previous publication from the same group (Nat Comm., Petti et al, 2019) showing SFPQ and NONO as suppressors of RNA-DNA hybrids regulating telomere stability. Overall, this is an extensive and careful study, using a combination of approaches to convincingly demonstrate that SFPQ interacts with single strand RNA associated with R-loops at repetitive DNA elements, and recruits DAXX to suppress a DNA damage response. They also show that SFPQ restricts the innate immune response associate with R-loop formation at these repeats. There are a few concerns that would help to improve the manuscript.

Specific Comments:

1. Fig 1.B. The total number of colocalization events for pATR and COT-1 seem very low, suggesting that this is not frequent. The authors should comment and consider why this is the case.
2. Fig. 2C and D. The SFPQ signal in IF is so overwhelming that it is difficult to conclude that there is selective colocalization with CREST or TRF2. The authors should consider using Proximity Ligation Assay to demonstrate colocalization by imaging.
3. Line 137. "...SFPQ contains a remarkable in vitro binding specificity towards three-stranded loop structures, in particular..with protruding RNA termini. " Figure 2 seems to show highest affinity for single stranded RNA, and the fact that an R-loop with extension of single stranded RNA has similar affinity to single stranded RNA is not remarkable, but consistent with predominant single strand RNA binding activity of SFPQ.
4. Line 176-178. The meaning of this sentence is unclear. Please rewrite!
"Thus, reduced expression in SFPQ loss of function H1299 cells recapitulates ATRX and DAXX expression status in U-2 OS cells."
5. Supplemental Figure 4, ChIP-seq data is not compelling and seems to be mostly random. Is the overlap of H3.3 and SFPQ more than expected by random chance?
6. Fig 5c. The authors should indicate the Ig band in the IP Western as it is very close to the myc-P domain. Also, the SFPQ nFS mutant seems to be unstable relative to the wild type. The authors should comment on this and quantify the Western blot for IP efficiencies.
7. The interaction of SFPQ and DAXX should be examined with native proteins in a both ALT and non-ALT cell lines. Does the interaction of DAXX with ATRX prevent SFPQ-DAXX interaction?
8. RNAseq data showing that siRNA to SFPQ induces innate immune response. The authors should provide at least one additional control that the siRNA itself is not generating the innate immune response through TLR signaling. At least one additional siControl similar to the siSFPQ in base composition should be tested to rule out this possibility.

Minor

Typo line 380. SPFQ

Version 1:

Reviewer comments:

Reviewer #2

(Remarks to the Author)

In the revised version of the manuscript entitled "SFPQ Directs Histone H3.3 Deposition to R-Loops in DNA Repeats to Protect Genome Stability" by Ferrando et al,

the authors have introduced most of the experiments I suggested as well as properly replied to all the questions I raised. I sincerely think that in the present version the manuscript has improved notably and is now acceptable for publication in Nature Communications.

However, prior to publication I strongly recommend to make some minor amendments that do neither affect the results nor the conclusions of the manuscript. These comments/corrections/mistakes are listed below:

Comments:

- Following my comments authors have included in this revised version WBs showing the extent of many SFPQ depletions. However, now they use RNaseH1 depletion in several experiments and it would also be nice to show the extent of this depletion at least in a WB.

- Gene descriptions on Suppl. Table 7 are cut and incomplete

- Lines 631 and following "... To evaluate target specificity of the S9.6 antibody, U-2 OS cells feed on cover slips were treated with RNaseT1 (NEB, 1U/μl; NEBuffer 2), RNaseT3 (NEB, 1U/μl NEBuffer or RNaseH1 (NEB, 1U/μl NEBuffer 2) at 37°C for 30 minutes. Samples were incubated at 65 °C for 20 minutes to inactivate the enzymes, washed in 1x PBS and processed following the immunofluorescence protocol."

It is unclear to me whether this treatment was prior or posterior to cell fixation.

- Line 694 "Immunoprecipitated chromatin was de-crosslinked by adding 200mM NaCl and leaving samples at 65°C overnight"

Is that right? To my knowledge this is usually performed using NaHCO₃ and not simply NaCl.

- Line 910 "Sonication was performed with BioRuptor (Diagenode) to obtain DNA fragments in a range between 150-300 bp."

Those are harsh conditions to check for R-loops in genomic DNA and in particular for detection of large R-loops (>300 bp) that are sensitive to sonication. Were these conditions really used? If so, authors are probably underestimating R-loop

abundance.

Mistakes:

I found many writing mistakes (largely in STAR methods section) to be corrected. Some examples are:

- Fig 7G nFS control is labeled in red $p=0.0078$. Is this a mistake?

-The sentence

“Briefly, cells were lysed in RIPA buffer (50mM Tris pH 7.4, 250mM NaCl, 1% Triton-X, 1% DOC, 0.1% SDS), sonicated (Fisherbrand Model 120 Sonic 563 Dismembrator, Fisher Scientific, 20 seconds, 20% amplitude).”
looks like inconsistent to me.

- Authors frequently misuse the verb “to result in”.

- Line 492: CO₂ for CO₂

- Line 579: siring for syringe

- Line 630 “recombinant RNaseH1 (NEB) for 1 hour at 37°C, according to manufacturer’s instruction,..”

- Line 652 “temperature. Cells were blocked with 5%BSA (in 1x BS) at 37°C for 20 minutes”

- Line 657 “citric acid, 82 mM Na₂HPO₄. Final pH 7.0), deionize formamide 70% final concentration”

- Line 782 “The pull-down experiments validate of His6-myc SFPQ with GST-DAXX-HA were performed in SEC Buffer” This sentence looks like incomplete

- Line 816 “RNase inhibitors RNaseOUT (0.16 units/μl, Thermo Fisher) was added to reactions containing RNA oligonucleotides.”

Either RNase inhibitor ... was added or RNase inhibitors ... were added

- Line 826 “Subsequently, cells were fixed by dropping MeOH/Acetic Acid (ratio 3:1) under constant agitation three times, altered by centrifugation at 1000g, 5 minutes, 4°C, and dropped on slides to obtain metaphase spreads.”
What do authors mean by “altered by centrifugation”?

- Line 872 “Cells were permeabilized with Cytobuffer..”

- Line 894 “..sucrose, 3mM MgCl₂..”

Reviewer #3

(Remarks to the Author)

The authors have provided substantial additional experimental data and extensively detailed rebuttal to address all of my previous concerns. The revised manuscript provides interesting new findings on the role of SFPQ in control of RNA-DNA hybrids, recruitment of DAXX-H3.3 to repetitive DNA regions, and the activation of the STING pathway. The data provided is sufficiently high technical quality to support the conclusions.

Reviewer #4

(Remarks to the Author)

The authors have made a substantial effort to improve the original version of the manuscript and have responded clearly to most of the reviewers’ previous comments. I support publication in principle; however, several important issues remain that must be addressed before acceptance.

DAXX/SFPQ Interaction Timing

The authors have not clarified at which stage of the cell cycle the DAXX/SFPQ interactions occur.

Effect of SFPQ Depletion on DAXX Expression

The authors did not explain why depletion of SFPQ (siSFPQ) leads to an increase in DAXX protein expression levels. A mechanistic rationale or supporting experimental data should be provided.

Cell Line Specificity of G2/M Arrest

The authors have not indicated whether G2/M arrest following SFPQ depletion is observed in cell lines other than U2OS. Extending this analysis would strengthen the generality of their findings.

Validation of STING Expression Controls

In Reviewer Figure 1, OVCAR4 cells are used as a positive control for STING expression. It would have been preferable to

include siRNA-mediated STING knockdown as an additional control to confirm band specificity. Furthermore, the qPCR results for STING (Figure 7D) should be validated by western blot analysis, especially since the authors have demonstrated that they possess a functional antibody (Reviewer Figure 1).

SFPQ Knockdown and ChIP Signal Interpretation

The authors' response regarding the limited reduction of SFPQ ChIP signals upon SFPQ knockdown (Supplementary Figure 6D) could be strengthened. It would be helpful to clarify whether the total amount of immunoprecipitated chromatin/DNA decreases in SFPQ ChIP after SFPQ knockdown.

Minor Comment – Supplementary Figure 1F

Supplementary Figure 1F appears to show SFPQ staining, yet the figure legend indicates measurement of S9.6 staining intensity. The image and legend should be aligned to accurately reflect the data presented.

Version 2:

Reviewer comments:

Reviewer #4

(Remarks to the Author)

I am satisfied with the author's responses to my comments, and support publication.

POINT BY POINT REPLY TO REVIEWER COMMENTS

Reviewer 1:

Ferrando et al., describe a role for SFPQ in H3.3 deposition at repetitive DNA repeats in human cells. The authors examine the impact of SFPQ disruption and reveal effects on R-loop binding and homeostasis through a connection to the H3.3 chaperone DAXX. By conducting ChIP-Seq experiments the authors present data showing that the distribution of DAXX and H3.3 at repeats are altered by depleting SFPQ. Furthermore, data is presented showing that these patterns may be anti-correlated with excessive R-loop formation. Ultimately, these alterations lead to mitotic defects, altered genome integrity and elicits innate immune signatures – which parallels circumstances seen in sarcoma. The authors have presented an extraordinary amount of data, comprehensively showing several effects that occur following SFPQ depletion in distinct cell lines. This is commendable. My critique of this paper is more that it represents a compendium of data, revealing little with respect to the mechanistic role of SFPQ. For instance, insights of how SFPQ contributes to DAXX and H3.3 deposition are lacking – largely based on colP of delta proline mutants. Biochemical/in vitro evidence is presented that SFPQ binds to R-loops in the context of D-loops. Yet, information regarding the in vivo context where this activity occurs is not clearly demonstrated. That these defects can be attributed to SFPQs RNA binding/proline rich domains is not particularly novel. Thus, the study provides much data, but the advance is limited.

Reply to reviewer: We thank the reviewers for their comments regarding the comprehensiveness of the presented data and their critical insights. With this revised version of the manuscript, we have addressed all reviewer concerns and aim to present a more robust and convincing body of evidence that underscores the novelty and impact of our findings.

In particular, our study uncovers a remarkable binding specificity of SFPQ to R-loop structures, its dynamic recruitment to newly formed R-loops within repetitive regions, and a correlated activity in R-loop suppression. Proteomic analyses identify the H3.3 chaperone DAXX as an interaction partner of SFPQ, recruited to R-loops in an SFPQ-dependent manner.

Furthermore, ChIP-Seq experiments demonstrate that loss of SFPQ leads to DAXX delocalization and impaired deposition of histone H3.3. These findings contribute valuable knowledge to the limited understanding of early nucleic acid-mediated events involved in the recruitment of DAXX and ATRX to repetitive DNA. To our knowledge, this is the first report describing an R-loop binding protein that facilitates the recruitment of a H3.3 histone chaperone, thereby counteracting chromatin destabilization and safeguarding genome integrity at DNA repeats.

We also identify functionally relevant domains of SFPQ and validate our mechanistic model using a cancer-associated patient mutation. Our manuscript provides a functional dissection of the mechanisms underlying SFPQ-mediated R-loop suppression and highlights its critical role in maintaining repeat stability. Notably, SFPQ loss-of-function promotes genomic instability, accumulation of cytoplasmic DNA, and activation of innate immune pathways—establishing a compelling link to tumor microenvironment modulation, with particular relevance to human sarcoma. Clinical data on patient survival further support the potential clinical significance of our proposed mechanism.

Major Points

Reviewer Comment 1:

Some specific issues that could be addressed include

1. Figure 1C and 1D. The colocalization data are not convincing and the depletion of RNaseH1 to induce R-loops is not ideal. Alternative approaches to induce R-loops should be employed and more controls included.

Reply to reviewer Comment 1:

We have repeated immunofluorescence experiments and performed high-resolution microscopy using an ZEISS ELYRA 7 Structured Illumination Super-Resolution microscope and confirmed between SFPQ-CREST and SFPQ-TRF2 colocalization after RNaseH1 knock-down (**NEW FIGURE 1C, D**). Colocalization of marker proteins was recapitulated when ectopic R-loops were induced by hydroxyurea treatment (**NEW SUPPLEMENTARY FIGURE 1F-H**). The related text can be found in **Line 108-111** of the modified version of the manuscript. Related methods were included into the corresponding material and methods section.

Reviewer Comment 2:

2. Figures 3-5. Greater attention/information relating to the context of the DAXX/SFPQ interaction and function would be very important to include in this study. Do these interactions occur in a particular cell cycle stage or are they transcription-coupled, or replication-coupled.

a) The IF for ATR in Figure 5E would suggest these effects may be linked to replication stress – where are those foci accumulating? It would be positive to elaborate on this.

Reply to Reviewer Comment 2a

In the revised manuscript, we employed high-resolution microscopy to validate that ectopic expression of dominant-negative SFPQ variants (ΔP and nFS) induces the formation of phosphorylated ATR (p-ATR) foci. A subset of these p-ATR foci co-localizes with TRF2 a protein that binds telomere repeats. This indicates that used SFPQ variants drive replication stress at telomeres—known hotspots for R-loop formation. These findings are presented in the revised manuscript as **NEW SUPPLEMENTARY FIGURE 5E**, with the corresponding text located at **Line 279-283**.

b) Also, the effect of SFPQ knockdown on ATR accumulation is striking. Can other rep stress markers be assessed by western (not ChIP) and loading clearly shown? Related to this are the increased R-loops generated co-transcriptionally or as an intermediate of replication?

Reply to Reviewer Comment 2b

We have performed western blotting experiments using U-2 OS cells transiently transfected with control and SFPQ specific siRNAs. SFPQ knock down results phosphorylation of Chk1 and RPA32Ser33, reproducing data from a recent study (PMID: 30341290). This data and corresponding references have been inserted into the revised manuscript as **NEW SUPPLEMENTARY FIGURE 1E**. The related text can be found in **Line 105-108**.

c) What about the relationship of this interaction/function in relation to DAXXs known functional interactions with SETDB1-KAP1 in H3.3 deposition at repeats and transposable elements? Is this SFPQ-DAXX complex distinct from that?

Reply to Reviewer Comment 2b

We fully agree with the reviewer that elucidating the functional relationship between SFPQ and both the ATRX-dependent and ATRX-independent roles of DAXX—particularly in the context of H3.3 deposition and the formation of the SMARCAD1-KAP1-SETDB1 complex at various classes of repeat elements and retrotransposons—represents a highly relevant and compelling research avenue. This topic has been extensively explored in prior studies, especially those utilizing mouse embryonic stem cell models (PMID: 21666679, 20651253, 29084956, 30902974, 40450002, 20075919, 11959841).

However, we believe that exploring the comprehensive integration these factors exceeds the scope of the current study and would be better addressed in a dedicated follow-up project. Such a study would involve extensive genome-wide profiling of chromatin-associated factors via ChIP-seq in SFPQ loss-of-function models as well as downstream functional experiments. In response to the reviewer's suggestion, we have incorporated a discussion on the potential role of SFPQ in recruiting DAXX to R-loops and its possible interaction with the SETDB1-KAP1-SMARCAD1 complex. This addition is now included in the revised manuscript at **Line 433-441**.

Reviewer Comment 3:

How does the depletion of affect DAXX and ATRX protein levels in the H1299 cells? Westerns in supplementary figures show and text describes effects but the basis for this is unexplained. This is a novel finding that could be explained.

Reply to Reviewer Comment 3:

In the revised version of the manuscript we show that treatment with the proteasome inhibitor MG132 protects ATRX protein in SFPQ knock-down H1299 cells. This demonstrates that degradation via the proteasome contributes to loss of ATRX protein under SFPQ knock-down conditions. This data has been inserted into the revised manuscript as **NEW SUPPLEMENTARY FIGURE 3N**. The related text can be found in **Line 194-198**.

Reviewer Comment 4.

Figure 6 – given the evident chromosomal instability of SFPQ cells, are these viable and if so, how severe are cell cycle and proliferation defects across multiple cell lines and genetic backgrounds? A G2/M arrest is predicted in U2OS cells but what about other cell lines?

We did not include this data into the revised version of the manuscript as several studies report extensively on cell cycle profiles in different SFPQ loss of function cells, including U-2 OS cells. Results differ between different cell models, presumably due to differences in genetic background (PMID: 20813759, PMID: 38103553, PMID: 30341290, PMID: 37424803, PMID: 20421735, PMID: 19439179). From these studies emerge that SFPQ loss of function U-2 OS cells show defects in S-Phase progression, increased population of cells in G2/M and subG1. In the revised version of the manuscript we refer to relevant studies and present new western blotting data that show that SFPQ depletion leads to Chk1 and RPA32Ser33 phosphorylation (**NEW SUPPLEMENTARY FIGURE 1E**).

We have inserted new text in **Line 105-108** of the revised version of the manuscript: *In line with the activation of ATR, we observed elevated phosphorylation of Chk1 and RPA32 in SFPQ depleted cells, as demonstrated by western blot analysis (Supplementary Figure 1E). The induction of replication stress is consistent with previously reported disruptions in S-phase progression following SFPQ depletion*⁵³⁻⁵⁵. (PMID: 38103553; PMID: 30341290; PMID: 19439179).

Minor Points

Reviewer Comment:

Figure 6C – what are these panels showing? Is this a broken metaphase? Not clear.

Reply to Reviewer Comment

The revised version of manuscripts contains a modified figure legend for Fig. 6C, indicating that arrowheads indicate events of ATR phosphorylation in vicinity to centromeres.

Reviewer Comment:

Figure 7B – extremely difficult to read and evaluate due to size of text

Reply to Reviewer Comment

We have increased text size in Figure 7B

Reviewer Comment:

Figure 7D – STING is reported to be suppressed in U2OS cells. How/Why doe SFPQ loss lead to its reactivation?

Reply to Reviewer Comment

Recent studies have reported low or undetectable levels of STING protein in U-2 OS cells (PMID: 32284536; PMID: 28179534; PMID: 33529438) In alternative, STING levels may eventually be elevated by an artefact derived form transfection with siRNAs, as suggested recently (PMID: 36343682) This unspecific effect may have an impact on our data. We repeated STING western blotting experiments with untreated and control siRNA transfected U-2 OS cells. We also loaded extracts from OVCAR-4 cells that express high STING protein levels. Again, we were able to detect low STING expression in untreated cells. The STING band obtained with U-2 OS cells shows up at same molecular weights as in OVCAR4 cells that were used as positive control (**REVIEWER FIGURE 1**). Transfection with control siRNAs did not increase STING expression (**REVIEWER FIGURE 1**).

We conclude that low STING levels in U-2 OS cells are sufficient to activate the cGAS/STING pathway leading to the activation of genes related innate immunity (Supplementary Figure 7). We want to underlined that this effect was also reproduced in unrelated H1299 cells (Supplementary Figure 7) In the revised version of the manuscript we refer to reported low/absent STING expression in the results section in **Line 375-379**: “U-2 OS cells have been reported express STING protein at low or undetectable levels.”

To date, we have not investigated the potential mechanisms by which SFPQ may influence STING expression. We consider this question to be beyond the scope of the current study and more appropriately addressed in a future investigation. Nonetheless, it is intriguing to speculate that the previously reported role of SFPQ as a transcriptional repressor (PMID: 24507715) may be relevant in this context.

Reviewer Figure 1. Western blotting using whole cell extracts from OVCAR-4 and U-2 OS cells that were untreated or transfected with the indicated concentration of control siRNAs. UNT, untreated; μg, loaded amount of extract; short and low exposure images of STING expression are shown. An HSP90 specific antibody was used as loading control.

Reviewer Comment:

Figure 7E – the change in IRF3 expression appears minimal in D – but the IF looks profound. Why is this?

Reply to Reviewer Comment

Figure 7D shows that IRF3 mRNA modestly increase after SFPQ knock-down, as detected by quantitative real-time PCR. In Figure 7E we show immunofluorescence data on phosphorylated IRF3

protein, indicative for the activation of the protein. A long exposure time was used to enable a visualization of low levels of pIRF3 in control cells.

Reviewer Comment:

For the ChIP-PCR, are the authors certain that the amplification of telomere repeats is robust? Also, the subtel data- are these solely for chromosome 16? If so, plz indicate on the figure?

Reply to Reviewer Comment

In this study, we utilized PCR primers specific to the subtelomeric region of the short arm of chromosome 16 (SubTel Chr. 16p). These primers were originally published by the Lieberman group, a leading authority in subtelomere research (PMID: 23010778). In the revised manuscript, we explicitly state that our subtelomeric analysis is restricted to chromosome 16p (**Line 118; Fig. 1F; Fig. 6A, B; Supp. Fig. 1I; Supp. Fig. 6A-D**). The reference to the Lieberman study is provided in Supplementary Table 11.

Reviewer Comment:

Supplementary Figure 3 – Per Cimprich and Chedin labs, S9.6 IF is not reliable. Should be validated with RNaseIII/T1/H treatments or removed.

Reply to Reviewer Comment

We performed control immunofluorescence experiments by treating fixed cells with RNaseIII, RNaseT1 and RNaseH. We found that only treatment with RNaseH eliminates S9.6 antibody staining. This supports that that elevated S9.6 signal intensities in SFPQ knock-down experiments are predominantly due to increased RNA:DNA hybrid levels. We provide this information in **NEW SUPPLEMENTARY FIGURE 3E**. We state in the revised result section that antibody specificity was validated (**Line 181-182**). I want to point out that we have performed an additional control experiment; in particular, **Supplementary figure 3D** shows that ectopic expression of RNaseH1 reduced S9.6 signal intensity in SFPQ knock-down experiments. We also show in new DRIP-PCR quantifications that pretreatment of DNA with recombinant RNaseH1 drastically reduces immunoprecipitation using S9.6 antibodies (**MODIFIED FIGURE 1F and MODIFIED SUPPLEMENTARY FIGURE 1H**). Finally, ectopic expression of RNaseH1 suppresses innate immunity pathways triggered by R-loop accumulation on SFPQ knock-down cells (**Figure 7I, Supplementary Figure 7I**). We are thus confident that all phenotypes and observation reported in our manuscript are R-loop dependent.

Reviewer Comment:

Supplementary Figure 6 – Knockdown of SFPQ does not seem to strongly reduce its levels as detected by ChIP. Why?

Reply to Reviewer Comment

These differences may arise from technical factors, specificity of the antibody for denatured SFPQ (western) or crosslinked SFPQ (ChIP). Eventually intrinsic properties of the SFPQ protein may impact results. SFPQ is highly abundant in the nucleus, and we hypothesize that while overall SFPQ protein levels are markedly reduced following knockdown, its depletion at R-loop hotspots may be less pronounced—potentially due to increased protein stability when bound to chromatin.

New experiments in the revised version provide further evidence for binding of SFPQ to R-loops. In particular we show that R-loop inducing conditions drive the recruitment of SFPQ to all repeat regions tested (**NEW FIGURE 1E**).

Reviewer 2

Major points:

In this manuscript the authors claim that depletion of SFPQ protein results in RNA::DNA hybrid accumulation at repetitive sequences that results in DNA damage and replicative stress. They also claim that SFPQ binds in vitro to R-loops, to chromatin containing R-loops and recruits the histone H3.3 specific chaperon DAXX. Moreover, they show a clear interaction of SFPQ with the histone chaperone DAXX and how SFPQ depletion results in nuclear delocalization of DAXX from repeat elements, reduction of histone H3.3 incorporation, replication stress mediated genome instability and presumably, the formation of cytoplasmic DNA species, which activate innate immunity pathways via the cGAS/STING pathway.

While the results reported are potentially interesting and some are sound for others I'm not convinced. In the present form the manuscript requires a major revision before acceptance.

Reply to reviewer: We thank the reviewer for the positive comments on the quality of scientific findings and also the critical comments. With this revised version of the manuscript, we have aimed to address all reviewer concerns and present now a more robust and convincing body of evidence that underscores the novelty and impact of our findings.

Reviewer Comment 1:

1. Concerning Section "SFPQ targets R-loops in repetitive elements"

Lines 87-88: "Immuno-dot blot experiments using genomic DNA from SFPQ depleted U-2 OS, H1299, MCF-7 and MCF-10A cells revealed a significant increase in steady state DNA:RNA hybrid levels" I note the tests are indicating significance though fold changes are only modest (1 to ~1.4-1.6 for H1299 and MCF-10A; 1 to ~2.5 for U-2 OS and MCF-7 cells).

Reply to Reviewer Comment 1

RNA dot blotting is not an extremely precise quantitative method, but we were able to detect increased global RNA:DNA hybrid levels across all SFPQ depleted cell lines. We have performed independent measurements of R-loops by immunofluorescence (Supplementary figure 3D; **NEW SUPPLEMENTARY FIGURE 1F**) and DRIP-PCR (Figure 1F, **NEW SUPPLEMENTARY FIGURE 1I**) in U-2 OS cells that confirm data from dot-blot experiments. Several control experiments validate anti-body specificity: Ectopic expression of RNaseH1 eliminated pan-nuclear S9.6 staining in siSFPQ cells (Supplementary figure 3C); additional control experiments using fixed cells to perform pre-treatments with RNaseIII, RNaseT1 and RNaseH were performed. Only treatment with RNaseH efficiently reduces S9.6 staining, suggesting that elevated S9.6 signal intensities in SFPQ knock-down experiments are predominantly due to increased RNA:DNA hybrid levels (**NEW SUPPLEMENTARY FIGURE 3E**). Therefore, we are confident that SFPQ is a central repressor of R-loop formation.

Reviewer Comment 2:

Lines 92-93: "intensity of C₀t-1 RNA FISH foci increased significantly after depletion of SFPQ from U-2 OS cells, indicative for elevated RNA levels at repeat elements (Fig. 1A).

Is this reproducible in the other cell lines as well?

Reply to Reviewer Comment 2

We have repeated immunofluorescence experiments and performed high-resolution microscopy using ZEISS ELYRA 7 Structured Illumination Super-Resolution Microscope microscope and confirmed elevated C₀t-1 RNA-FISH signals in SFPQ depleted U-2 OS cells (**NEW FIGURE 1A, B**). These findings were reproduced in SFPQ or RNaseH1 knock-down H1299 cells (**NEW SUPPLEMENTARY FIGURE 1C**). The main text was adjusted accordingly (**Line 97-98**), related methods were included into the corresponding material and methods section (Supplementary information).

Reviewer Comment 3:

Lines 95-97: “Importantly, C₀t-1 RNA FISH signals in SFPQ depleted U-2 OS cells frequently co-localized with p-ATR-(Thr1989) and FANCD2, indicative for replication stress at repetitive elements”. To my eyes colocalization of C₀t-1 signal with p-ATR and FANCD2 in SFPQ depleted cells is low; i.e. 2-3 events/nucleus when total number of dots for each of them is much larger. Moreover, this was derived from conventional IF microscopy and not even confocal or super-resolution microscopy so that it was co-localization at low resolution. What is the ratio of co-localized p-ATR/FANCD2 vs total spot number? I think this ratio will more accurately reflect the results shown. Make high resolution.

Reply to Reviewer Comment 3

We repeated the immunofluorescence experiments and performed high-resolution imaging using a ZEISS ELYRA 7 Structured Illumination Super-Resolution microscope, confirming co-localization of C₀t-1 RNA with phosphorylated ATR (P-ATR) and FANCD in SFPQ-depleted cells (**NEW FIGURE 1B, NEW SUPPLEMENTARY FIGURE 1D**). Due to the limited sensitivity of RNA-FISH, we acknowledge that our data may not provide a fully accurate or quantitative assessment of all C₀t-1 transcription events at related regions in control or RNaseH1/SFPQ loss-of-function cells. It is plausible that C₀t-1 repeat regions are transcribed at low levels in control cells, but remain undetectable under our experimental conditions (**see Line 102-105 of revised manuscript**)

Through the quantification presented in **NEW FIGURE 1B** and **NEW SUPPLEMENTARY FIGURE 1D**, we aim to demonstrate that transcription of C₀t-1 RNAs can induce replication stress *in cis*. In particular when their expression is elevated and may be paralleled by R-loop formation. The observation that depletion of either SFPQ or RNaseH1—both known to act as R-loop suppressors—leads to an increased frequency of P-ATR and C₀t-1 RNA co-localization events supports the hypothesis that ectopic R-loops are responsible for ATR activation *in cis*. We therefore, we consider the current mode of quantification shown in Figure 1 and Supplementary Figure 1D as appropriate and informative.

To better discuss our observations we improved the text related to C₀t-1 RNA-FISH and replication stress (**Line 102-105**): “C₀t-1 RNA-FISH lacks sufficient sensitivity to monitor transcriptional activities across all repeat sequences. Nevertheless, our findings indicate that loss of RNaseH1 or SFPQ leads to aberrant RNA metabolism at loci containing repeat elements that promotes replication stress”.

Reviewer Comment 4:

Lines 99-102. Fig 1C. Where does the increased SFPQ signal observed in Fig 1C upon transient RNaseH1 depletion come from? Again, due to the high number of SFPQ spots colocalization of events with CREST is very modest. What is the ratio of co-localized/total CREST spots? Same for TRF2 and SFPQ (Fig 1D).

Reply to Reviewer Comment 4

We repeated experiment shown in Fig 1C, D or the original manuscript, performing high-resolution microscopy. We confirmed SFPQ/CREST and SFPQ/TRF2 colocalization in RNaseH1 depleted U-2 OS cells and calculated the ratio of co-localized/total CREST spots (**NEW FIGURE 1C, D**). New ChIP experiments (**NEW FIGURE 1E**) further demonstrate that ectopic increase of R-loop levels by transient depletion of RNaseH1 (see Supplementary figure 1I for increased R-loops) lead to increased occupation of the very same repeat elements by SFPQ. Together with experiments from EMSA experiments our data demonstrates the recruitment of SFPQ to chromatin containing unprogrammed R-loops.

Reviewer Comment 5:

Lines 111-112 “We conclude that SFPQ is recruited to R-loops and protects different categories of repetitive elements from unscheduled R-loops.”

I disagree with this conclusion at this point. The presence of SFPQ at R-loops must be shown directly either microscopically showing co-localization of SFPQ and R-loops and/or with higher resolution techniques like Chip-seq/DRIP-seq or similar. The experiments shown only indirectly suggest

that. A relevant overlap of R-loops caused by SFPQ depletion with SFPQ ChIP-seq peaks would substantiate this conclusion.

Reply to Reviewer Comment 5

Detection of R-loops by immunofluorescence using S9.6 antibodies (following methanol-acetic acid fixation) and conventional immunostainings on protein epitopes (using paraformaldehyde fixation) rely on fixation protocols that are inherently incompatible. Despite performing a series of test experiments, we were unable to obtain reliable immunofluorescence results when attempting to combine S9.6 staining with standard immunofluorescence procedures.

To address the issue raised by the reviewer, we conducted a new set of DRIP-PCR and ChIP-PCR experiments. By knocking down RNaseH1 in U-2 OS cells, we experimentally increased R-loop formation at repeat regions and observed SFPQ recruitment across our full panel of DNA repeat categories (**NEW FIGURE 1E**). Furthermore, the revised manuscript clearly demonstrates that depletion of either RNaseH1 or SFPQ leads to R-loop accumulation in these regions (**NEW FIGURE 1F, MODIFIED SUPPLEMENTARY FIGURE 1I**).

In addition, an expanded series of EMSA experiments using competitor probes (provides further evidence of direct SFPQ binding to R-loop structures **NEW SUPPLEMENTARY FIGURE 2D**). Given the robust results obtained from DRIP-PCR and complementary assays, we opted not to pursue DRIP-seq. Optimization, execution, and analysis of DRIP-seq would likely have exceeded the available revision timeframe and required substantial financial resources.

Reviewer Comment 6:

2. The whole section “SFPQ has in vitro binding specificity for R-loops” must be improved before I can accept the authors’ conclusions. Frequently, where authors claim to see a complex I rather see some probe precipitation. This usually takes place at the bottom of wells and while in some pictures the wells are visible in others are not. In a few gels a faint indication of a complex is visible (e.g. Fig 2B middle and right panels, 2C all panels, Suppl. Fig 2c left panel) but these results must be clearly improved to become convincing. Additional controls showing that the different probes indeed correspond to the expected structures are missing. And if relative affinities are analyzed, competition experiments with unlabeled probes must be performed.

Reply to Reviewer Comment 6

We have revised Figure 2 and Supplementary Figure 2 in the updated manuscript to include representative EMSA gels showing the full gel images, including loading slots. SFPQ has been reported to undergo phase separation, and previous studies have shown that it can form high-molecular-weight complexes in the presence of specific nucleic acid structures, which exhibit reduced migration or fail to enter EMSA gels efficiently (PMID: 25765647). In our experiments, the formation of such high-weight complexes is selectively triggered by particular nucleic acid structures. Notably, double-stranded DNA and perfectly matched RNA:DNA hybrids do not induce a mobility shift in the presence of SFPQ.

As requested, we performed competition assays demonstrating SFPQ’s preferential binding to R-loop structures. These data are presented in **NEW SUPPLEMENTARY FIGURE 2D**. Importantly, only a 10-fold molar excess of single-stranded RNA is capable of competing with R-loop structures for SFPQ binding, underscoring the binding specificity of SFPQ for R-loops. dsDNA is unable to compete with R-loops for SFPQ. The corresponding text is included at **Line 148-153** in the revised manuscript.

In addition, we provide information on the identity of the used substrates in **REVIEWER FIGURE 2**

Reviewer Comment 7:

3. Concerning section "SFPQ is a binding partner of the Death Domain Associated Protein (DAXX)" Lines 155-157: Authors stated "Immunoprecipitation of endogenously expressed proteins demonstrated that a significant fraction of total SFPQ and DAXX are present in a protein complex in U-2 OS cells (Fig. 3D).".

In fact, I think that the fraction that ColPs is minor as demonstrated by its detection only after a long exposure using apparently good antibodies (see controls).

Reply to Reviewer Comment 6

We agree with the reviewer and state now in the revised version of the manuscript:

"Immunoprecipitation of endogenously expressed proteins demonstrated that a relevant fraction of total SFPQ and DAXX are present in a protein complex in U-2 OS cells" (Line 172)

Reviewer Comment 8:

Lines 174-178: In the paragraph is written "Transient depletion of SFPQ from H1299 cells resulted a significant reduction of steady state ATRX protein levels without altering ATRX mRNA expression as detected by immunofluorescence, western blotting and quantitative RT-PCR analysis (Supplementary figure 3H, J, K). Thus, reduced ATRX expression in SFPQ loss of function H1299 cells recapitulates ATRX and DAXX expression status in U-2 OS cells." I don't understand what does it mean that SFPQ loss of function recapitulates ATRX and DAXX expression status in U-2 OS cells. I couldn't find ATRX expression in SFPQ loss of function U-2 OS cells (Suppl. Fig 3G does not include it). On the other hand, I can appreciate an increase of DAXX expression in these conditions that it is not even mentioned in the text and deserve some explanation.

Reply to Reviewer Comment 8

We have performed additional experiments and rephrased the section in the revised version of the manuscript.

We show that treatment with the proteasome inhibitor MG132 "protects" bulk ATRX protein levels in SFPQ knock-down H1299 cells. This demonstrates that degradation via the proteasome contributes to loss of ATRX protein under SFPQ loss of function conditions. This data has been inserted into the revised manuscript as **NEW SUPPLEMENTARY FIGURE 7N**. The related text was rephrased and can be found in **Line 194-198**: *"Treatment of SFPQ knock-down H1299 cells with MG132 rescued ATRX protein levels. This suggests that, in the absence of SFPQ, ATRX is targeted for degradation by the proteasome (Supplementary figure 3N). We conclude that SFPQ controls DAXX and ATRX function in cancer cells by directing DAXX localization and ensuring ATRX protein expression."*

Reply to comment on increased DAXX protein levels upon SFPQ depletion in old Supplementary figure 3H: please note that increased ACTIN levels in siSFPQ samples versus siControl samples indicate unequal loading of protein extracts. This explains different intensities of DAXX bands in the western blot.

Reviewer Comment 9:

Concerning section "SFPQ controls DAXX dependent histone H3.3 deposition"

DAXX peaks are reduced in SFPQ depleted cells but H3.3 peaks clearly increase. How can this be explained?

Lines 222-224: "We conclude that SFPQ binding of R-loops recruits DAXX dependent H3.3 chaperon deposition to suppress the formation of unprogrammed R-loops at repeat elements." But being DAXX a chaperone for H3.3 how it comes that there are more H3.3 peaks in SFPQ depleted cells? Since DAXX is depleted from repetitive elements and H3.3 is depleted as well, where are those extra peaks? Is H3.3 deposited independent of DAXX elsewhere?

Reply to Reviewer Comment 9

We showed by ChIP-seq that H3.3 peak distribution is changing upon DAXX delocalization in SFPQ depleted cells. We found that in SFPQ depleted cells the height of all detected (genome wide) H3.3 peaks is reduced. At the same time the number of these low peaks is significantly increased.

To get more insights into this phenomenon, we analyzed the H3.3 log₂ Fold Change and performed area under the curve (AUC) calculations (across the entire genome).

AUC over a given peak region is the integral of the signal across the width of the peak itself.

Mathematically:

$$AUC = \int_i^L coverage_i$$

where L = number of bases in the peak; $coverage_i$ is the per-base signal (e.g. bigWig value) at position i . This method captures both how tall the peak is, representing the maximum coverage, and how wide it is, explaining how many bases are enriched. A high AUC can either come from a sharp, high summit with many reads piling up at a narrow spot, from moderate enrichment spread over a broad region, or anywhere in between.

Although the number H3.3 peaks in SFPQ knock-down versus those in control cells is substantially increased, the H3.3 AUC in SFPQ knock-down cells is significantly lower than in control conditions (**NEW SUPPLEMENTARY FIGURE 4F, new text Line 232-237**). This support a scenario where H3.3 shifts from high-occupancy sites to more widespread, low-occupancy deposition.

In addition to this we show by western blotting that bulk H3.3 protein levels are not altered by RNAi mediated depletion of SFPQ. (**NEW SUPPLEMENTARY FIGURE 4H**). The related text can be found in **Lines 247-249**. Figure legends and Material and Methods sections were modified accordingly.

Reviewer Comment 10:

Concerning section "The Proline-rich domain and RRM motifs of SFPQ are central for R-loop suppression" In overexpression experiments it is mandatory to show levels of overexpression. To my eye they look quite high to reasonably address colocalization experiments and certainly much higher than the endogenous levels shown in Fig 1C, D.

Reply to Reviewer Comment 10

Stable expression of myc-tagged SFPQ deletion constructs in U-2 OS cells was monitored with an anti-myc antibody in western blotting experiments (Supplementary figure 5D). Performing anti-myc immunofluorescence experiments we found that expression of SFPQ constructs in stably transfected cells is very heterogeneous and sometimes limited to only 10% of the cells (data not shown). We therefore concentrated the quantification of immunofluorescence data (in Fig. 5E-F and Supplementary figure 5F, G) on myc-positive cells. In the revised version of the manuscript we state in the corresponding section of figure legends: "*Myc-positive cells were considered for quantification*"

Reviewer Comment 11:

6. Concerning section "SFPQ dependent R-loop resolution suppresses innate immunity pathways in human sarcoma"

Lines 314-318 "Independent quantitative RT-PCR analysis and western blotting confirmed the upregulation of classic target genes linked to the cGAS/STING pathway, interferon response, inflammation and the NFκB pathway in U-2 OS but also in H1299 SFPQ knock-down cells (Fig.7C, D Supplementary figure 7D, E). This indicates general relevance for SFPQ in suppressing innate immunity pathways."

While this is a potential explanation, I wonder whether any excess of R-loops triggers interferon response and immunity pathways. For instance, Histone H1 depletion has been shown to trigger interferon response in T47D cells via heterochromatic repeats (Izquierdo-Bouldstridge et al, Nucleic Acids Res. 2017 45:11622-11642). This point should be discussed.

Reply to Reviewer Comment 11

We thank the reviewer for this helpful comment. We are convinced that SFPQ mediated accumulation of R-loop drives genome instability and generation of cytoplasmic DNA. In the revised version of the manuscript we show now that SFPQ depletion does not only result in an increased frequency of micronuclei but also in increased abundance of cytoplasmic DNA, as shown by immunofluorescence using dsDNA specific monoclonal 3519 antibodies (**NEW SUPPLEMENTARY FIGURE 7F**). Related text can be found in **Line 350-352**)

To further address the reviewer's suggestion on immunity pathway activation via R-loops or Histone H1 depletion state now in the discussion section (**Lines 463-465**): "*Thus, R-loop mediated genome instability represents a primary source for the production of cytoplasmic DNA and activation of cGAS/STING signaling in SFPQ loss of function cells. However, we cannot exclude a contribution of repeat RNAs that escape R-loop structures at DNA repeats, cytoplasmic R-loops or an alteration in chromatin structure in promoters connected to interferon response genes*^{99,100}"

Reviewer Comment 12:

Lines 339-341: "The expression of the direct transcriptional target of SFPQ, IL8 (reference 55), CXCL10 and IRF3 was found to be R-loop independent (Supplementary figure 7H)".

In fact upon RNase H overexpression they even increase, remarkably CXCL10 and IL8. Could the authors explain these results? Isn't it curious that being direct targets of SFPQ they are R-loop independent?

Reply to Reviewer Comment 11

We thank the reviewer for pointing out this misleading part of the results section. We have removed data on CXCL10 from original Supplementary Figures 7H and J. **MODIFIED FIGURES 7H, I and MODIFIED SUPPLEMENTARY FIGURE 7I** include now IRF-3 into the list of R-loop induced genes, as ectopic expression of RNaseH1 prevents siSFPQ induced activation of IRF-3 when compared to RNaseH1 expressing and control siRNA transfected U-2 OS (p value = n.s.). As reported previously, IL-8 is subjected to transcriptional repression by SFPQ in an R-loop independent manner (PMID: 36457191). Thus IL-8 levels increase upon SFPQ depletion and do not drop when SFPQ knock-down is combined with ectopic expression of Cherry-tagged RNaseH1.

In the revised version of the manuscript we changed the text accordingly (**Lines 367-373**):

"Using U-2 OS cells carrying a doxycycline inducible mCherry-tagged RNaseH1 (reference 66) we found that ectopic expression of RNaseH1 was able to suppress micronuclei formation in SFPQ or DAXX depleted cells but also the activation of 10 out of 11 candidate pro-inflammatory genes (Fig. 7 H, I, Supplementary figure 7I). Given the reported role of SFPQ as transcriptional repressor of IL8 (reference 48) SFPQ depletion increased IL8 expression in an RNaseH1 independent manner (Supplementary figure 7I)".

Reviewer Comment 13:

7. Concerning several sections and Figures.

a) In DRIPseq and DRIPqPCR experiments RNaseH sensitivity is usually shown as an indication of true R-loops presence. The authors, however, do not show it in the Figs and they “subtracted” these results assuming they are noise indications. I’m unsure about that calculation because RNaseH sensitivity varies with sequence. Rather, I’d like to see RNaseH sensitivity control in all of the experiments reported.

Reply to Reviewer Comment 13a:

We have changed the corresponding figures. We including RNaseH1 treated samples in individual quantifications of DRIP-PCR experiments (**MODIFIED FIGURE 1F and SUPPLEMENTARY FIGURE 1I**). RNaseH1 pre-treatment dramatically reduces PCR amplification in DRIP experiments.

b) On the other hand, the authors pretend to make a clear connection between SFPQ absence and R-loop formation but I see nowhere how the DRIPseq regions overlap with the SFPQ ChIPseq peaks. This information must be provided together with some representative profiles of regions showing the overlaps in a Figure.

Reply to Reviewer Comment 13b:

Our study we does not contain DRIP-Seq experiments.

We performed ChIP-seq experiments on SFPQ, DAXX and H3.3. and observed chromatin defects in repeat regions after SFPQ knock-down (Figure 4 and Supplementary figure 4). Defects in chromatin structure were validated at representative regions by ChIP-PCR (Figure 6, Supplementary figure 6). Aberrant accumulation of R-loops in SFPQ loss of function conditions was thoroughly validated by DRIP-PCR in the very same regions (**MODIFIED FIGURE 1F, MODIFIED SUPPLEMENTARY FIGURE 1I**). We were not able to perform new DRIP-Seq experiments for the revision of the manuscript as the DRIP-Seq optimization, performance and data analysis and downstream validation experiments may not have been completed within the allotted revision timeframe and would have also required substantial financial resources. Together with molecular data from EMSA experiments, DNA damage signaling analysis by ChIP, protein-protein interaction studies, a SFPQ deletion analysis, the use of SFPQ containing a patient derived mutation, genome instability analyses and the downstream activation of innate immunity we are convinced that SFPQ acts as central factor in protecting repeat regions from R-loop mediated genome instability.

Reviewer Comment 14:

8. If authors want to keep in the abstract that loss of SFPQ results in the formation of cytoplasmic DNA species they must show the presence of this DNA directly, not only indirectly via cGAS/STING activation. Micronuclei are not equivalent to cytoplasmic DNA.

Reply to Reviewer Comment 14:

In the revised version of the manuscript we show now that loss of SFPQ leads to a substantial increase of cytoplasmic DNA as evidenced by immunofluorescence staining using dsDNA specific 3519 monoclonal antibodies (**NEW SUPPLEMENTARY FIGURE 7F**).

Minor Points

Reviewer Comment:

- It would be convenient to include an introductory paragraph devoted to SFPQ protein to put in context the reader.

Reply to Reviewer Comment:

We have included this information in the revised version of the manuscript. (**Lines 72-77**)

Reviewer Comment:

- Suppl. Fig 4A It would be clearer if instead of labelling "siS" siSFPQ is used. Also, What is "n" in Fig 4E?

Reply to Reviewer Comment:

We have changed the figure and figure legend accordingly. In Fig 4E, "N" refers to the number of independent experiments (biological replicates)

Reviewer Comment:

- Line 298 Labeling mistake. It is Fig. 6G, and Supplementary figure 6G

Reply to Reviewer Comment:

Thanks for pointing that out. We have changed the text accordingly.

Reviewer Comment:

- I can't find Suppl. Table 7

Reply to Reviewer Comment:

Supplementary table 7 shows differentially expressed genes in SFPQ knock-down U-2 OS cells and should have been submitted with the rest of supplementary information.

Reviewer Comment:

- Line 664 "Gene expression analysis by RT-PCR". It is not indicated whether the RTs were performed using oligodT or random oligomers. If oligodT was used all non-polyadenylated transcripts derived from repetitive elements would have been missed...

Reply to Reviewer Comment:

We used random primers for quantitative real-time PCR. We have included the information into the revised version material and methods section "*Gene expression analysis by RT-PCR*".

Reviewer 3:

This is a strong followup paper to previous publication from the same group (Nat Comm., Petti et al, 2019) showing SFPQ and NONO as suppressors of RNA-DNA hybrids regulating telomere stability. Overall, this is an extensive and careful study, using a combination of approaches to convincingly demonstrate that SFPQ interacts with single strand RNA associated with R-loops at repetitive DNA elements, and recruits DAXX to suppress a DNA damage response. They also show that SFPQ restricts the innate immune response associate with R-loop formation at these repeats. There are a few concerns that would help to improve the manuscript.

Reply to reviewer: We thank the reviewer for the positive comments on the relevance and quality of our manuscript. With this revised version of the manuscript, we have aimed to address all reviewer concerns and present now a more robust and convincing body of evidence that underscores the novelty and impact of our findings.

Major Points

Reviewer Comment 1:

Fig 1.B. The total number of colocalization events for pATR and C₀T-1 seem very low, suggesting that this is not frequent. The authors should comment and consider why this is the case.

Reply to Reviewer Comment:

We have repeated immunofluorescence experiments and performed high-resolution microscopy using a ZEISS ELYRA 7 Structured Illumination Super-Resolution microscope and confirmed C₀t1 and p-ATR/FANCD colocalization data (**NEW FIGURE 1B, NEW SUPPLEMENTARY FIGURE 1D**). New data from high-resolution C₀t-1 RNA-FISH confirms increased C₀t-1 RNA-FISH foci in RNaseH1 or SFPQ depleted U-2 OS and H1299 cells (**FIGURE 1A, NEW SUPPLEMENTARY FIGURE 1C**). Our data demonstrate that the presence of aberrantly increased C₀t-1 RNA-FISH foci leads to replication stress at a subset of repeat loci, as shown by C₀t-1 RNA – P-ATR/FANCD2 colocalization. Given that RNA-FISH has limited sensitivity we are not able to get an complete and quantitatively accurate picture on total “C₀t-1 transcription/R-loop events” and frequency of replication stress *in cis* in control and RNaseH1/SFPQ loss of function cells. In fact, we hypothesize that numerous repeat loci may be transcribed (and engage in R-loop formation), but cannot be detected by RNA-FISH in our experiments.

With the quantification in **NEW FIGURE 1B and NEW SUPPLEMENTARY FIGURE 1** we want to make evident that conditions of increased C₀t-1 number and signal intensity in RNaseH1 or SFPQ depletion correlate with replication stress *in cis*. The fact that both, loss of the R-loop suppressors SFPQ or RNaseH1 show increased frequency of pATR and C₀t-RNA colocalization events support the hypothesis that ectopic R-loops are responsible for ATR activation. We therefore think it is adequate to keep the mode of quantification shown in **NEW FIGURE 1B and NEW SUPPLEMENTARY FIGURE 1C**.

In summary, panels of **NEW FIGURE 1B and NEW SUPPLEMENTARY FIGURE 1C** highlight a link between SFPQ and repeat DNA, which is validated by quantitative and site-specific methods such as ChIP (replication stress and DNA damage markers, Fig. 6; Supplementary figure 6) and DRIP (R-loops, Fig 1F; Supplementary figure 1E).

In the revised version of the manuscript improved text related to C₀t-1 RNA-FISH and replication stress (**Lines 102-105**): *“C₀t-1 RNA-FISH lacks sufficient sensitivity to monitor transcriptional activities across all repeat sequences. Nevertheless, our findings indicate that loss of RNaseH1 or SFPQ leads to aberrant RNA metabolism at loci containing repeat elements that promotes replication stress.*

Reviewer Comment 2:

Fig. 2C and D. The SFPQ signal in IF is so overwhelming that it is difficult to conclude that there is selective colocalization with CREST or TRF2. The authors should consider using Proximity Ligation Assay to demonstrate colocalization by imaging.

Reply to Reviewer Comment:

We have repeated immunofluorescence experiments and performed high-resolution microscopy using an ZEISS ELYRA 7 Structured Illumination Super-Resolution microscope and confirmed co-localization between SFPQ-CREST and SFPQ-TRF2 in conditions of elevated R-loop levels driven by RNaseH1 depletion or hydroxyurea treatment (**NEW FIGURE 1C, D; NEW SUPPLEMENTARY FIGURE 1 G, H**). The related text can be found in **Line 108-114** of the modified version of the manuscript. Related methods were included into the corresponding material and methods section.

Reviewer Comment 3:

Line 137. "...SFPQ contains a remarkable in vitro binding specificity towards three-stranded loop structures, in particular..with protruding RNA termini. "Figure 2 seems to show highest affinity for single stranded RNA, and the fact that an R-loop with extension of single stranded RNA has similar affinity to single stranded RNA is not remarkable, but consistent with predominant single strand RNA binding activity of SFPQ.

Reply to Reviewer Comment:

We have revisited the Figure 2 and Supplementary Figure 2 in the revised version of the manuscript. We show now entire gels of representative EMSA experiments including slots slots. SFPQ is reported to engage in phase separation and a previous has demonstrated that SFPQ can form high-weight molecular complexes with o low migration speed in EMSA gels (**PMID: 25765647**). I want to point out that complex formation is a specific feature triggered by selected nucleic acid structures. Double stranded DNA or perfectly matched RNA:DNA hybrids do not mediate an electrophoretic shift with SFPQ. In addition, we have performed competition experiments that demonstrate preferred binding of SFPQ to R-loop structures. This data has been inserted as **NEW SUPPLEMENTARY FIGURE 2D**. In fact, only 10-fold molar excess of single stranded RNA is able to compete with R-loop structures for SFPQ binding. The related text can be found in **Line 148-153** of the revised version of the manuscript. We also provide reviewers with data related to the correct preparation of EMSA probes. (**REVIEWER FIGURE 2**).

Reviewer Figure 2. Quality control of protocols for the annealing of oligonucleotides. Indicated nucleic acids were loaded on native polyacrylamide gels. Green, nucleic acid strands; red, RNA strand; star indicates FAM labels. For more details on procedure see material and methods.

Reviewer Comment 4:

Line 176-178. The meaning of this sentence is unclear. Please rewrite!

"Thus, reduced expression in SFPQ loss of function H1299 cells recapitulates ATRX and DAXX expression status in U-2 OS cells."

Reply to Reviewer Comment:

We have rephrased the section in the revised version of the manuscript (**Line 194-198**)

We show now that treatment of cells with the proteasome inhibitor MG132 protects ATRX protein levels in SFPQ knock-down H1299 cells. This demonstrates that degradation via the proteasome

contributes to loss of ATRX under SFPQ loss of function conditions. This data has been inserted into the revised manuscript as **NEW SUPPLEMENTARY FIGURE 3N**. The related text can be found in **Line 194-198**: *“Treatment of SFPQ knock-down H1299 cells with MG132 rescued ATRX protein levels, anticipating an involvement of the proteasome in degrading ATRX in the absence of SFPQ (Supplementary figure 3N). We conclude that SFPQ controls DAXX and ATRX function in cancer cells by directing DAXX localization and ensuring ATRX protein expression.”*

Reviewer Comment 5:

Supplemental Figure 4, CHIP-seq data is not compelling and seems to be mostly random. Is the overlap of H3.3 and SFPQ more than expected by random chance?

Reply to Reviewer Comment:

Our CHIP seq data demonstrate that loss of SFPQ leads to a delocalization pattern of DAXX across the genome (Fig. 4 and Supplementary Figure 4). At the same time, we found by CHIP-seq that H3.3 peak height and distribution changes in SFPQ depleted cells. We found that in SFPQ depleted cells the overall height of H3.3 peaks is reduced; at the same time the overall number of these low H3.3 peaks increases significantly. This context hampered the clear association of SFPQ peaks (that were lost upon SFPQ depletion) with an associated change in H3.3 peak profile.

To get more insights into this H3.3 distribution in SFPQ depleted cells, we analyzed H3.3 log₂ fold changes and performed area under the curve (AUC) calculations. In this context, the AUC over a given peak region is the integral of the signal across the width of the peak itself. Mathematically:

$$AUC = \int_i^L coverage_i$$

where L = number of bases in the peak; $coverage_i$ is the per-base signal (e.g. bigWig value) at position i . This method captures both how tall the peak is, representing the maximum coverage, and how wide it is, explaining how many bases are enriched. A high AUC can either come from a sharp, high summit with many reads piling up at a narrow spot, from moderate enrichment spread over a broad region, or anywhere in between.

Although the number H3.3 peaks in SFPQ knock-down versus those in control cells is substantially increased, the H3.3 AUC in SFPQ knock-down cells is significantly lower than in control conditions (**NEW SUPPLEMENTARY FIGURE 4F**). This supports a scenario where H3.3 shifts from high-occupancy sites to more widespread, moderate-occupancy deposition in SFPQ depleted cells. Our data anticipate that these effects are derived from the de-localization of the corresponding H3-3 histone chaperon DAXX in SFPQ depleted cells.

In addition to this we show by western blotting that bulk H3.3 protein levels are not altered by RNAi mediated depletion of SFPQ. (**NEW SUPPLEMENTARY FIGURE 4H**).

The related text can be found in **Line 233-237**. Figure legends and Material and Methods sections were modified accordingly.

Our data highlight a central role for SFPQ in regulating DAXX localization and H3.3 deposition. However, we cannot exclude the possibility that SFPQ may also participate in additional pathways involved in the regulation of repetitive DNA elements. Several studies in mouse embryonic stem cells have reported that DAXX interacts with the SMARCAD1–KAP1–SETDB1 complex, which promotes a repressive chromatin environment at endogenous retroelements that is characterized by H3K9me3. Notably, SMARCAD1 activity is critical for the incorporation of H3.3 within these regions. It is therefore intriguing to speculate that SFPQ may be functionally linked to the

SMARCD1-containing complex, potentially influencing H3.3 deposition through an alternative mechanism. This hypothesis has been incorporated into the discussion section (see Line 433-441) and will be addressed in a separate scientific project.

Reviewer Comment 6:

-Fig 5c. The authors should indicate the Ig band in the IP Western as it is very close to the myc-P domain. Also, the SFPQ nFS mutant seems to be unstable relative to the wild type. The authors should comment on this and quantify the Western blot for IP efficiencies.

Reply to Reviewer Comment:

We have included the requested information and quantification of IP efficacy into the revised version of the manuscript: **MODIFIED FIGURE 5C** and related text in the figure legend.

Reviewer Comment 7:

The interaction of SFPQ and DAXX should be examined with native proteins in a both ALT and non-ALT cell lines. Does the interaction of DAXX with ATRX prevent SFPQ-DAXX interaction?

Reply to Reviewer Comment:

We did perform this type of immunofluorescence experiment. During the assembly of the original manuscript assembly, we inadvertently misidentified the cell lines used in the immunoprecipitation experiments shown in Fig. 3B and 3C, referring to them as “U-2 OS” instead of “H1299 cells (that express ATRX, see Supp. Fig. 3I). This error has been corrected in the revised manuscript, which now clearly states that the experiments in Fig. 3B and 3C were conducted using ATRX-proficient H1299 cells (see Line 170-171 and the updated figure legend Fig. 3). ATRX expression in H1299 was validated in Supplementary figure 3I.

Reviewer Figure 3. Quantitative real-time PCR showing mRNA expression levels of indicated genes. U-2 OS cells per transiently transfected with the indicated siRNAs. siControl, control siRNAs; siSFPQ, pool of 4 different SFPQ specific siRNAs; siSFPQ 5'UTR, siRNA targeting 5'UTR of SFPQ; siSFPQ 5'UTR scramble, scrambled SFPQ 5'UTR siRNA; N, number of biological replicates; a student's t-test was used to calculate statistical significance; actin was used as reference gene

Legend:
 □ siControl
 ◻ siSFPQ
 ◻ siSFPQ 5'UTR
 ◻ siSFPQ 5'UTR scramble
 N=3

Reviewer Comment 8:

RNaseq data showing that siRNA to SFPQ induces innate immune response. The authors should provide at least one additional control that the siRNA itself is not generating the innate immune response through TLR signaling. At least one additional siControl similar to the siSFPQ in base composition should be tested to rule out this possibility.

Reply to Reviewer Comment:

We have generated an additional siRNA control (siSFPQ-5'URT scramble) by modifying our SFPQ targeting siRNAs (siSFPQ-5'UTR). Transfection of cells with siSFPQ-5'UTR induced marker genes for innate immunity (CCL5, STING, IFI44), as detected by quantitative real-time PCR. Importantly the siSFPQ-5'UTR scramble siRNA did not induce innate immunity marker genes. We provide this data to

the reviewers (**REVIEWER FIGURE 3**) and state in the revised version of the manuscript: “*Transfection of U-2 OS cells with scrambled SFPQ siRNAs did not induce the expression of marker genes of innate immunity, confirming that observed phenotypes are related to loss of SFPQ and not to ectopic TLR signaling (data not shown)*”. This text can be found in **Line 373-375** of the revised version of the manuscript.

Minor Points

Reviewer Comment

Typo line 380. SPFQ

Reply to Reviewer Comment

We corrected the typo in line 380.

POINT BY POINT REPLY TO REVIEWER COMMENTS

Reviewer #2 (Remarks to the Author):

General Comment:

In the revised version of the manuscript entitled “SFPQ Directs Histone H3.3 Deposition to R-Loops in DNA Repeats to Protect Genome Stability” by Ferrando et al, the authors have introduced most of the experiments I suggested as well as properly replied to all the questions I raised. I sincerely think that in the present version the manuscript has improved notably and is now acceptable for publication in Nature Communications. However, prior to publication I strongly recommend to make some minor amendments that do neither affect the results nor the conclusions of the manuscript. These comments/corrections/mistakes are listed below:

Reply to general comment:

We thank Reviewer 2 for considering the revised version of our manuscript suitable for publication in *Nature Communications*. We greatly appreciate the reviewer’s additional observations and suggestions, which have been instrumental in enhancing the quality of our work. All issues raised by the reviewer have been thoroughly addressed.

Reply to comments of reviewer #2:

Reviewer Comment 1:

Following my comments authors have included in this revised version WBs showing the extent of many SFPQ depletions. However, now they use RNaseH1 depletion in several experiments and it would also be nice to show the extent of this depletion at least in a WB.

Reply to reviewer Comment 1:

To address the reviewer’s comment, we have included representative Western blot analyses demonstrating the efficacy RNaseH1 knockdown (**NEW SUPPLEMENTARY FIGURE 1d**). SFPQ knock-down efficacy is shown in **Supplementary figures 3h, I, n** and **Supplementary figure 7j**. The related text can be found in the results section in **lines 99-101**.

Reviewer Comment 2:

Gene descriptions on Suppl. Table 7 are cut and incomplete

Reply to reviewer Comment 1:

The complete Supplementary Table 7 is now provided as a downloadable Excel file.

Reviewer Comment 3:

-Lines 631 and following “... To evaluate target specificity of the S9.6 antibody, U-2 OS cells feed on cover slips were treated with RNaseT1 (NEB, 1U/μl; NEBuffer 2), RNaseT3 (NEB, 1U/μl NEBuffer or RNaseH1 (NEB, 1U/μl NEBuffer 2) at 37°C for 30 minutes. Samples were incubated at 65 °C for 20 minutes to inactivate the enzymes, washed in 1x PBS and processed following the immunofluorescence protocol.”

It is unclear to me whether this treatment was prior or posterior to cell fixation.

Reply to reviewer Comment 3:

Sample treatment with the indicated enzymes was performed following fixation, as described in **line 155-158** of the revised STAR Methods section. The corresponding text has been edited to enhance clarity and language.

Reviewer Comment 4:

- Line 694 "Immunoprecipitated chromatin was de-crosslinked by adding 200mM NaCl and leaving samples at 65°C overnight"

Is that right? To my knowledge this is usually performed using NaHCO and not simply NaCl.

Reply to reviewer Comment 4:

We confirm that our protocol uses 200mM NaCl during the decrosslinking phase.

Reviewer Comment 5:

- Line 910 "Sonication was performed with BioRuptor (Diagenode) to obtain DNA fragments in a range between 150-300 bp."

Those are harsh conditions to check for R-loops in genomic DNA and in particular for detection of large R-loops (>300 bp) that are sensitive to sonication. Were these conditions really used? If so, authors are probably underestimating R-loop abundance.

Reply to reviewer Comment 5:

We thank the reviewer for highlighting these errors. The indicated fragment size refers to ChIP-seq experiments. For the DRIP-PCR experiment, DNA was sheared to an average size of 500-1200 base pairs. This error has been corrected in the revised manuscript (revised STAR Methods, **line 243**).

Reviewer Comments 6-13:

Mistakes:

I found many writing mistakes (largely in STAR methods section) to be corrected. Some examples are:

Comment: Fig 7G nFS control is labeled in red p=0.0078. Is this a mistake?

Reply: We intentionally used red coloring to indicate that the p-value refers to differences between SFPQ nFS and control DNA-transfected cells. This is explained in the legend related to **Figures 5f**.

Comment:

The sentence "Briefly, cells were lysed in RIPA buffer (50mM Tris pH 7.4, 250mM NaCl, 1% Triton-X, 1% DOC, 0.1% SDS), sonicated (Fisherbrand Model 120 Sonic 563 Dismembrator, Fisher Scientific, 20 seconds, 20% amplitude)."

looks like inconsistent to me.

Reply: We have corrected this sentence in the material and methods section (**line 79 – 84**) of the STAR Methods section of the revised version of the manuscript.

Comment: Authors frequently misuse the verb "to result in".

Reply: The corresponding positions in the text was corrected accordingly throughout the revised version of the manuscript.

Comment: Line 492: CO2 for CO₂

Reply: The corresponding position in the text was corrected accordingly (**line 5**, revised STAR Methods)

Comment: Line 579: siring for syringe

Reply: The corresponding position in the text was corrected accordingly (**line 101**, revised STAR Methods)

Comment: Line 630 “recombinant RNaseH1 (NEB) for 1 hour at 37°C, according to manufacturer<sb> instruction<sb>,..”

Reply: The corresponding position in the text was corrected accordingly (**lines 154-158**, revised STAR Methods)

Comment: Line 652 “temperature. Cells were blocked with 5%BSA (in 1x BS) at 37°C for 20 minutes”

Reply: The corresponding positions in the text was corrected accordingly (**line 176**, revised STAR Methods)

Comment: Line 657 “citric acid, 82 mM Na₂HPO₄. Final pH 7.0), deionize<db> formamide 70% final concentration”

Reply: The corresponding position in the text was corrected accordingly (**line 180-184**, revised STAR Methods)

Comment: Line 782 “The pull-down experiments validate of His6-myc SFPQ with GST-DAXX-HA were performed in SEC Buffer” This sentence looks like incomplete

Reply: The corresponding position in the text was corrected accordingly (**line 314-317**, revised STAR Methods)

Comment: Line 816 “RNase inhibitors RNaseOUT (0.16 units/μl, Thermo Fisher) was added to reactions containing RNA oligonucleotides.”

Either RNase inhibitor ... was added or RNase inhibitors ... were added

Reply: The corresponding position in the text was corrected accordingly (**line 351-352**, revised STAR Methods)

Comment: Line 826 “Subsequently, cells were fixed by dropping MeOH/Acetic Acid (ratio 3:1) under constant agitation three times, altered by centrifugation at 1000g, 5 minutes, 4°C, and dropped on slides to obtain metaphase spreads.”

What do authors mean by “altered by centrifugation”?

Reply: The corresponding position in the text was corrected accordingly (**line 362-365**, revised STAR Methods)

Comment: Line 872 “Cell<sb> were permeabilized with Cytobuffer..”

Reply: The corresponding position in the text was corrected accordingly (**line 395-398**, revised STAR Methods)

Comment: Line 894 “..sucrose, 3mM MgCl.”

Reply: The corresponding position in the text was corrected accordingly (**line 412**, revised STAR Methods)

Reviewer #3 (Remarks to the Author):

General Comment:

The authors have provided substantial additional experimental data and extensively detailed rebuttal to address all of my previous concerns. The revised manuscript provides interesting new findings on the role of SFPQ in control of RNA-DNA hybrids, recruitment of DAXX-H3.3 to repetitive DNA regions, and the activation of the STING pathway. The data provided is sufficiently high technical quality to support the conclusions.

Reply to general comment

We thank Reviewer 3 for the positive comments on the revised version of our manuscript, noting that the current version meets the quality standards required for publication in *Nature Communications*.

Reply to comments of reviewer 3:

Reviewer 3 has not identified any shortcomings in the manuscript.

Reviewer #4 (Remarks to the Author):

General Comment:

The authors have made a substantial effort to improve the original version of the manuscript and have responded clearly to most of the reviewers' previous comments. I support publication in principle; however, several important issues remain that must be addressed before acceptance.

Reply to general comment

We thank Reviewer 4 (who replaced the original Reviewer 1) for the positive comments on the revised version of the manuscript and for the additional suggestions to further improve its quality. Below, we present a series of experiments that address the reviewer's concerns.

Reply to comments of reviewer 4:

Reviewer Comment 1: DAXX/SFPQ Interaction Timing:

The authors have not clarified at which stage of the cell cycle the DAXX/SFPQ interactions occur.

Reply to reviewer Comment 1:

Reviewer 4 refers to a comment from original reviewer 1 that requested information on SFPQ-DAXX interaction related to transcription, replication and during the cell cycle. To provide information on SFPQ-DAXX interaction related to transcription and replication, we treated U-2 OS cells with hydroxyurea (blocks DNA replication) or arrested transcriptional elongation using alpha-amanitin. Both conditions increase R-loop levels as demonstrate by control immunofluorescence analysis using S9.6 monoclonal antibodies (**Figure for reviewer 1A**). Performing super-resolution microscopy, we found elevated SFPQ-DAXX colocalization events in both treatment conditions (**Figure 1B for reviewers**). These findings provide support for DAXX-SFPQ interaction upon R-loop formation triggered by alterations of transcription and DNA replication.

We do not present data on the cell cycle–dependent complex formation, as we are currently investigating DAXX-SFPQ function in different biologically relevant conditions including drug-treatments. We therefore prefer not to anticipate such results at this stage. Nonetheless, our manuscript presents multiple lines of evidence supporting a coordinated role for SFPQ and DAXX in preserving genome stability by suppressing R-loop mediated replication stress.

Figure for reviewers 1: SFPQ-DAXX localization upon alteration of replication and elongation of transcription

(A) Treatment of U-2 OS cells with hydroxyurea and α -amanitin causes elevated R-loop levels, as indicated by immunofluorescence analysis using S9.6 monoclonal antibodies. Treatment of fixed cells with recombinant RNaseH1 prior to antibody staining validates antibody specificity. Representative images are shown. (B) Combined immunofluorescence with anti-SFPQ and anti-DAXX antibodies in U-2 OS cells treated with hydroxyurea (HU, 16 hrs) or α -amanitin (4 hrs). Representative images are shown (left panels); quantification of colocalization events (right panel). Both treatment conditions induce significant increase in colocalization events. Images were obtained by super-resolution microscopy; N= number of independent experiments; n= number of analysed nuclei; a Student's t-test was used to calculate statistical significance; p-values are shown.

Reviewer Comment 2:

The authors did not explain why depletion of SFPQ (siSFPQ) leads to an increase in DAXX protein expression levels. A mechanistic rationale or supporting experimental data should be provided.

Reply to reviewer Comment 2:

In the revised manuscript, we demonstrate that RNAi-mediated depletion of SFPQ results in a ~20% increase in DAXX protein levels as demonstrated by quantification data shown in **MODIFIED SUPPLEMENTARY FIGURE 3h, I; results section lines 185-187**. Elevated DAXX expression is consistent with previous studies reporting that Type I interferon signaling mediates DAXX upregulation (PMID: 23302889). Additional studies further support this connection (PMID: 11420043, PMID: 12391177). Our RNA-Seq data, together with qRT-PCR, reveal an enhanced interferon response in SFPQ-depleted U-2 OS cells (**Fig. 7, Supplementary Fig. 7**). To further substantiate the activation of interferon signaling upon SFPQ depletion, we used our RNA-Seq data to perform detailed analyses of enriched

Gene Ontology categories and the expression profiles of individual interferon response genes to the reviewers (**Figure for reviewers 2**). Our data confirms activation of Type I interferon signaling in SFPQ depleted cells (**Figure for reviewers 2**). We feel confident that the reported connection between Type I IFN signaling and DAXX expression provides an explanation of elevated DAXX expression in SFPQ depleted cells. The cited PMID: 23302889 has been incorporated into the manuscript as a **new reference 64**. The connection between DAXX expression and Type I IFN signaling is discussed in **lines 335-336** of the revised version of our manuscript.

Reviewer Comment 3: Cell Line Specificity of G2/M Arrest

The authors have not indicated whether G2/M arrest following SFPQ depletion is observed in cell lines other than U2OS. Extending this analysis would strengthen the generality of their findings.

Reply to reviewer Comment 3:

We performed cell cycle analyses on SFPQ depleted U-2 OS cells and MCF10A cells, that were used for the assessment of metaphase defects. SFPQ knock-down in U-2 OS cells resulted cell cycle defects as previously reported for this cell line (**references 53-55 in manuscript; NEW SUPPLEMENTARY FIGURE 1g**). Cell cycle analysis of SFPQ-depleted MCF10A cells (primary, immortalized, non-transformed human mammary epithelial cells) revealed delayed progression through S phase and M phase (**NEW SUPPLEMENTARY FIGURE 6h**). Results are explained in **lines 108-109 (U-2 OS) and 320-321 (MCF 10A)**.

Reviewer Comment 4: Validation of STING Expression Controls

In Reviewer Figure 3, OVCAR4 cells are used as a positive control for STING expression. It would have been preferable to include siRNA-mediated STING knockdown as an additional control to confirm band specificity. Furthermore, the qPCR results for STING (Figure 7D) should be validated by western blot analysis, especially since the authors have demonstrated that they possess a functional antibody (Reviewer Figure 1).

Reply to reviewer Comment 4:

Original Supplementary figure 7j shows total and phospho-STING protein levels in experimental cells. We have performed requested control experiments related to STING antibody specificity as shown in the **new Figure for reviewers 3**. We demonstrate reduced expression of STING after transfection of U-2 OS cells with STING specific siRNAs (40 μ M). Transfection of control siRNAs (10-40 μ M) does not alter STING expression. Importantly, anti-STING bands in U-2 OS cells appear at same molecular weight as in

OVCR-4 cells (that are characterized by high STING expression). We conclude that used STING antibodies have sufficient specificity to support the conclusions of our manuscript.

Figure for Reviewer 3: Western blot analysis demonstrating the specificity of STING antibodies U-2 OS and OVCAR-4 cells were transiently transfected with the indicated siRNAs. Increasing concentrations of control siRNAs did not alter STING expression, thereby excluding indirect effects on STING levels due to siRNA loading. In contrast, transient transfection with STING-specific siRNAs efficiently reduced STING protein expression. OVCAR-4 cells displayed high basal levels of STING, which were markedly decreased following transfection with STING-specific siRNAs. STING-specific bands in both U-2 OS and OVCAR-4 cells were detected at the same molecular weight. To verify equal protein loading, the Ponceau-stained western blot membrane is shown. SE, short exposure; ME, medium exposure; LE, long exposure

Reviewer Comment 5: SFPQ Knockdown and CHIP Signal Interpretation

The authors' response regarding the limited reduction of SFPQ CHIP signals upon SFPQ knockdown (Supplementary Figure 6D) could be strengthened. It would be helpful to clarify whether the total amount of immunoprecipitated chromatin/DNA decreases in SFPQ CHIP after SFPQ knockdown.

Reply to reviewer:

During the course of CHIP experiments an aliquot of crosslinked chromatin (starting material) obtained from control and SFPQ knock-down cells was reverse crosslinked and obtained DNA was quantified. This enabled us to provide a good estimate on the "starting concentration" of crosslinked chromatin preparations from all experimental samples. For all CHIP seq experiments on SFPQ depleted and control U-2 OS cells, chromatin corresponding to 1 μg of reverse crosslinked DNA was used. We provide reviewers with the total amount of DNA recovered from reverse crosslinked CHIP starting material and from immunoprecipitated chromatin (using anti-SFPQ, DAXX or H3.3 antibodies). As expected, SFPQ depletion resulted reduced amount of immunoprecipitated chromatin when SFPQ, DAXX or H3.3 specific antibodies were used for immunoprecipitation. 10ng of reversed crosslinked DNA was used for library preparation and massive parallel sequencing (**Table 1 for reviewers**). We add details on these CHIP procedures to the STAR Methods section (**Lines 207- 232**).

Biological replicate 1						
siControl				siSFPQ		
Total obtained (ng)	5250			4610		
ChIP antibodies	Used in ChIP (ng)	Amount immuno-precipitated (ng)	used for ChIP seq library preparation (ng)	Used in ChIP (ng)	Amount immuno-precipitated (ng)	used for ChIP seq library preparation (ng)
DAXX	1000	69,1	10	1000	45,3	10
SFPQ	1000	93,1	10	1000	31,05	10
H3.3	1000	73,9	10	1000	58,9	10

Biological replicate 2						
siControl				siSFPQ		
Total obtained (ng)	3060			3070		
ChIP antibodies	Used in ChIP (ng)	Amount immuno-precipitated (ng)	used for ChIP seq library preparation (ng)	Used in ChIP (ng)	Amount immuno-precipitated (ng)	used for ChIP seq library preparation (ng)
DAXX	1000	19	10	1000	12	10
SFPQ	1000	31,7	10	1000	17,2	10
H3.3	1000	31,1	10	1000	21,7	10

Table 1 for reviewer. Quantification of DNA prepared from reverse crosslinked DNA before and after immunoprecipitation. Amount of DNA used for library preparation is indicated. Values for 2 biological replicates are shown.

Reviewer Comment 6: Minor Comment – Supplementary Figure 1F

Supplementary Figure 1F appears to show SFPQ staining, yet the figure legend indicates measurement of S9.6 staining intensity. The image and legend should be aligned to accurately reflect the data presented.

Reply to reviewer Comment 6:

We thank the reviewer for the careful inspections of the figures of the manuscript. We have corrected the labelling of the figure panel.

POINT BY POINT REPLY TO REVIEWER COMMENTS

Reviewer #1

We did not receive request for modification of the manuscript from reviewer 1

Reviewer #2

We did not receive request for modification of the manuscript from reviewer 2

Reviewer #3

We did not receive request for modification of the manuscript from reviewer 3

Reviewer #4 (Remarks to the Author):

General Comment:

I am satisfied with the author's responses to my comments, and support publication.

Reply to general comment:

We thank Reviewer 4 for supporting the publication of our manuscript for publication in *Nature Communications*.

We appreciate the effort of the reviewer during the revision phase.